# Black Box Uncertainty Analysis for Semantic Segmentation

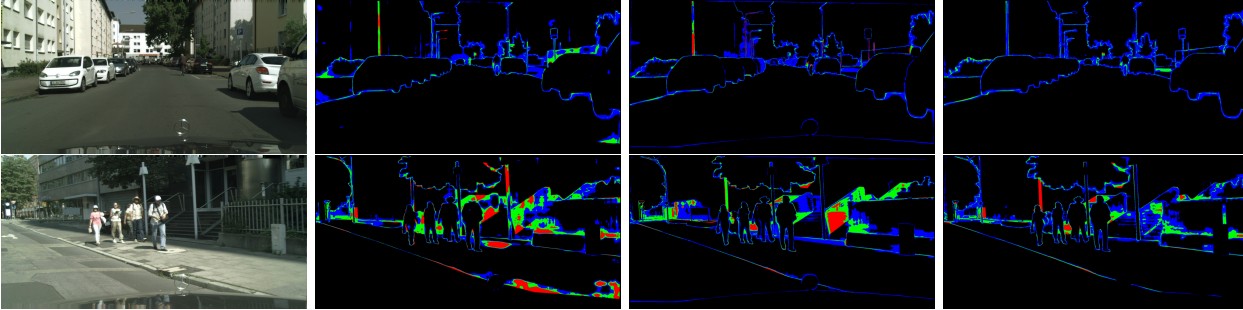

Figure 1: A couple of examples show how well the entropy of predicted outputs correlates with misclassified pixels. Green indicates misclassified pixels with high entropy, red shows misclassified pixels with low entropy, and blue shows correctly classified pixels with high entropy. Images in each row show the input and outputs from DRN D-22, OneFormer ConvNeXt-L and SegFormer B5, respectively. We note that entropy captures misclassified pixels with high recall.

## Abstract

Semantic segmentation has become an important task in computer vision with the growth of self-driving cars, medical image segmentation, etc. Although current models provide excellent results, they are still far from perfect. While there has been significant work in trying to improve the performance, both with respect to accuracy and speed of segmentation, there has been little work which analyses the failure cases of such systems. In this work, we aim to provide an analysis of how segmentation fails across different models and consider the question of whether these can be predicted reasonably at test time. To do so, we explore existing uncertainty-based metrics and see how well they correlate with misclassifications, allowing us to define the degree of trust we put in the output of our prediction models. Through several experiments on three different models across three datasets, we show that simple measures such as entropy can be used to capture misclassification with high recall.

## 1 Introduction

Semantic segmentation is defined as a task which involves taking an image and labelling each pixel of the image as belonging to one of a set of predefined classes. With the advent of self-driving cars, path-finding robots, and various other such problems, semantic segmentation has become essential in the field of computer vision, as all of these applications rely on being able to segment, parse and understand the scene they see in order to take appropriate action. Semantic segmentation is even used in the medical field to segment areas of medical images for analysis. When used for such critical tasks, it is of utmost importance that the models we propose provide us with accurate results or some confidence metrics which can be used to rely on their outputs.

Most of the earlier approaches to semantic segmentation involve using Convolutional Neural Networks (CNN) to extract features and predict the segmentation class for each pixel. Examples of such approaches include Chen et al. (2014); Yu et al. (2017); Chen et al. (2017). Some of these approaches also utilise Conditional Random Fields (CRF) to improve their results further. More recent approaches use Transformer-based

architectures such as Cheng et al. (2021); Xie et al. (2021); Jain et al. (2023b;a). Cheng et al. (2021) also utilises masked predictions for each class rather than simple pixel-wise multi-class predictions.

Although we now have a wealth of approaches that target semantic segmentation, the uncertainty quantification for their predictions is relatively less well-studied. This is especially true during test time since we cannot judge the model against ground truth predictions. For example, consider the case when there is a domain shift in the inputs during test time; how do we trust the output of our network without any form of confidence or trust metric?

There have been works which look at uncertainty in deep neural networks in general (Gawlikowski et al., 2023; Grathwohl et al., 2019; Hendrycks & Gimpel, 2016), including relatively new ones like Vazhentsev et al. (2022), which specifically considers uncertainty in Transformer networks. For semantic segmentation, we have works such as works such as Siddiqui et al. (2020) and Wang et al. (2021) which consider using entropy for the purposes of active learning and domain adaptation, respectively. Similarly, Xia et al. (2020); Rahman et al. (2022), aim to detect failures and out-of-distribution samples. However, they use separate networks that require additional training. Our work explicitly focuses on obtaining such information at test time without any extra networks or training. By this, we mean that without any knowledge of the architecture, at test time, we wish to know whether our model has succeeded in semantic segmentation or not. Considering the gravity of this task, we believe that there should be a study of the same for the sake of understanding these networks better.

In this paper, we first look at where these segmentation networks fail to predict the correct classes (Section 3). We then ask questions such as, *"Knowing where these networks fail, can we predict these failures?"*, and *"How well do current approaches of gauging uncertainty correlate with misclassification?"*. To this end, we perform a series of experiments, analyse the results and show that certain uncertainty methods can be used to decide how much we should trust the output of our networks (Section 5.3) and that we can improve upon our results by using calibrated models (Section 5.4). We show that these techniques work even when we consider domain shifts (Section 5.5) or noise (Section 5.6) in the input. For instance, when performing transfer learning on a network trained on Cityscapes to the Dark Zurich dataset, we validate that we can predict the specific cases of failure in segmentation well. We hope our work provides insight into existing segmentation networks and helps drive further research to ensure their trustworthiness. Figure 1 shows some examples of our approach where we use entropy to judge whether a pixel is likely to be misclassified. We note that simple entropy can identify the likely misclassified regions with high recall. Further detailed analysis is presented in Section 5.

## 2 Related Works

Semantic segmentation has been researched by the vision community for over a decade, and has been well documented in survey papers such as Guo et al. (2018); Hao et al. (2020). However, we will primarily concern ourselves with relatively recent works that use deep neural networks for performing this task. Among these, most earlier works, such as Chen et al. (2014); Long et al. (2015); Yu et al. (2017); Zhao et al. (2017); Chen et al. (2017); Yang et al. (2018), are convolutional in structure and generally make use of existing Convolutional Neural Networks (CNN) such as VGG (Simonyan & Zisserman, 2014), ResNet (He et al., 2016), ResNeXt (Xie et al., 2017), for feature extraction purposes followed by combining extracted features, possibly at multiple scales, in novel ways. Generally, the classification is done at a lower resolution than the input image and is upsampled back to the original dimensions near the end of the pipeline. The improvement in performance across these approaches mainly derives from coming up with new novel ways of pooling and combining the extracted features, such as the usage of Dilated Convolutions, Pyramid Pooling and Atrous Spatial Pyramids, among others.

Newer approaches such as Cheng et al. (2021); Xie et al. (2021); Strudel et al. (2021); Gu et al. (2022); Jain et al. (2023b;a), are based on Transformer networks introduced in Vaswani et al. (2017). Initially proposed for natural language processing tasks, Transformers also proved effective in computer vision. These approaches follow two major pipelines for extracting features from the input. Either they use CNN feature extractors, much like earlier works, followed by Transformer decoders for refinement. Alternatively, they use ViT-like (Dosovitskiy et al., 2020) architectures where the input image is divided into patches, which are directly fed to a Transformer encoder for feature generation, followed by a Transformer decoder and segmentation head.

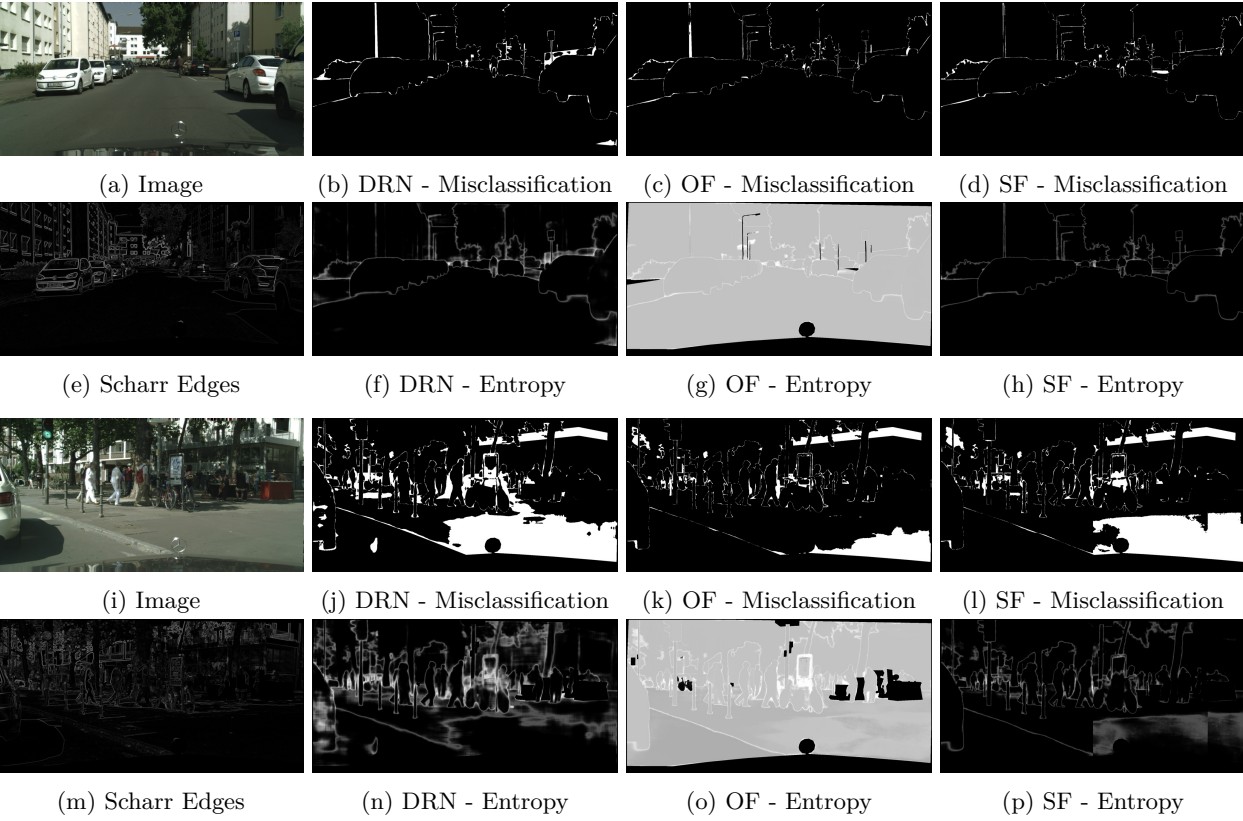

Figure 2: Comparison of misclassified pixels and entropy of DRN D-22 (DRN), OneFormer ConvNeXt-L (OF) and SegFormer B5 (SF) networks on a couple of Cityscapes validation images, along with the images themselves and edges detected using the Scharr operator. It is to be noted that OneFormer generally produces high entropy outputs, which cause most of the image to be grey. However, we can see that the highest entropy regions, in bright white, still correspond well to misclassified pixels.

We also consider how to extract uncertainty information from deep neural networks and refer to some existing works. Works like Deep Ensembles (Lakshminarayanan et al., 2017) and Bayesian Networks (Blundell et al., 2015) allow us to measure the uncertainty of predictions relatively easily, but they require special training procedures and are not applicable to already existing networks. However, Gal & Ghahramani (2016) shows that Dropout layers can be used to generate multiple predictions for a given input and thus allow us to compute a certain measure of model uncertainty. Similarly, Houlsby et al. (2011) proposed a metric originally meant for active learning but can also be used as an uncertainty measure for existing networks.

To the best of our knowledge, this type of work focusing on semantic segmentation has not been done earlier. While there have been works such as Siddiqui et al. (2020) and Wang et al. (2021) which utilise uncertainty for active learning and domain adaptation purposes in semantic segmentation, they do not aim to predict misclassifications. Works such as Jammalamadaka et al. (2012); Parikh & Zitnick (2011); Vazhentsev et al. (2022); Xia et al. (2020); Rahman et al. (2022) are closest to our work. Vazhentsev et al. (2022) proposes and evaluates several uncertainty metrics for gauging the uncertainty of Transformer-based architectures and has similar goals of misclassification detection using uncertainty; however, the paper only explores these within the setting of Transformer architectures, and while most newer segmentation architectures are Transformer-based, older CNN-based networks are still used. Thus, we focus on making our evaluation and analysis as general as possible.

## 3 Failure Analysis

Before we develop a trust score or metric, we must first look at and try to understand the failure cases of our models. We consider a few standard segmentation networks - Dilated Residual Networks (Yu et al., 2017)

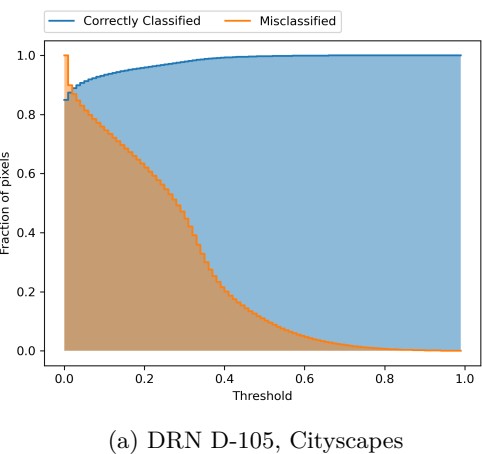
(a) DRN D-105, Cityscapes

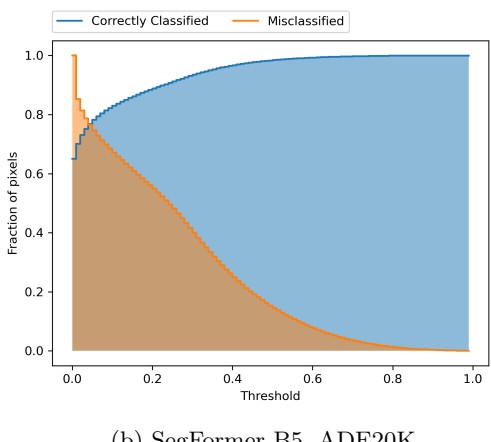
(b) SegFormer B5, ADE20K

Figure 3: A plot showing the fraction of correctly classified (blue) and misclassified (orange) pixels with uncertainty values below and above the given value, respectively. Thus, if we choose any threshold value along the x-axis we can directly read what fraction of correctly classified and misclassified pixels are captured.

(DRN), OneFormer (Jain et al., 2023a) and SegFormer (Xie et al., 2021) and look at their outputs. We specifically choose a mix of older and newer networks for this since they are quite different architecturally and should allow us to focus on generalities rather than the peculiarities of any single architectural family. We use pre-trained weights on the Cityscapes dataset (Cordts et al., 2016) for all networks and observe their outputs on the validation set. Figures 2b to 2d and 2j to 2l highlight the misclassified pixels for the networks on a couple of such images. Black indicates correctly classified or ignored pixels, whereas white indicates misclassified pixels.

From the images, it is clear that OneFormer and SegFormer are better than DRN at segmenting images. Indeed, mIoU for DRN D-22, OneFormer ConvNeXT-L, and SegFormer B5 are 0.6790, 0.8287, and 0.8239, respectively. However, we note that the failure modes for the networks are very similar. All of these networks misclassify pixels in similar regions of the images. It is also evident that several of these misclassifications lie along the edges of various objects in the images. This makes intuitive sense since it can be challenging, even for humans, to assign the boundary pixels to a particular object exactly.

This naturally brings us to the question, *"Can we reasonably predict which pixels are most likely to be misclassified?"*. Even if we cannot accurately predict which pixels will be misclassified, we would like to have some measure of confidence in these classifications. Since we would like to apply whatever approach we come up with on existing networks without any re-training or finetuning, we need to consider what kind of information we can extract and use from just the images and their segmentation predictions since that is all we have during test time. Edge detection may seem an obvious first choice, as we have already established that several misclassified pixels lie along object boundaries. A better choice is some measure of uncertainty, such as entropy.

Figures 2e and 2m show the edges obtained when the Scharr filter is applied to the images. The gradient magnitudes thus obtained are scaled between $[0, 1]$, with 0 being black and 1 being white. All ignored pixels are also in black. Since the edges are obtained from just the image rather than the predictions, the network used does not matter. This already tells us that edge detection may not be particularly effective in predicting misclassifications, which depend on the segmentation network used. While some edges align well with the misclassifications, most are correctly classified, especially the edges within each object. The fact remains that only the edges that align with ground truth label boundaries are misclassified. Moreover, by its very nature, edge detection completely fails to capture any misclassified pixels within objects; this is especially evident in the second image (Figure 2m).

On the other hand, the simple entropy of predicted labels, as shown in Figures 2f to 2h and 2n to 2p, is far more aligned with the misclassifications. Similar to edges, the entropy is scaled between $[0, 1]$ for each

image and ignored pixels are set to 0. Here, we see a marked difference between OneFormer and the other two networks; DRN and SegFormer provide very confident predictions and result in most pixels having very low entropy, whereas OneFormer's classifications are under-confident, resulting in most pixels having higher entropy. However, even then, at a glance, we notice that the highest entropy pixels align well with misclassifications.

For a more quantitative look, we plot the fraction of correctly classified and misclassified pixels with entropy values lower than and higher than a given threshold in Figure 3. The plot allows us to directly read the percentage of correct and incorrect pixels predicted if we threshold entropy at any given value and consider anything below the threshold to be correctly classified and anything above it to be misclassified.

With these observations in mind, we focus on quantitative analysis of various uncertainty measures, which can be derived at test time, on pre-trained networks and consider how well they manage to predict misclassified pixels.

## 4 Uncertainty Metrics

We now describe the uncertainty metrics used for our experiments and how they are calculated. We only consider metrics that can be obtained purely at test time and those that work with nearly all neural networks. This rules out any approach for gauging uncertainty that requires us to change the network structure or fine-tune it. Specifically, we consider the following techniques.

- **Probability Scores** - Some of the simplest possible uncertainty metrics we can obtain involve directly using the highest probability scores output by the network (Hendrycks & Gimpel, 2016). We consider two different but related metrics. The first one is often called the Variation Ratio and is defined as follows:

$$\text{VR} = 1 - p_m \tag{1}$$

where, $p_m = \max_{k=1}^{K}(p_k)$ represents the highest probability across all classes, and $p_k$ represents the probability of a pixel belonging to class $k$. The second metric, Probability Margin, takes into account the difference between the highest and the second highest probability and is defined as:

$$\text{PM} = 1 - \left\{ p_m - \max_{k=1, k \neq m}^{K}(p_k) \right\} \tag{2}$$

- **Entropy** - Another simple and possibly the most commonly used uncertainty metric we can obtain from a model. The only requirement is that the model outputs a probability distribution per pixel over all classes, which is true for nearly all (if not all) semantic segmentation models. Formally, the entropy for a single pixel is defined as:

$$\text{H} = -\sum_{k=1}^{K} p_k \log p_k \tag{3}$$

where, $K$ is the number of classes and $p_k$ represents the probability of the pixel belonging to class $k$.

The above-mentioned techniques can be used with just a single output mask per image, but for metrics like BALD we need multiple outputs. To generate such outputs for each input image, we consider the following techniques.

- **Monte Carlo Dropout (MC Dropout) (Gal & Ghahramani, 2016)** - Typically, Dropout layers (Hinton et al., 2012) in neural networks are disabled during test time and replaced with a simple scaling of the features. MC Dropout utilises these layers to generate multiple predictions for the same image by keeping the Dropout layers active during test time. This introduces a source of randomness in the network, which allows it to produce different outputs for the same input on each forward pass. However, one drawback of the technique is that it can only be used with networks trained with Dropout layers.

- **Noise** - Yet another approach to obtaining multiple outputs is to introduce a tiny amount of random noise in the input, which perturbs the output. Like MC Dropout, this allows us to generate multiple predictions for the same input by using multiple forward passes (each with a different noise). For our purposes, we use zero mean Gaussian noise, with a small standard deviation.

- **Scaling** - Segmentation networks already use multi-scale inputs to improve their performance (Jain et al., 2023a; Yu et al., 2017). Essentially, it involves running forward passes on multiple scales of the input, which generate multiple segmentations for the same image. The networks can then scale back the outputs to the original resolution and average them to get better results than a single pass. We can, however, keep these generated masks for computing uncertainty.

Once we obtain multiple predictions for each input image, we can extract uncertainty information from these in the following manner.

- **Averaged Probability Scores** - Given multiple probability distributions over a pixel, we can take their mean to obtain a single representative probability distribution for said pixel. Mathematically we have:

$$\overline{p_k} = \frac{1}{N} \sum_{i=1}^{N} p_k^i \tag{4}$$

where $\overline{p_k}$ represents the average probability of a pixel belonging to class $k$, and $p_k^i$ represents the probability of the pixel belonging to class $k$ for the $i^{\text{th}}$ prediction. $N$ is the total number of predictions. Once we have the average probability distribution per pixel, we can compute Variation Ratio and Probability Margin values over them as follows:

$$\text{VR} = 1 - \overline{p_m}, \quad \text{where} \quad \overline{p_m} = \max_{k=1}^{K}(\overline{p_k}) \tag{5}$$

$$\text{PM} = 1 - \left\{ \overline{p_m} - \max_{k=1, i \neq m}^{K} (\overline{p_k}) \right\} \tag{6}$$

- **Averaged Entropy** - Similar to averaged probability scores, we can use Equation (4) to obtain the average probability distribution per pixel and compute entropy over it as:

$$\text{H} = - \sum_{k=1}^{K} \overline{p_k} \log \overline{p_k} \tag{7}$$

- **Variance** - Variance of the probability distribution has also been used as a measure of uncertainty in Bayesian settings (Gal et al., 2017; Smith & Gal, 2018). Since we have multiple predictions, we can compute the empirical variance of class $k$ for any given pixel as:

$$\sigma_k^2 = \frac{1}{N} \sum_{i=1}^{N} (p_k^i - \overline{p_k})^2 \tag{8}$$

where, $\overline{p_k}$ is calculated as in eq. (4). To obtain a single value per pixel, we consider taking both the average as well as the maximum across across all classes.

- **Bayesian Active Learning by Disagreement (BALD)** (Houlsby et al., 2011) - Originally proposed as an acquisition function for selecting samples in a Bayesian active learning setting, BALD can also be used as an uncertainty metric. The BALD score is defined as:

$$\text{BALD} = - \sum_{k=1}^{K} \overline{p_k} \log \overline{p_k} + \frac{1}{N} \sum_{k=1, i=1}^{K,N} p_k^i \log p_k^i \tag{9}$$

where, $\overline{p_k}$ and $p_k^i$ are same as defined earlier. Essentially, it checks the disagreement between the entropy of the expected prediction and the expected entropy across all predictions. Higher disagreement represents more uncertainty.

# 5 Experiments and Observation

In this section, we discuss the various experiments performed along with their results and observe the degree of correlation between existing uncertainty metrics and misclassifications in semantic image segmentation. We also look at how well we are able to capture misclassifications when there are domain shifts in the inputs or when the input is noisy. We also consider calibrating the models to improve our results further.

## 5.1 Datasets and Models

We consider the following datasets for our experiments, as they are quite standard for image segmentation tasks and the networks we have chosen provide pre-trained weights for these datasets.

- **Cityscapes (Cordts et al., 2016)** - The primary dataset used for our experiments consists of dashcam videos from cars in several German cities targeted towards understanding urban street scenes. The validation set is comprised of 500 finely annotated images of $2048 \times 1024$ resolution. Pixels in each image are annotated as belonging to one of 30 defined classes, of which 19 are considered for classification and evaluation. Any pixels not belonging to one of these classes are effectively ignored for evaluation purposes.

- **Dark Zurich (Sakaridis et al., 2019)** - A dataset analogous to Cityscapes with a small validation set of only 50 images with resolution of $1920 \times 1080$; it consists of urban scenes from the city of Zurich at night time as opposed to Cityscapes where the images are taken at day time. We use this dataset to check how well our metrics perform when there is a domain shift of the input.

- **ADE20K (Zhou et al., 2019)** - A rather large dataset with a validation set of 2000 images of varying resolutions. Unlike Cityscapes and Dark Zurich, ADE20K has a much larger number of segmentation classes, 150 and the images are not only from dashcam videos but cover a wide range of subjects.

- **COCO (Lin et al., 2014)** - An older dataset with relatively smaller images of varying resolutions and 133 classes. It consists of 5000 validation images and provides ground-truth data for panoptic segmentation from which ground truth for semantic segmentation is extracted. Similar to ADE20K, the images cover a wide range of subjects.

We now describe the segmentation networks that were used in our experiments. We choose our models such that they range from small (DRN D-22) to large (OneFormer Swin-L) and cover a wide variety of architectures.

- **Dilated Residual Networks (Yu et al., 2017)** - Convolutional Neural Network (CNN) based on ResNet (He et al., 2016) but using Dilated Convolutions instead, which helps it achieve better performance in semantic segmentation tasks. We use the pre-trained weights for DRN D-22 and DRN D-105 variants provided by the authors for the Cityscapes dataset.

- **OneFormer (Jain et al., 2023a)** - Transformer-based architecture that utilises features from a backbone network and passes them through a Transformer decoder to get pixel-level classifications. We use pre-trained weights with ConvNeXt-L and Swin-L backbones on Cityscapes and ADE20K datasets, as provided by the authors.

- **SegFormer (Xie et al., 2021)** - Another Transformer-based architecture that uses a Hierarchical Transformer encoder and a Multi-Layer Perceptron (MLP) decoder. We use pre-trained weights of the B5 variant for Cityscapes and ADE20K datasets as provided by the authors.

## 5.2 Evaluation Metrics and Scenarios

In order to measure how well a particular uncertainty metric corresponds to misclassifications, we use Precision, Recall, Area Under the Receiver Operating Characteristic (AUROC) and Area Under the Precision Recall Curve (AUPRC) metrics. We use bootstrapping to compute 95% confidence intervals and report the

bootstrapped mean. AUROC involves plotting the False Positive Rate against the True Positive Rate at different thresholds and then computing the area covered by the curve. Similarly, AUPRC computes the area under the Precision Recall Curve at different thresholds. While AUROC and AUPRC consider multiple threshold values, we need to choose a single threshold value to binarise the uncertainty scores for computing Precision and Recall. Unfortunately, as seen in Figure 2, the scales of these uncertainty values can vary quite a lot across different models and even across different images for a single model. As such, we cannot use a single constant threshold value across all models or images. Therefore, we define a few processes to dynamically choose a threshold for each image. These methods rely on the fact that the fraction of misclassified pixels is not very high and that the misclassified and correctly classified pixels form clusters.

1. For the first method, we check the ratio of predicted misclassified pixels to total pixels at multiple uniformly spaced threshold levels and select the threshold that causes the largest change in this value.

2. We note that the ratio of predicted misclassified pixels to total pixels increases as the threshold increases. For the second method, we specify a maximum value allowed for this ratio and choose the highest threshold, which results in a ratio lower than the specified value.

3. For the third approach, we simply use K-Means on the uncertainty values to cluster them into 2 clusters and consider the cluster with lower uncertainty values as correctly classified and the other one misclassified.

Before we discuss the results, we clarify the meaning of each scenario:

- **Base** - The simplest scenario where we only consider a single forward pass. As such, only Variation Ratio, Probability Margin and Entropy can be used as uncertainty metrics for this setting.

- **Noise** - The images are injected with a Gaussian noise of zero mean and 0.01 standard deviation. The noise is injected after normalizing the images. We use ten forward passes per image in this setting.

- **Scale** - This represents using multi-scaled inputs for generating multiple outputs. We use scales of $\{0.5, 0.75, 1.0, 1.25, 1.5\}$ for our purposes.

- **Drop** - MC Dropout is applied in this scenario. Since it requires the network to have Dropout layers, we did not use this technique with DRN. Similar to the Noise scenario, we use ten passes per image.

Before moving ahead, we would like to state that although a large number of experiments under various scenarios were performed, we only highlight some of those results for brevity and compactness in the following text. The complete set of results for all the experiments is provided in the appendix for this paper. Similarly, since we obtain the metrics for each pixel rather than each image, we consider both micro-averaging and macro-averaging over the images. However, we only show the micro-averaged results in this text and provide the rest in the appendix. The appendix also includes classwise results for several experiments.

### 5.3 Primary Analysis

As AUROC allows us to get a good overview of how our uncertainty metrics correlate with misclassification, we first look at these in Figure 4 for the Cityscapes dataset. We observe that simple metrics like Entropy and Probability Scores perform better than more involved metrics such as Variance and BALD. We also look at the values of AUROC on ADE20K dataset in Figure 5. The differences are smaller between various metrics on this dataset. Across both datasets the Scale scenario performs the best. We also show a couple of scenarios on the COCO dataset in Figure 6.

We show the AUPRC-Error and AUPRC-Success results on the Cityscapes dataset for the SegFormer model in Figure 7 and Figure 8. AUPRC-Error denotes the AUPRC obtained when the misclassifications are

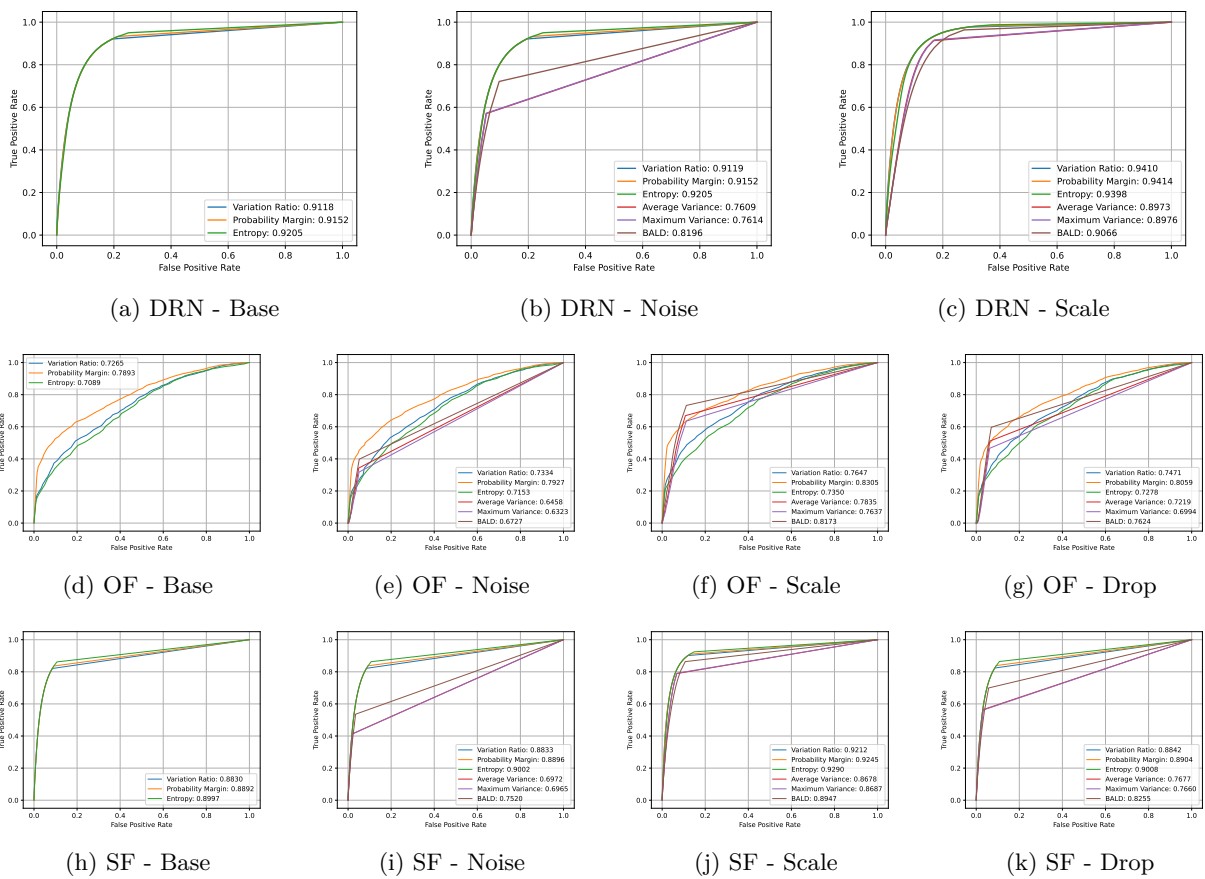

Figure 4: ROC curves on the Cityscapes dataset. The AUROC values are specified within the plot legends. We observe that we can obtain high AUROC scores even with simple uncertainty metrics such as Entropy.

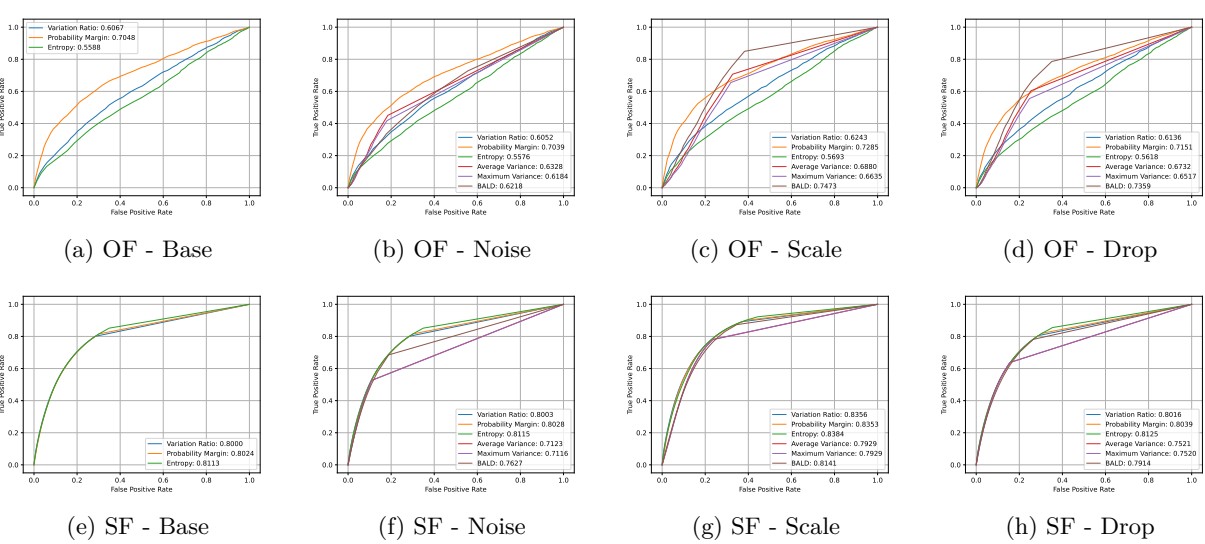

Figure 5: ROC curves on the ADE20K dataset. While we do not perform as well as on Cityscapes our AUROC values are still reasonably good.

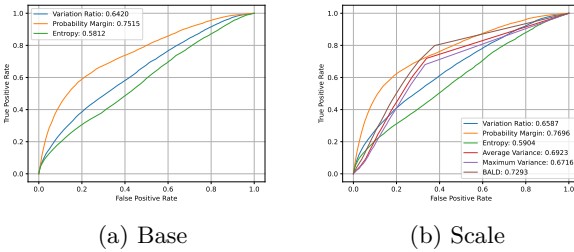

(a) Base        (b) Scale

Figure 6: ROC curves on the COCO dataset with OneFormer. We only experiment with the Base and Scale scenarios for this dataset.

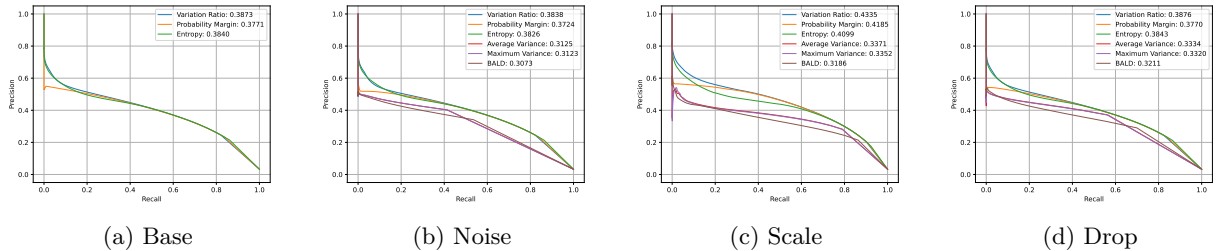

(a) Base     (b) Noise     (c) Scale     (d) Drop

Figure 7: Precision-Recall curves with misclassifications as the positive class on the Cityscapes dataset for SegFormer B5. The plots contain the AUPRC values in the legend.

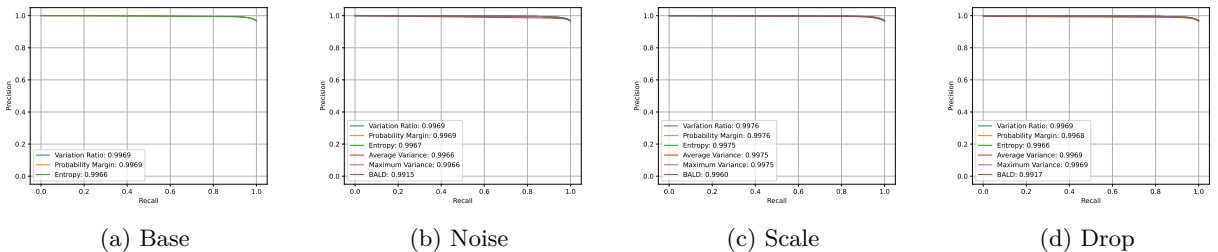

(a) Base     (b) Noise     (c) Scale     (d) Drop

Figure 8: Precision-Recall curves with correct classifications as the positive class on the Cityscapes dataset for SegFormer B5. The plots contain the AUPRC values in the legend.

Table 1: Precision, Recall and F1 Score on Cityscapes dataset using largest difference as thresholding strategy. All of the results shown use Entropy as the uncertainty metric.

| Scenario | Precision | Recall | F1 Score | Pixel % |
|---|---|---|---|---|
| | | DRN D-22 | | |
| Base | $\mathbf{17.51 \pm 0.76}$ | $95.07 \pm 1.27$ | $29.57 \pm 1.08$ | $28.95 \pm 0.81$ |
| Noise | $\mathbf{17.51 \pm 0.76}$ | $95.08 \pm 1.25$ | $\mathbf{29.57 \pm 1.07}$ | $28.95 \pm 0.81$ |
| Scale | $13.04 \pm 0.66$ | $\mathbf{98.77 \pm 0.67}$ | $23.04 \pm 1.04$ | $40.39 \pm 1.03$ |
| | | OneFormer ConvNeXt-L | | |
| Base | $14.10 \pm 1.32$ | $81.37 \pm 3.00$ | $24.03 \pm 1.93$ | $18.41 \pm 1.24$ |
| Noise | $\mathbf{14.23 \pm 1.32}$ | $80.62 \pm 3.33$ | $\mathbf{24.18 \pm 1.92}$ | $18.08 \pm 1.15$ |
| Scale | $12.10 \pm 1.07$ | $\mathbf{82.76 \pm 3.91}$ | $21.10 \pm 1.65$ | $21.83 \pm 1.41$ |
| | | SegFormer B5 | | |
| Base | $\mathbf{21.42 \pm 0.63}$ | $86.28 \pm 2.64$ | $\mathbf{34.32 \pm 0.73}$ | $13.14 \pm 0.40$ |
| Noise | $21.37 \pm 0.61$ | $86.42 \pm 2.63$ | $34.26 \pm 0.72$ | $13.20 \pm 0.40$ |
| Scale | $17.36 \pm 0.69$ | $\mathbf{92.50 \pm 2.40}$ | $29.23 \pm 0.93$ | $17.39 \pm 0.52$ |

Table 2: Precision and Recall on Cityscapes dataset using maximum percentage of pixels as thresholding strategy. All of the results shown use Entropy as the uncertainty metric.

| Scenario | Max 5% pixels | | Max 10% pixels | | Max 15% pixels | |
|---|---|---|---|---|---|---|
| | Precision | Recall | Precision | Recall | Precision | Recall |
| | | | DRN D-22 | | | |
| Base | $\mathbf{44.05 \pm 0.83}$ | $\mathbf{39.81 \pm 2.08}$ | $34.82 \pm 1.02$ | $63.72 \pm 2.81$ | $28.13 \pm 1.05$ | $76.52 \pm 2.78$ |
| Noise | $44.03 \pm 0.83$ | $39.80 \pm 2.09$ | $34.81 \pm 1.02$ | $63.72 \pm 2.86$ | $28.13 \pm 1.04$ | $76.54 \pm 2.85$ |
| Scale | $43.69 \pm 1.07$ | $39.57 \pm 1.95$ | $\mathbf{35.28 \pm 1.16}$ | $\mathbf{64.06 \pm 2.91}$ | $\mathbf{28.69 \pm 1.23}$ | $\mathbf{78.68 \pm 2.74}$ |
| | | | OneFormer ConNeXt-L | | | |
| Base | $30.86 \pm 0.89$ | $44.62 \pm 3.33$ | $23.81 \pm 0.82$ | $64.40 \pm 4.66$ | $20.05 \pm 0.78$ | $71.02 \pm 5.32$ |
| Noise | $31.27 \pm 0.92$ | $45.27 \pm 3.42$ | $\mathbf{24.17 \pm 0.88}$ | $65.43 \pm 4.27$ | $\mathbf{20.44 \pm 0.77}$ | $71.90 \pm 4.66$ |
| Scale | $\mathbf{31.46 \pm 1.04}$ | $\mathbf{46.31 \pm 3.23}$ | $23.62 \pm 0.88$ | $\mathbf{66.84 \pm 4.58}$ | $19.19 \pm 0.75$ | $\mathbf{74.20 \pm 5.13}$ |
| | | | SegFormer B5 | | | |
| Base | $36.62 \pm 0.72$ | $54.88 \pm 2.83$ | $26.09 \pm 0.71$ | $75.08 \pm 3.28$ | $\mathbf{22.41 \pm 0.66}$ | $82.65 \pm 3.05$ |
| Noise | $36.50 \pm 0.72$ | $54.66 \pm 2.86$ | $26.05 \pm 0.74$ | $74.97 \pm 3.35$ | $22.36 \pm 0.65$ | $82.69 \pm 3.10$ |
| Scale | $\mathbf{37.90 \pm 0.89}$ | $\mathbf{56.23 \pm 3.05}$ | $\mathbf{26.49 \pm 0.96}$ | $\mathbf{78.57 \pm 3.26}$ | $20.75 \pm 0.86$ | $\mathbf{87.02 \pm 2.87}$ |

Table 3: Precision, Recall and F1 Score on Cityscapes dataset using K-Means as thresholding strategy. All of the results shown use Entropy as the uncertainty metric.

| Scenario | Precision | Recall | F1 Score | Pixel % |
|---|---|---|---|---|
| | | DRN D-22 | | |
| Base | $\mathbf{36.07 \pm 0.82}$ | $71.52 \pm 2.09$ | $\mathbf{47.95 \pm 0.60}$ | $10.57 \pm 0.33$ |
| Noise | $36.03 \pm 0.82$ | $71.60 \pm 2.09$ | $47.93 \pm 0.60$ | $10.59 \pm 0.34$ |
| Scale | $30.36 \pm 1.09$ | $\mathbf{86.79 \pm 1.82}$ | $44.98 \pm 1.23$ | $15.24 \pm 0.53$ |
| | | OneFormer ConvNeXt-L | | |
| Base | $6.37 \pm 0.97$ | $72.50 \pm 3.44$ | $11.70 \pm 1.63$ | $36.43 \pm 4.13$ |
| Noise | $6.61 \pm 1.04$ | $73.08 \pm 3.44$ | $12.11 \pm 1.75$ | $35.38 \pm 4.25$ |
| Scale | $\mathbf{7.08 \pm 0.98}$ | $\mathbf{77.78 \pm 3.02}$ | $\mathbf{12.97 \pm 1.66}$ | $35.16 \pm 4.09$ |
| | | SegFormer B5 | | |
| Base | $\mathbf{35.95 \pm 0.51}$ | $62.66 \pm 2.54$ | $\mathbf{45.68 \pm 0.62}$ | $5.69 \pm 0.19$ |
| Noise | $35.81 \pm 0.52$ | $62.70 \pm 2.54$ | $45.58 \pm 0.61$ | $5.71 \pm 0.19$ |
| Scale | $32.55 \pm 0.92$ | $\mathbf{76.37 \pm 2.67}$ | $45.64 \pm 0.91$ | $7.66 \pm 0.28$ |

treated as the positive class, whereas AUPRC-Success is obtained when correctly classified pixels are treated as the positive class. Since all of our uncertainty metrics are normalized between $[0, 1]$ and higher values represent more uncertainty, we flip these values as $1 - x$ for calculating AUPRC-Success. Similar to AUROC, the Scale setting performs the best.

We now take a look at how well our first method for dynamic thresholding works. Table 1 shows the results of using the largest difference as a thresholding technique on the Cityscapes dataset. We use Entropy as the uncertainty metric. We observe that while we get very high recall rates, we also select quite a large fraction of pixels with values reaching as high as 40% for the Scale setting with DRN. As a result, precision suffers quite a bit in these cases. However, we will see later that this technique is useful when the network produces highly inaccurate outputs, such as when there is a domain shift in the input.

We now consider the performance of the second thresholding method. Table 2 shows the results for the same on the Cityscapes dataset. As expected, recall improves at the cost of precision across all scenarios when we increase the maximum fraction of pixels allowed. However, even with as few as 10% pixels, we can get 60-70% recall using Entropy as an uncertainty metric. Among the different scenarios, Scale performs better than the others, especially for Transformer-based networks.

Table 4: p-values obtained using the Wilcoxon signed-rank test to test for statistical significance of the differences in results obtained using different uncertainty metrics. The results shown are on the Cityscapes dataset with the SegFormer model for AUROC, AUPRC-Error and AUPRC-Success metrics.

| Scenario | Var. Ratio - Prob. Margin | | | Var. Ratio - Entropy | | | Prob. Margin - Entropy | | |
| | ROC | PRC-E | PRC-S | ROC | PRC-E | PRC-S | ROC | PRC-E | PRC-S |
| --- | --- | --- | --- | --- | --- | --- | --- | --- | --- |
| Base | 1.3e-83 | 6.7e-82 | 1.4e-16 | 1.3e-83 | 4.2e-03 | 3.3e-02 | 6.7e-82 | 1.4e-16 | 1.8e-02 |
| Scale | 9.6e-83 | 1.2e-69 | 1.5e-14 | 5.1e-79 | 3.2e-59 | 1.1e-04 | 5.0e-69 | **1.6e-01** | **4.7e-01** |
| Noise | 1.3e-83 | 1.1e-79 | **2.9e-01** | 1.3e-83 | **9.1e-01** | 3.9e-04 | 1.3e-83 | 9.8e-45 | 1.9e-08 |
| Drop | 1.3e-83 | 3.5e-81 | 8.2e-03 | 1.3e-83 | 1.1e-03 | 5.2e-03 | 1.3e-83 | 8.1e-37 | 1.7e-03 |

Table 5: ECE, Brier Score (BS), AUROC, AUPRC-Error and AUPRC-Success for uncalibrated and calibrated models in Base setting with Entropy as uncertainty metric for the Cityscapes dataset.

| Model | ECE ↓ | BS ↓ | AUROC ↑ | AUPRC-Error ↑ | AUPRC-Success ↑ |
| --- | --- | --- | --- | --- | --- |
| | | | OneFormer ConvNeXT-L | | |
| Uncalibrated | 0.7887 | 0.8012 | 0.7089 | 0.1193 | 0.9847 |
| Calibrated | **0.1173** | **0.0668** | **0.8387** | **0.2685** | **0.9885** |
| | | | SegFormer B5 | | |
| Uncalibrated | 0.0631 | 0.0516 | 0.8997 | **0.3840** | 0.9966 |
| Calibrated | **0.0282** | **0.0506** | **0.9357** | 0.3382 | **0.9975** |

Table 3 shows the results of the third thresholding strategy on the Cityscapes dataset using Entropy as the uncertainty metric. We observe that K-Means as a thresholding strategy works quite well on DRN and SegFormer but fails quite badly on OneFormer. However, when it works well, we get upwards of 40% F1 score.

We also perform statistical significance testing using the Wilcoxon signed-rank test for some of the metrics and display the results in Table 4. Since most of the p-values are well under the 95% confidence level, we highlight those that are not in bold.

## 5.4 Calibration Experiments

In order to improve our scores, we also consider calibrating the networks we are using. We compute the Expected Calibration Error and Brier Score of all the networks tested and find that OneFormer, especially, is not well-calibrated. We use temperature scaling on the logits while computing softmax to counter this

Table 6: Precision, Recall and AUROC on the Dark Zurich dataset, with Entropy as the uncertainty metric.

| Scenario | Largest Difference | | | | AUROC |
| | Precision | Recall | F1 Score | Pixel % | |
| --- | --- | --- | --- | --- | --- |
| | | | DRN D 22 | | |
| Base | 68.92 | 80.93 | 74.44 | 77.21 | 0.6054 |
| Noise | 68.77 | **80.95** | 74.37 | 77.39 | 0.6024 |
| Scale | **71.92** | 79.24 | **75.40** | 72.45 | **0.6680** |
| | | | OneFormer ConNeXt-L | | |
| Base | 38.41 | 56.71 | 45.80 | 42.67 | 0.5667 |
| Noise | 41.33 | 56.58 | 47.77 | 39.57 | 0.5641 |
| Scale | **42.47** | **73.21** | **53.76** | 49.82 | **0.6377** |
| Drop | 39.10 | 57.44 | 46.53 | 42.46 | 0.5755 |

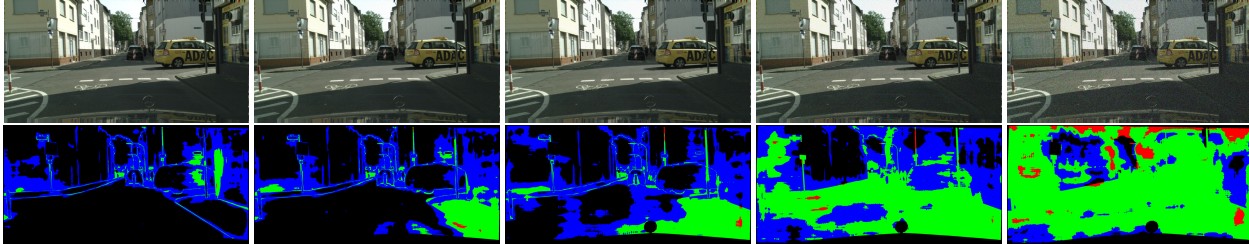

Figure 9: An example of how DRN misclassifies inputs when noise is added and how well entropy can capture it. From left to right, we have the original image followed by increasing amounts of noise added to it. The bottom row shows misclassified pixels with high entropy in green, misclassified pixels with low entropy in red and correctly classified pixels with high entropy in blue.

Table 7: AUROC for increasing noise levels on the Cityscapes dataset for DRN and OneFormer models.

| Model | Base | Noise 5 | Noise 10 | Noise 25 | Noise 50 |
|---|---|---|---|---|---|
| DRN D-22 | 0.9205 | 0.8554 | 0.7866 | 0.7156 | 0.6427 |
| OneFormer ConvNeXt-L | 0.7089 | 0.6975 | 0.7008 | 0.6810 | 0.6375 |

and generally find that our AUROC scores using Entropy as the uncertainty metric do improve on the calibrated models. This experiment indicates that considering the underlying causes for uncertainty, such as calibration, and fixing these can help further identify the misclassified pixels. We show the results of this experiment in Table 5. We use temperatures of 0.2 and 2 for OneFormer and SegFormer models respectively. We found that DRN models were already well calibrated and further temperature tuning was unnecessary.

## 5.5 Domain Shift

In order to check whether these uncertainty metrics can capture misclassification in the case of input domain shift, we consider testing the models trained with the Cityscapes dataset on the Dark Zurich dataset. Dark Zurich is similar to Cityscapes in the sense that both include dashcam images from German cities. The major difference is that in Dark Zurich the images are taken during nighttime while in Cityscapes the images are taken during the day. This represents a natural domain shift one may encounter in the real world. Table 7 shows the results of using Cityscape- trained DRN and OneFormer models on the Dark Zurich dataset using Entropy as the uncertainty metric. The models perform poorly on this dataset, with mIoU of 0.0710 for DRN and 0.3994 for OneFormer. Even though the models perform poorly, we note that we can still use Entropy to get good Precision and Recall rates using the largest difference method for dynamic thresholding. We also note that using multiple scales generally performs better than other scenarios.

## 5.6 Noisy Inputs

We also consider introducing noise in the input images and observing how it affects our results. Effectively, this represents another form of domain shift. We introduce varying levels of noise in the input and check the performance of the networks on them. The noise introduced is Gaussian in nature with standard deviations of $\{5, 10, 25, 50\}$. The noise is introduced into the image before any preprocessing to ensure we do not subvert any mechanisms used by the networks to protect against such cases. Table 7 shows the AUROC values in this scenario on DRN D-22 and OneFormer ConvNeXt-L models. We also provide a visual example of the same in Figure 9. As can be seen with increasing noise levels, the outputs become highly inaccurate, but we can still capture most of these using Entropy as the uncertainty metric.

# 6    Discussion

Through this work, we attempt to thoroughly analyse the failure modes of semantic segmentation networks and aim to study the ability of commonly available test-time uncertainty metrics to *predict* misclassifications. This is a relatively less well-understood and explored topic in semantic segmentation space. Our work aims to provide benchmarks and poses this novel task in the semantic segmentation space. We now discuss the key insights we have from our several experiments.

In Section 3, we observe that the key points of failure in semantic segmentation networks are at the edges of objects in the image. While this is intuitive, through examples, we show that this is indeed the case. We also note that in several cases, the boundaries are actually quite ambiguous, and it is not really possible to assign a pixel to a single category. However, existing metrics of mIoU and mAcc can change quite a bit depending on how correctly these boundaries are classified. This opens up discussions of perhaps alternative evaluation metrics for segmentation which take into account such inherent fuzziness of object boundaries.

In the same section, we also observe that while a simple uncertainty metric like entropy correlates well with correctly classified and misclassified pixels, the actual magnitudes of said entropy can vary wildly from one network to another, as evidenced by OneFormer with its naturally high entropy output. This makes us rethink how to properly threshold these uncertainty values when they fluctuate so much from image to image and network to network.

In our primary analysis in Section 5.3, our ROC curves show that for DRN and SegFormer models, we can achieve approximately 90% TPR at 20% FPR on Cityscapes. For ADE20K, we get about 80% TPR at 30% FPR, which, while worse than Cityscapes, is overall not particularly bad considering it has 150 classes. In the same section, we also note that while both the first (largest difference) and third (K-Means) methods of dynamic thresholding depend on the uncertainties of both classes being somewhat clustered, K-Means performs much better.

We do note, however, in Section 5.5 and Section 5.6 that the largest difference thresholding can work really well when there are actually a lot of misclassifications, and we recommend it for such situations. We also recommend calibrating the models whenever possible since calibrated models always provide better AUROC, Precision and Recall, as shown in Section 5.4.

The exception to all these is OneFormer, where the uncertainties are unreasonably high and thus less properly clustered. The unnaturally high uncertainty values of OneFormer means less difference between uncertainties of correctly classified and misclassified pixels. Thus, for any method relying on the difference of these values to predict misclassification is going to perform poorly. We thus recommend not to use uncertainty-based metrics for misclassification prediction if the uncertainties are, in general, high. The question of why OneFomer has such high uncertainties may have to do with the fact that out of the networks tested, it is the only one that was trained to do more than just semantic segmentation.

# 7    Conclusion

In this paper, we present a new form of analysis for semantic segmentation tasks focused on trusting the output of such networks during test time. Our work is primarily concerned with evaluating existing uncertainty metrics and seeing how well they correlate with and are able to capture misclassifications. Our analysis suggests that it is indeed possible to obtain valid evaluation measures that can be used to predict the test time performance of existing segmentation networks. We have shown that even simple measures such as Probability Margins and Entropy can be used as a proxy for capturing misclassified pixels with high recall rates. Because of their simplicity and generality, such tests can be used with nearly all existing architectures without any computational overhead and can help us know when the outputs are to be trusted. We believe that our work will provide new insights into the nature of semantic segmentation and help drive further research in evaluating the trustworthiness of these networks.

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

# A  Cityscapes, ADE20K and COCO

First, we provide all results related to Cityscapes, ADE20K and COCO datasets. We provide results for Dark Zurich in a separate section, as experiments on it involve input domain shift. Table 8 collates all the performance metrics for the Base setting of all networks. We compute Mean Intersection Over Union (mIoU), Expected Calibration Error (ECE) and Brier Score (BS). For all of the metrics, we show both micro-averaged ($\mu$ Avg.) as well as macro-averaged (M Avg.) results.

As expected, DRN performs the worst among all the models, while OneFormer and SegFormer perform much better. We also note that DRN and SegFormer are relatively well-calibrated, but OneFormer is not. Results on Cityscapes are much better than on ADE20K since the latter is a much more challenging dataset with more variety and a much larger number of classes.

Table 8: Micro and macro-averaged ($\mu$ and M, respectively) Mean Intersection Over Union (mIoU), Expected Calibration Error (ECE) and Brier Score (BS) on Cityscapes, ADE20K and COCO datasets.

| Model | mIoU ↑ | | ECE ↓ | | BS ↓ | |
|---|---|---|---|---|---|---|
| | $\mu$ Avg. | M Avg. | $\mu$ Avg. | M Avg. | $\mu$ Avg. | M Avg. |
| Cityscapes | | | | | | |
| DRN D-22 | 0.6790 | 0.5479 | 0.0380 | 0.0177 | 0.0808 | 0.0824 |
| DRN D-105 | 0.7551 | 0.6330 | **0.0337** | 0.0171 | 0.0617 | 0.0631 |
| OneFormer ConvNeXt-L | 0.8287 | 0.6733 | 0.7887 | 0.8383 | 0.8012 | 0.8011 |
| OneFormer Swin-L | **0.8288** | 0.6682 | 0.7973 | 0.8400 | 0.8034 | 0.8034 |
| SegFormer B5 | 0.8239 | **0.6875** | 0.0631 | **0.0160** | **0.0516** | **0.0524** |
| ADE20K | | | | | | |
| OneFormer ConvNeXt-L | 0.5684 | 0.4854 | 0.8753 | 0.8195 | 0.9690 | 0.9686 |
| OneFormer Swin-L | **0.5698** | **0.4921** | 0.8948 | 0.8261 | 0.9689 | 0.9687 |
| SegFormer B5 | 0.5100 | 0.4434 | **0.1051** | **0.1068** | **0.2646** | **0.2845** |
| COCO | | | | | | |
| OneFormer Swin-L | 0.6722 | 0.5043 | | | | |

Table 9, Table 10 and Table 11 show the Area Under Receiver Operating Characteristic (AUROC) as well as the Precision, Recall and fraction of selected pixels on Cityscapes, ADE20K and COCO datasets, respectively.

We observe that in most cases, Scale seems to provide the best AUROC and Recall, while Noise generally provides good Precision. However, in most cases, Base comes quite close to these values and has the advantage of being computationally cheaper. We also note that simpler metrics such as Variation Ratio, Probability Margin and Entropy are often better than or very close to more involved metrics such as Variance and BALD. The Recall values often reach higher than 90% for Cityscapes and 70% for ADE20K, although often at the cost of selecting too many pixels in the image.

Table 9: Micro and macro-averaged ($\mu$ and M, respectively) Area Under Receiver Operating Characteristic (AUROC) and Precision, Recall and the fraction of pixels selected for largest difference thresholding on the Cityscapes dataset.

| Scenario | Uncertainty Metric | AUROC ↑ | | Largest Difference Thresholding | | | | | |
|---|---|---|---|---|---|---|---|---|---|
| | | | | Precision ↑ | | Recall ↑ | | Pixel % | |
| | | $\mu$ Avg. | M Avg. | $\mu$ Avg. | M Avg. | $\mu$ Avg. | M Avg. | $\mu$ Avg. | M Avg. |
| | DRN D-22 | | | | | | | | |
| Base | Var. Ratio | 0.9118 | 0.9277 | 21.10 | 20.18 | 92.13 | 94.71 | 23.29 | 23.51 |
| | Prob. Margin | 0.9152 | 0.9302 | 19.71 | 18.84 | 93.31 | 95.62 | 25.26 | 25.48 |
| | Entropy | 0.9205 | 0.9337 | 17.52 | 16.74 | 95.04 | 96.89 | 28.94 | 29.21 |

| | | | | | | | | |
|---|---|---|---|---|---|---|---|---|
| Noise | Var. Ratio | 0.9119 | 0.9278 | 21.10 | 20.18 | 92.15 | 94.73 | 23.30 | 23.52 |
| | Prob. Margin | 0.9152 | 0.9303 | 19.71 | 18.85 | 93.32 | 95.63 | 25.26 | 25.49 |
| | Entropy | 0.9205 | 0.9337 | 17.52 | 16.74 | 95.05 | 96.90 | 28.95 | 29.22 |
| | Avg. Var. | 0.7609 | 0.7722 | **37.97** | **36.82** | 57.26 | 59.58 | 08.04 | 08.13 |
| | Max. Var. | 0.7614 | 0.7724 | 37.71 | 36.55 | 57.42 | 59.71 | 08.12 | 08.21 |
| | BALD | 0.8196 | 0.8276 | 29.28 | 28.36 | 72.32 | 73.90 | 13.18 | 13.30 |
| Scale | Var. Ratio | 0.9410 | **0.9466** | 15.90 | 15.22 | 97.74 | 98.46 | 32.80 | 33.08 |
| | Prob. Margin | **0.9414** | 0.9465 | 14.80 | 14.19 | 98.14 | 98.76 | 35.36 | 35.66 |
| | Entropy | 0.9398 | 0.9444 | 13.04 | 12.52 | **98.74** | **99.19** | 40.38 | 40.72 |
| | Avg. Var. | 0.8973 | 0.8998 | 23.50 | 22.51 | 91.38 | 92.38 | 20.74 | 20.95 |
| | Max. Var. | 0.8976 | 0.8995 | 23.18 | 22.20 | 91.69 | 92.65 | 21.11 | 21.31 |
| | BALD | 0.9066 | 0.9075 | 16.49 | 15.82 | 96.37 | 97.10 | 31.17 | 31.44 |

### DRN D-105

| | | | | | | | | |
|---|---|---|---|---|---|---|---|---|
| Base | Var. Ratio | 0.8939 | 0.9206 | 23.04 | 22.19 | 85.84 | 90.61 | 14.76 | 14.87 |
| | Prob. Margin | 0.8995 | 0.9247 | 21.77 | 20.94 | 87.36 | 91.77 | 15.90 | 16.02 |
| | Entropy | 0.9094 | 0.9322 | 19.75 | 18.95 | 89.88 | 93.64 | 18.03 | 18.17 |
| Noise | Var. Ratio | 0.8949 | 0.9209 | 23.09 | 22.22 | 86.03 | 90.70 | 14.76 | 14.87 |
| | Prob. Margin | 0.9004 | 0.9250 | 21.82 | 20.98 | 87.54 | 91.84 | 15.90 | 16.01 |
| | Entropy | 0.9103 | 0.9325 | 19.79 | 18.98 | 90.05 | 93.70 | 18.03 | 18.16 |
| | Avg. Var. | 0.7471 | 0.7666 | **39.53** | **38.65** | 52.68 | 56.64 | 05.28 | 05.34 |
| | Max. Var. | 0.7472 | 0.7664 | 39.46 | 38.56 | 52.71 | 56.61 | 05.29 | 05.36 |
| | BALD | 0.8077 | 0.8249 | 32.48 | 31.63 | 66.59 | 70.05 | 08.12 | 08.21 |
| Scale | Var. Ratio | 0.9341 | 0.9474 | 19.48 | 18.60 | 93.73 | 95.92 | 19.07 | 19.21 |
| | Prob. Margin | 0.9366 | 0.9490 | 18.33 | 17.51 | 94.52 | 96.48 | 20.43 | 20.57 |
| | Entropy | **0.9400** | **0.9509** | 16.45 | 15.71 | **95.79** | **97.33** | 23.07 | 23.23 |
| | Avg. Var. | 0.8893 | 0.9007 | 27.81 | 26.58 | 84.51 | 87.00 | 12.04 | 12.15 |
| | Max. Var. | 0.8900 | 0.9009 | 27.59 | 26.37 | 84.74 | 87.17 | 12.17 | 12.28 |
| | BALD | 0.9111 | 0.9197 | 20.60 | 19.70 | 91.20 | 93.16 | 17.54 | 17.67 |

### OneFormer ConvNeXt-L

| | | | | | | | | |
|---|---|---|---|---|---|---|---|---|
| Base | Var. Ratio | 0.7265 | 0.8822 | 15.02 | 16.52 | 84.68 | 87.75 | 18.03 | 18.17 |
| | Prob. Margin | 0.7893 | 0.9073 | 14.37 | 15.63 | 89.19 | 91.32 | 19.84 | 19.98 |
| | Entropy | 0.7089 | 0.8569 | 14.13 | 16.23 | 81.30 | 84.88 | 18.40 | 18.65 |
| Noise | Var.Ratio | 0.7334 | 0.8851 | 15.04 | 16.66 | 84.72 | 87.85 | 18.00 | 18.13 |
| | Prob. Margin | 0.7927 | 0.9087 | 14.15 | 15.72 | 89.07 | 91.37 | 20.12 | 20.25 |
| | Entropy | 0.7153 | 0.8612 | 14.25 | 16.54 | 80.60 | 85.05 | 18.08 | 18.21 |
| | Avg. Var. | 0.6458 | 0.6474 | 18.71 | 28.78 | 34.52 | 35.06 | 05.90 | 05.99 |
| | Max. Var. | 0.6323 | 0.6324 | 17.11 | 27.49 | 31.93 | 32.20 | 05.97 | 06.05 |
| | BALD | 0.6727 | 0.6742 | 20.21 | **29.16** | 40.01 | 40.55 | 06.33 | 06.42 |
| Scale | Var. Ratio | 0.7647 | 0.8972 | 12.96 | 14.53 | 88.64 | 90.63 | 21.87 | 22.06 |
| | Prob. Margin | **0.8305** | **0.9215** | 12.58 | 13.53 | **92.36** | **93.71** | 23.46 | 23.68 |
| | Entropy | 0.7350 | 0.8736 | 12.11 | 14.36 | 82.76 | 87.42 | 21.84 | 21.90 |
| | Avg. Var. | 0.7835 | 0.7880 | 16.78 | 21.70 | 65.40 | 67.80 | 12.46 | 12.54 |
| | Max. Var. | 0.7637 | 0.7677 | 15.75 | 20.82 | 62.04 | 64.34 | 12.59 | 12.67 |
| | BALD | 0.8173 | 0.8213 | 17.66 | 21.76 | 71.71 | 74.12 | 12.98 | 13.05 |
| Drop | Var. Ratio | 0.7471 | 0.8936 | 14.89 | 15.82 | 85.54 | 89.39 | 18.36 | 18.44 |
| | Prob. Margin | 0.8059 | 0.9166 | 13.51 | 14.47 | 91.12 | 93.09 | 21.56 | 21.71 |
| | Entropy | 0.7278 | 0.8712 | 13.76 | 15.12 | 82.96 | 87.21 | 19.28 | 19.40 |
| | Avg. Var. | 0.7219 | 0.7264 | 21.08 | 27.12 | 49.93 | 52.42 | 07.57 | 07.68 |
| | Max. Var. | 0.6994 | 0.7023 | 19.55 | 25.98 | 45.49 | 47.68 | 07.44 | 07.54 |
| | BALD | 0.7624 | 0.7667 | **21.71** | 27.02 | 58.30 | 60.90 | 08.58 | 08.69 |

### OneFormer Swin-L

| | | | | | | | | |
|---|---|---|---|---|---|---|---|---|
| Base | Var. Ratio | 0.7379 | 0.8909 | 13.68 | 15.60 | 86.62 | 89.80 | 20.24 | 20.37 |
| | Prob. Margin | 0.7937 | 0.9127 | 13.30 | 14.59 | 89.82 | 92.61 | 21.57 | 21.72 |

| Scenario | Metric | μ Avg. | M Avg. | μ Avg. | M Avg. | μ Avg. | M Avg. | μ Avg. | M Avg. |
|---|---|---|---|---|---|---|---|---|---|
| Noise | Entropy | 0.7238 | 0.8677 | 13.57 | 15.52 | 82.85 | 86.61 | 19.51 | 19.73 |
| | Var.Ratio | 0.7432 | 0.8927 | 14.34 | 15.84 | 86.49 | 89.67 | 19.28 | 19.48 |
| | Prob. Margin | 0.7976 | 0.9139 | 13.76 | 14.87 | 90.09 | 92.60 | 20.93 | 21.10 |
| | Entropy | 0.7288 | 0.8709 | 13.95 | 15.60 | 83.60 | 87.08 | 19.15 | 19.39 |
| | Avg. Var. | 0.6603 | 0.6611 | 23.56 | 29.87 | 36.01 | 36.33 | 04.88 | 05.00 |
| | Max. Var. | 0.6467 | 0.6463 | 22.08 | 28.63 | 33.36 | 33.46 | 04.83 | 04.95 |
| | BALD | 0.6866 | 0.6874 | **25.56** | **30.50** | 41.36 | 41.61 | 05.17 | 05.24 |
| Scale | Var. Ratio | 0.7816 | 0.9109 | 12.76 | 13.84 | 89.92 | 92.58 | 22.52 | 22.69 |
| | Prob. Margin | **0.8415** | **0.9304** | 11.87 | 12.90 | **93.29** | **95.15** | 25.12 | 25.40 |
| | Entropy | 0.7557 | 0.8951 | 12.30 | 13.65 | 87.69 | 90.83 | 22.78 | 22.95 |
| | Avg. Var. | 0.8093 | 0.8159 | 19.67 | 22.86 | 70.28 | 71.57 | 11.42 | 11.59 |
| | Max. Var. | 0.7929 | 0.7985 | 18.86 | 22.18 | 67.53 | 68.59 | 11.44 | 11.61 |
| | BALD | 0.8352 | 0.8421 | 20.03 | 22.73 | 75.31 | 76.74 | 12.02 | 12.20 |
| Drop | Var. Ratio | 0.7543 | 0.8972 | 13.39 | 14.67 | 87.99 | 90.76 | 20.99 | 21.20 |
| | Prob. Margin | 0.8078 | 0.9179 | 12.95 | 14.15 | 90.91 | 93.25 | 22.43 | 22.68 |
| | Entropy | 0.7396 | 0.8779 | 12.60 | 14.31 | 85.42 | 88.66 | 21.65 | 21.85 |
| | Avg. Var. | 0.7211 | 0.7397 | 23.02 | 28.08 | 49.79 | 53.67 | 06.91 | 07.04 |
| | Max. Var. | 0.6994 | 0.7164 | 21.31 | 27.09 | 45.57 | 49.20 | 06.83 | 07.01 |
| | BALD | 0.7593 | 0.7765 | 24.27 | 28.27 | 57.69 | 61.29 | 07.60 | 07.69 |
| | | | | | SegFormer B5 | | | | |
| Base | Var. Ratio | 0.8830 | 0.9087 | 24.60 | 24.07 | 82.10 | 86.94 | 10.90 | 10.95 |
| | Prob. Margin | 0.8892 | 0.9142 | 23.40 | 22.87 | 83.65 | 88.29 | 11.68 | 11.73 |
| | Entropy | 0.8997 | 0.9234 | 21.43 | 20.89 | 86.20 | 90.47 | 13.14 | 13.20 |
| Noise | Var.Ratio | 0.8833 | 0.9089 | 24.53 | 24.00 | 82.22 | 87.02 | 10.95 | 11.00 |
| | Prob. Margin | 0.8896 | 0.9143 | 23.33 | 22.81 | 83.77 | 88.37 | 11.73 | 11.78 |
| | Entropy | 0.9002 | 0.9236 | 21.37 | 20.84 | 86.34 | 90.55 | 13.20 | 13.26 |
| | Avg. Var. | 0.6972 | 0.7147 | **40.19** | **40.11** | 41.61 | 45.11 | 03.38 | 03.40 |
| | Max. Var. | 0.6965 | 0.7136 | 40.18 | 40.07 | 41.46 | 44.90 | 03.37 | 03.39 |
| | BALD | 0.7520 | 0.7697 | 34.08 | 34.03 | 53.80 | 57.30 | 05.16 | 05.19 |
| Scale | Var. Ratio | 0.9212 | 0.9391 | 20.33 | 19.77 | 90.13 | 93.35 | 14.48 | 14.56 |
| | Prob. Margin | 0.9245 | 0.9418 | 19.21 | 18.67 | 91.02 | 94.10 | 15.48 | 15.56 |
| | Entropy | **0.9290** | **0.9459** | 17.37 | 16.85 | **92.46** | **95.29** | 17.39 | 17.48 |
| | Avg. Var. | 0.8678 | 0.8826 | 28.23 | 27.55 | 78.95 | 81.98 | 09.13 | 09.19 |
| | Max. Var. | 0.8687 | 0.8832 | 28.04 | 27.36 | 79.20 | 82.18 | 09.23 | 09.28 |
| | BALD | 0.8947 | 0.9091 | 21.36 | 20.81 | 86.42 | 89.33 | 13.22 | 13.28 |
| Drop | Var. Ratio | 0.8842 | 0.9097 | 24.44 | 23.91 | 82.38 | 87.17 | 11.01 | 11.06 |
| | Prob. Margin | 0.8904 | 0.9151 | 23.24 | 22.72 | 83.92 | 88.50 | 11.80 | 11.85 |
| | Entropy | 0.9008 | 0.9242 | 21.28 | 20.74 | 86.46 | 90.66 | 13.27 | 13.33 |
| | Avg. Var. | 0.7677 | 0.7931 | 37.02 | 36.66 | 56.64 | 61.71 | 05.00 | 05.02 |
| | Max. Var. | 0.7660 | 0.7915 | 37.06 | 36.69 | 56.31 | 61.38 | 04.96 | 04.98 |
| | BALD | 0.8255 | 0.8519 | 28.97 | 28.73 | 70.12 | 75.25 | 07.91 | 07.93 |

Table 10: Micro and macro-averaged ($\mu$ and M, respectively) Area Under Receiver Operating Characteristic (AUROC) and Precision, Recall and the fraction of pixels selected for largest difference thresholding on the ADE20K dataset.

| Scenario | Uncertainty Metric | AUROC ↑ | | Largest Difference Thresholding | | | | | |
| | | | | Precision ↑ | | Recall ↑ | | Pixel % | |
| | | μ Avg. | M Avg. | μ Avg. | M Avg. | μ Avg. | M Avg. | μ Avg. | M Avg. |
|---|---|---|---|---|---|---|---|---|---|
| | | | | OneFormer ConvNeXt-L | | | | | |
| Base | Var. Ratio | 0.6067 | 0.6941 | 25.45 | 28.92 | 50.31 | 63.73 | 29.11 | 30.34 |

| | | | | | | | | |
|---|---|---|---|---|---|---|---|---|
| | Prob. Margin | 0.7048 | 0.7750 | 25.15 | 29.12 | 63.92 | 75.96 | 37.43 | 38.25 |
| | Entropy | 0.5588 | 0.6164 | 23.07 | 27.39 | 41.38 | 54.30 | 26.42 | 27.64 |
| Noise | Var. Ratio | 0.6052 | 0.6927 | 25.73 | 28.89 | 50.62 | 63.82 | 28.97 | 30.43 |
| | Prob. Margin | 0.7039 | 0.7727 | 25.48 | 29.24 | 63.66 | 75.54 | 36.79 | 37.66 |
| | Entropy | 0.5576 | 0.6151 | 23.34 | 27.23 | 41.43 | 54.34 | 26.13 | 27.57 |
| | Avg. Var. | 0.6328 | 0.6235 | **30.67** | **35.27** | 41.61 | 45.27 | 19.98 | 22.40 |
| | Max. Var. | 0.6184 | 0.6084 | 29.52 | 34.66 | 38.94 | 42.18 | 19.42 | 21.76 |
| | BALD | 0.6218 | 0.6672 | 26.34 | 29.24 | 54.24 | 58.38 | 30.32 | 33.04 |
| Scale | Var. Ratio | 0.6243 | 0.7078 | 24.23 | 28.00 | 52.29 | 65.56 | 31.78 | 32.76 |
| | Prob. Margin | 0.7285 | **0.7883** | 24.87 | 28.95 | 65.42 | 76.88 | 38.74 | 39.19 |
| | Entropy | 0.5693 | 0.6304 | 22.37 | 26.39 | 42.37 | 55.78 | 27.89 | 29.02 |
| | Avg. Var. | 0.6880 | 0.6728 | 27.69 | 30.54 | 61.78 | 66.51 | 32.85 | 35.02 |
| | Max. Var. | 0.6635 | 0.6489 | 26.92 | 29.84 | 58.47 | 62.64 | 31.98 | 34.25 |
| | BALD | **0.7473** | 0.7279 | 28.66 | 30.16 | **73.48** | **77.93** | 37.76 | 39.69 |
| Drop | Var. Ratio | 0.6136 | 0.6956 | 25.19 | 28.55 | 50.22 | 64.06 | 29.36 | 30.82 |
| | Prob. Margin | 0.7151 | 0.7778 | 25.10 | 29.06 | 63.51 | 76.03 | 37.26 | 38.10 |
| | Entropy | 0.5618 | 0.6169 | 22.72 | 26.59 | 40.91 | 54.33 | 26.51 | 27.99 |
| | Avg. Var. | 0.6732 | 0.6636 | 29.53 | 33.02 | 54.03 | 59.30 | 26.95 | 29.28 |
| | Max. Var. | 0.6517 | 0.6415 | 28.65 | 32.46 | 50.24 | 54.95 | 25.82 | 28.23 |
| | BALD | 0.7359 | 0.7258 | 29.66 | 31.50 | 68.32 | 74.11 | 33.92 | 36.23 |
| OneFormer Swin-L | | | | | | | | | |
| Base | Var. Ratio | 0.6115 | 0.6977 | 24.79 | 28.04 | 52.66 | 64.95 | 30.16 | 31.21 |
| | Prob. Margin | 0.7066 | 0.7770 | 25.18 | 28.13 | 66.49 | 76.84 | 37.49 | 38.22 |
| | Entropy | 0.5661 | 0.6222 | 22.96 | 26.26 | 43.63 | 55.65 | 26.97 | 28.53 |
| Noise | Var. Ratio | 0.6110 | 0.6958 | 24.86 | 27.82 | 53.28 | 65.05 | 30.42 | 31.46 |
| | Prob. Margin | 0.7066 | 0.7748 | 24.83 | 27.98 | 65.73 | 76.59 | 37.58 | 38.38 |
| | Entropy | 0.5652 | 0.6208 | 22.82 | 26.20 | 43.62 | 55.28 | 27.13 | 28.37 |
| | Avg. Var. | 0.6331 | 0.6281 | 28.40 | **34.11** | 43.92 | 47.86 | 21.95 | 24.05 |
| | Max. Var. | 0.6180 | 0.6114 | 27.42 | 33.52 | 40.58 | 44.08 | 21.01 | 23.09 |
| | BALD | 0.6510 | 0.6806 | 26.38 | 29.88 | 57.68 | 62.16 | 31.04 | 33.33 |
| Scale | Var. Ratio | 0.6296 | 0.7142 | 23.84 | 27.27 | 55.82 | 67.39 | 33.23 | 33.60 |
| | Prob. Margin | 0.7322 | **0.7933** | 24.05 | 27.69 | 68.38 | 78.83 | 40.37 | 40.47 |
| | Entropy | 0.5742 | 0.6358 | 21.99 | 25.40 | 45.23 | 57.34 | 29.20 | 30.27 |
| | Avg. Var. | 0.6994 | 0.6891 | 27.97 | 30.37 | 64.48 | 68.74 | 32.72 | 34.31 |
| | Max. Var. | 0.6735 | 0.6643 | 26.78 | 29.50 | 60.42 | 64.77 | 32.02 | 33.75 |
| | BALD | **0.7563** | 0.7427 | 28.37 | 29.68 | **74.23** | **79.18** | 37.15 | 38.73 |
| Drop | Var. Ratio | 0.6173 | 0.7005 | 24.31 | 27.57 | 53.69 | 65.76 | 31.35 | 32.33 |
| | Prob. Margin | 0.7156 | 0.7804 | 24.41 | 28.04 | 65.93 | 76.85 | 38.34 | 38.83 |
| | Entropy | 0.5693 | 0.6241 | 22.34 | 25.76 | 43.40 | 55.83 | 27.57 | 28.91 |
| | Avg. Var. | 0.6665 | 0.6565 | 27.86 | 31.72 | 55.14 | 59.57 | 28.10 | 30.11 |
| | Max. Var. | 0.6459 | 0.6333 | 26.98 | 31.18 | 51.19 | 54.98 | 26.93 | 28.93 |
| | BALD | 0.7302 | 0.7182 | **28.76** | 30.76 | 69.12 | 73.75 | 34.11 | 35.92 |
| SegFormer B5 | | | | | | | | | |
| Base | Var. Ratio | 0.8000 | 0.8388 | 35.54 | 33.27 | 80.01 | 87.38 | 36.81 | 38.62 |
| | Prob. Margin | 0.8024 | 0.8397 | 34.56 | 32.45 | 81.51 | 88.36 | 38.56 | 40.28 |
| | Entropy | 0.8113 | 0.8484 | 32.20 | 30.34 | 84.63 | 90.97 | 42.96 | 44.99 |
| Noise | Var. Ratio | 0.8356 | 0.8628 | 31.86 | 30.41 | 88.13 | 93.03 | 45.22 | 46.72 |
| | Prob. Margin | 0.8353 | 0.8615 | 30.93 | 29.67 | 88.88 | 93.51 | 46.97 | 48.33 |
| | Entropy | **0.8384** | **0.8661** | 28.80 | 27.87 | **90.24** | **94.82** | 51.21 | 52.85 |
| | Avg. Var. | 0.7929 | 0.8096 | 39.47 | 37.55 | 77.79 | 83.11 | 32.22 | 33.33 |
| | Max. Var. | 0.7929 | 0.8089 | 39.28 | 37.40 | 78.06 | 83.31 | 32.48 | 33.60 |
| | BALD | 0.8141 | 0.8266 | 32.97 | 31.46 | 86.96 | 91.18 | 43.11 | 44.50 |
| Scale | Var. Ratio | 0.8003 | 0.8388 | 35.55 | 33.26 | 80.02 | 87.38 | 36.79 | 38.59 |

| | | | | | | | | |
|---|---|---|---|---|---|---|---|---|
| | Prob. Margin | 0.8028 | 0.8397 | 34.57 | 32.44 | 81.51 | 88.35 | 38.54 | 40.24 |
| | Entropy | 0.8115 | 0.8483 | 32.12 | 30.34 | 84.27 | 90.91 | 42.89 | 44.93 |
| | Avg. Var. | 0.7123 | 0.7490 | **48.46** | **45.09** | 52.74 | 61.89 | 17.79 | 19.46 |
| | Max. Var. | 0.7116 | 0.7483 | 48.38 | 45.03 | 52.65 | 61.80 | 17.79 | 19.45 |
| | BALD | 0.7627 | 0.7952 | 41.45 | 38.38 | 68.51 | 76.96 | 27.02 | 29.59 |
| Drop | Var. Ratio | 0.8016 | 0.8398 | 35.32 | 33.08 | 80.47 | 87.70 | 37.24 | 39.05 |
| | Prob. Margin | 0.8039 | 0.8407 | 34.34 | 32.26 | 81.92 | 88.65 | 38.99 | 40.71 |
| | Entropy | 0.8125 | 0.8491 | 31.92 | 30.16 | 84.70 | 91.18 | 43.37 | 45.43 |
| | Avg. Var. | 0.7521 | 0.7955 | 44.41 | 41.56 | 63.76 | 73.73 | 23.47 | 24.96 |
| | Max. Var. | 0.7520 | 0.7955 | 44.35 | 41.52 | 63.85 | 73.85 | 23.54 | 25.04 |
| | BALD | 0.7914 | 0.8299 | 36.50 | 34.17 | 78.07 | 86.38 | 34.96 | 37.22 |

Table 11: Micro and macro-averaged ($\mu$ and M, respectively) Area Under Receiver Operating Characteristic (AUROC) and Precision, Recall and the fraction of pixels selected for largest difference thresholding on the COCO dataset.

| Scenario | Uncertainty Metric | AUROC ↑ | | Largest Difference Thresholding | | | | | |
|---|---|---|---|---|---|---|---|---|---|
| | | | | Precision ↑ | | Recall ↑ | | Pixel % | |
| | | $\mu$ Avg. | M Avg. | $\mu$ Avg. | M Avg. | $\mu$ Avg. | M Avg. | $\mu$ Avg. | M Avg. |
| | | | | OneFormer Swin-L | | | | | |
| Base | Var. Ratio | 0.6420 | 0.7200 | 25.23 | 26.58 | 53.93 | 69.86 | 36.49 | 36.74 |
| | Prob. Margin | 0.7515 | 0.7998 | 26.30 | 28.23 | 65.27 | 79.62 | 42.38 | 42.54 |
| | Entropy | 0.5812 | 0.6143 | 21.43 | 23.05 | 41.02 | 57.68 | 32.69 | 32.90 |
| Scale | Var. Ratio | 0.6587 | 0.7324 | 24.84 | 26.36 | 56.05 | 71.52 | 38.52 | 38.68 |
| | Prob. Margin | **0.7696** | **0.8151** | 26.16 | 28.13 | **67.72** | **81.37** | 44.20 | 44.36 |
| | Entropy | 0.5904 | 0.6269 | 21.09 | 22.49 | 42.63 | 59.40 | 34.51 | 34.74 |
| | Avg. Var. | 0.6923 | 0.7011 | 31.77 | **32.39** | 60.76 | 68.36 | 32.65 | 33.01 |
| | Max. Var. | 0.6716 | 0.6807 | 30.96 | 31.81 | 57.62 | 64.90 | 31.88 | 32.22 |
| | BALD | 0.7293 | 0.7369 | **32.12** | 32.06 | 66.35 | 74.73 | 35.27 | 35.68 |

We also show the Area Under Precision Recall Curve (AUPRC) and False Positive Rate at 95% True Positive Rate (FPR@95%TPR) for selected models on the Cityscapes dataset in Table 12. For AUPRC we show both AUPRC Error and Success. In AUPRC-Error the misclassified pixels are treated as the positive class whereas in AUPRC-Success correctly classified pixels are treated as the positive class. We observe that much like in the case of AUROC, Scale setting peforms the best.

Table 12: Micro and macro-averaged ($\mu$ and M, respectively) Area Under Precision Recall Curve (AUPRC) and False Positive Rate at 95% True Positive Rate (FPR@95%TPR) on the Cityscapes dataset.

| Scenario | Uncertainty Metric | AUPRC-Error ↑ | | AUPRC-Success ↑ | | FPR@95%TPR ↓ | |
|---|---|---|---|---|---|---|---|
| | | $\mu$ Avg. | M Avg. | $\mu$ Avg. | M Avg. | $\mu$ Avg. | M Avg. |
| | | | | DRN D-22 | | | |
| Base | Var. Ratio | 0.4383 | 0.4483 | 0.9950 | 0.9933 | 1.0000 | 0.4042 |
| | Prob. Margin | 0.4208 | 0.4308 | 0.9951 | 0.9934 | 1.0000 | 0.3716 |
| | Entropy | 0.4249 | 0.4316 | 0.9953 | 0.9936 | 0.2515 | 0.3104 |
| Noise | Var. Ratio | 0.4375 | 0.4483 | 0.9951 | 0.9933 | 1.0000 | 0.4037 |
| | Prob. Margin | 0.4198 | 0.4307 | 0.9951 | 0.9934 | 1.0000 | 0.3702 |
| | Entropy | 0.4246 | 0.4317 | 0.9953 | 0.9936 | 0.2515 | 0.3089 |
| | Avg. Var. | 0.3364 | 0.3320 | 0.9926 | 0.9912 | 1.0000 | 1.0000 |

| | | | | | | | |
|---|---|---|---|---|---|---|---|
| | Max. Var. | 0.3347 | 0.3300 | 0.9925 | 0.9912 | 1.0000 | 1.0000 |
| | BALD | 0.3231 | 0.3162 | 0.9892 | 0.9883 | 1.0000 | 0.9982 |
| Scale | Var. Ratio | **0.4901** | **0.4850** | **0.9964** | **0.9954** | 0.2134 | 0.2177 |
| | Prob. Margin | 0.4665 | 0.4624 | **0.9964** | **0.9954** | 0.2040 | 0.2070 |
| | Entropy | 0.4469 | 0.4402 | 0.9963 | 0.9952 | **0.1988** | **0.1938** |
| | Avg. Var. | 0.3167 | 0.3048 | 0.9946 | 0.9933 | 1.0000 | 0.6696 |
| | Max. Var. | 0.3153 | 0.2983 | 0.9946 | 0.9932 | 1.0000 | 0.6395 |
| | BALD | 0.3052 | 0.2940 | 0.9942 | 0.9931 | 0.2742 | 0.3224 |
| | OneFormer ConvNeXt-L | | | | | | |
| Base | Var. Ratio | 0.1310 | 0.3090 | 0.9853 | 0.9633 | 0.8001 | 0.6999 |
| | Prob. Margin | 0.2300 | 0.3562 | 0.9885 | 0.9722 | 0.7587 | 0.5619 |
| | Entropy | 0.1193 | 0.2832 | 0.9847 | 0.9548 | 0.8034 | 0.7569 |
| Noise | Var.Ratio | 0.1356 | 0.3134 | 0.9849 | 0.9628 | 0.7973 | 0.6880 |
| | Prob. Margin | 0.2319 | 0.3590 | 0.9877 | 0.9691 | 0.7601 | 0.5556 |
| | Entropy | 0.1244 | 0.2886 | 0.9844 | 0.9540 | 0.7926 | 0.7674 |
| | Avg. Var. | 0.1255 | 0.2178 | 0.9913 | 0.9874 | 1.0000 | 0.9981 |
| | Max. Var. | 0.1141 | 0.2038 | 0.9909 | 0.9865 | 1.0000 | 0.9981 |
| | BALD | 0.1426 | 0.2335 | 0.9877 | 0.9858 | 1.0000 | 0.9981 |
| Scale | Var. Ratio | 0.1704 | 0.3258 | 0.9871 | 0.9741 | 0.7766 | 0.5904 |
| | Prob. Margin | **0.2921** | **0.3808** | 0.9905 | 0.9794 | **0.7167** | **0.4413** |
| | Entropy | 0.1424 | 0.2903 | 0.9860 | 0.9702 | 0.7961 | 0.6500 |
| | Avg. Var. | 0.1567 | 0.2303 | **0.9931** | 0.9886 | 1.0000 | 0.9970 |
| | Max. Var. | 0.1391 | 0.2101 | 0.9924 | 0.9883 | 1.0000 | 0.9971 |
| | BALD | 0.1916 | 0.2515 | 0.9928 | **0.9897** | 1.0000 | 0.9957 |
| Drop | Var. Ratio | 0.1454 | 0.3154 | 0.9855 | 0.9634 | 0.7865 | 0.6321 |
| | Prob. Margin | 0.2453 | 0.3614 | 0.9886 | 0.9703 | 0.7427 | 0.4868 |
| | Entropy | 0.1332 | 0.2888 | 0.9853 | 0.9537 | 0.7991 | 0.6994 |
| | Avg. Var. | 0.1480 | 0.2360 | 0.9921 | 0.9884 | 1.0000 | 0.9987 |
| | Max. Var. | 0.1353 | 0.2164 | 0.9916 | 0.9878 | 1.0000 | 0.9987 |
| | BALD | 0.1673 | 0.2557 | 0.9910 | 0.9890 | 1.0000 | 0.9987 |
| | SegFormer B5 | | | | | | |
| Base | Var. Ratio | 0.3873 | 0.4049 | 0.9969 | 0.9963 | 1.0000 | 0.7554 |
| | Prob. Margin | 0.3771 | 0.3935 | 0.9969 | 0.9963 | 1.0000 | 0.6954 |
| | Entropy | 0.3840 | 0.4038 | 0.9966 | 0.9961 | 1.0000 | 0.5757 |
| Noise | Var. Ratio | 0.3838 | 0.4021 | 0.9969 | 0.9964 | 1.0000 | 0.7575 |
| | Prob. Margin | 0.3724 | 0.3901 | 0.9969 | 0.9963 | 1.0000 | 0.6901 |
| | Entropy | 0.3826 | 0.4023 | 0.9967 | 0.9961 | 1.0000 | 0.5704 |
| | Avg. Var. | 0.3125 | 0.3210 | 0.9966 | 0.9962 | 1.0000 | 1.0000 |
| | Max. Var. | 0.3123 | 0.3204 | 0.9966 | 0.9962 | 1.0000 | 1.0000 |
| | BALD | 0.3073 | 0.3162 | 0.9915 | 0.9912 | 1.0000 | 1.0000 |
| Scale | Var. Ratio | **0.4335** | **0.4422** | **0.9976** | **0.9971** | 1.0000 | 0.3976 |
| | Prob. Margin | 0.4185 | 0.4256 | **0.9976** | **0.9971** | 1.0000 | 0.3642 |
| | Entropy | 0.4099 | 0.4233 | 0.9975 | 0.9965 | 1.0000 | **0.2990** |
| | Avg. Var. | 0.3371 | 0.3334 | 0.9975 | **0.9971** | 1.0000 | 0.9981 |
| | Max. Var. | 0.3352 | 0.3303 | 0.9975 | **0.9971** | 1.0000 | 0.9981 |
| | BALD | 0.3186 | 0.3155 | 0.9960 | 0.9955 | 1.0000 | 0.7832 |
| Drop | Var. Ratio | 0.3876 | 0.4051 | 0.9969 | 0.9963 | 1.0000 | 0.7481 |
| | Prob. Margin | 0.3770 | 0.3934 | 0.9968 | 0.9962 | 1.0000 | 0.6829 |
| | Entropy | 0.3843 | 0.4036 | 0.9966 | 0.9961 | 1.0000 | 0.5651 |
| | Avg. Var. | 0.3334 | 0.3425 | 0.9969 | 0.9965 | 1.0000 | 1.0000 |
| | Max. Var. | 0.3320 | 0.3410 | 0.9969 | 0.9965 | 1.0000 | 1.0000 |
| | BALD | 0.3211 | 0.3327 | 0.9917 | 0.9914 | 1.0000 | 0.9925 |

Finally, Table 13, Table 14, and Table 15 show Precision and Recall for thresholding based on a maximum fraction of pixels allowed on Cityscapes, ADE20K and COCO datasets, respectively. We show results across three threshold values.

As expected, we see that as we increase the maximum number of pixels allowed, we get higher Recall at the cost of Precision. This observation holds true across all of the experiments. Again, we note that Scale often has the best performance with simple uncertainty metrics across both datasets.

Table 13: Micro and macro-averaged ($\mu$ and M, respectively) Precision and Recall for thresholding based on a maximum fraction of pixels allowed on the Cityscapes dataset.

| Scene | Unc. Metric | Max 5% pixels | | | | Max 10% pixels | | | | Max 15% pixels | | | |
|---|---|---|---|---|---|---|---|---|---|---|---|---|---|
| | | Precision ↑ | | Recall ↑ | | Precision ↑ | | Recall ↑ | | Precision ↑ | | Recall ↑ | |
| | | $\mu$ | M | $\mu$ | M | $\mu$ | M | $\mu$ | M | $\mu$ | M | $\mu$ | M |
| | | | | | | DRN D-22 | | | | | | | |
| Base | VR | 45.19 | 45.40 | 41.52 | 50.74 | 35.04 | 35.18 | 64.13 | 73.92 | 28.48 | 28.45 | 76.44 | 84.64 |
| | PM | 44.66 | 44.83 | 41.35 | 50.58 | 34.77 | 34.92 | 64.13 | 73.95 | 28.14 | 28.19 | 76.51 | 84.74 |
| | E | 44.04 | 44.23 | 39.76 | 48.74 | 34.80 | 35.03 | 63.64 | 73.61 | 28.11 | 28.23 | 76.42 | 84.73 |
| Noise | VR | 45.15 | 45.35 | 41.51 | 50.75 | 35.01 | 35.16 | 64.10 | 73.94 | 28.46 | 28.44 | 76.39 | 84.64 |
| | PM | 44.60 | 44.78 | 41.27 | 50.56 | 34.76 | 34.91 | 64.07 | 73.93 | 28.15 | 28.18 | 76.50 | 84.77 |
| | E | 44.02 | 44.20 | 39.75 | 48.77 | 34.79 | 35.02 | 63.64 | 73.64 | 28.12 | 28.23 | 76.45 | 84.78 |
| | AV | 40.06 | 39.93 | 32.11 | 38.45 | 38.03 | 37.47 | 50.03 | 55.62 | 37.92 | 36.97 | 55.23 | 58.76 |
| | MV | 39.79 | 39.66 | 31.88 | 38.17 | 37.81 | 37.22 | 49.85 | 55.45 | 37.65 | 36.70 | 55.34 | 58.86 |
| | B | 36.39 | 36.38 | 30.65 | 36.02 | 32.57 | 32.42 | 50.90 | 57.46 | 30.13 | 29.82 | 63.10 | 68.56 |
| Scale | VR | **47.90** | **48.21** | 43.77 | 52.63 | 36.67 | 36.94 | 67.41 | 76.23 | 29.11 | 29.28 | 79.80 | **86.84** |
| | PM | 47.17 | 47.44 | 43.46 | 52.41 | 36.33 | 36.59 | 67.23 | 76.08 | 28.92 | 29.10 | **79.84** | 86.82 |
| | E | 43.68 | 43.86 | 39.53 | 47.72 | 35.26 | 35.54 | 63.98 | 73.68 | 28.67 | 28.92 | 78.57 | 86.18 |
| | AV | 32.12 | 32.29 | 29.36 | 34.35 | 30.02 | 30.11 | 54.35 | 62.02 | 27.29 | 27.15 | 71.54 | 78.61 |
| | MV | 31.42 | 31.55 | 28.76 | 33.67 | 29.60 | 29.67 | 53.63 | 61.29 | 27.02 | 26.87 | 71.09 | 78.23 |
| | B | 31.13 | 31.32 | 28.39 | 32.45 | 28.38 | 28.55 | 51.55 | 57.83 | 25.34 | 25.41 | 68.24 | 74.57 |
| | | | | | | DRN D-105 | | | | | | | |
| Base | VR | 39.63 | 39.78 | 49.13 | 59.67 | 29.24 | 29.21 | 70.10 | 80.33 | 24.61 | 24.23 | 79.31 | 87.50 |
| | PM | 39.32 | 39.46 | 48.98 | 59.54 | 28.83 | 28.86 | 70.23 | 80.53 | 23.91 | 23.60 | 79.69 | 87.97 |
| | E | 39.72 | 39.95 | 48.70 | 59.33 | 28.85 | 28.94 | 70.48 | 80.80 | 23.15 | 22.99 | 80.26 | 88.63 |
| Noise | VR | 39.57 | 39.70 | 49.01 | 59.57 | 29.24 | 29.19 | 70.09 | 80.33 | 24.62 | 24.24 | 79.34 | 87.55 |
| | PM | 39.23 | 39.35 | 48.89 | 59.48 | 28.82 | 28.83 | 70.23 | 80.55 | 23.94 | 23.62 | 79.72 | 88.01 |
| | E | 39.66 | 39.90 | 48.58 | 59.24 | 28.83 | 28.92 | 70.51 | 80.82 | 23.17 | 23.00 | 80.32 | 88.67 |
| | AV | 40.51 | 40.32 | 40.62 | 48.14 | 39.37 | 38.74 | 50.96 | 56.09 | 39.41 | 38.64 | 52.14 | 56.53 |
| | MV | 40.36 | 40.17 | 40.64 | 48.16 | 39.28 | 38.64 | 51.00 | 56.08 | 39.34 | 38.55 | 52.17 | 56.51 |
| | B | 37.20 | 37.19 | 40.30 | 47.75 | 32.99 | 32.68 | 59.12 | 66.07 | 32.29 | 31.73 | 64.62 | 69.48 |
| Scale | VR | **42.41** | **42.67** | 52.51 | 62.94 | 30.32 | 30.47 | 74.36 | 83.62 | 23.97 | 23.82 | 84.09 | 91.08 |
| | PM | 42.07 | 42.31 | 52.42 | 62.88 | 30.07 | 30.23 | 74.33 | 83.65 | 23.48 | 23.45 | **84.15** | 91.18 |
| | E | 41.11 | 41.40 | 49.76 | 60.38 | 30.00 | 30.29 | 73.84 | 83.50 | 23.16 | 23.24 | 84.14 | **91.32** |
| | AV. | 35.74 | 35.87 | 43.55 | 51.63 | 30.48 | 30.24 | 68.84 | 77.10 | 28.21 | 27.52 | 79.36 | 84.93 |
| | MV. | 35.35 | 35.49 | 43.03 | 51.07 | 30.32 | 30.07 | 68.69 | 76.96 | 28.03 | 27.36 | 79.33 | 84.95 |
| | B | 33.45 | 33.64 | 40.73 | 47.25 | 27.51 | 27.57 | 65.47 | 73.22 | 23.51 | 23.33 | 79.31 | 85.81 |
| | | | | | | OneFormer ConvNeXt-L | | | | | | | |
| Base | VR | 32.16 | 32.31 | 47.73 | 57.40 | 24.77 | 24.84 | 67.92 | 77.28 | 20.80 | 21.03 | 74.47 | 82.96 |
| | PM | 33.41 | 33.78 | 50.18 | 60.30 | 24.95 | 24.99 | 71.11 | 80.37 | 20.58 | 20.68 | 79.04 | 86.67 |
| | E | 30.89 | 31.47 | 44.48 | 54.09 | 23.82 | 24.11 | 64.18 | 74.15 | 20.07 | 20.51 | 70.79 | 79.86 |
| Noise | VR | 32.43 | 32.39 | 48.17 | 57.91 | 24.70 | 24.74 | 67.97 | 77.59 | 20.72 | 20.89 | 74.96 | 83.38 |
| | PM | 33.53 | 33.86 | 50.60 | 60.76 | 24.90 | 25.00 | 70.69 | 80.41 | 20.82 | 20.85 | 79.28 | 86.78 |
| | E | 31.29 | 31.41 | 45.12 | 54.88 | 24.19 | 24.24 | 65.25 | 74.79 | 20.46 | 20.73 | 71.70 | 80.37 |

| | | | | | | | | | | | | | |
|---|---|---|---|---|---|---|---|---|---|---|---|---|---|
| | AV | 28.32 | 30.65 | 23.29 | 27.49 | 25.39 | 29.84 | 27.17 | 30.78 | 24.23 | 29.78 | 28.87 | 32.20 |
| | MV | 26.90 | 29.38 | 21.05 | 24.74 | 23.40 | 28.55 | 24.70 | 27.96 | 22.19 | 28.49 | 26.33 | 29.30 |
| | B | 29.88 | 31.11 | 27.14 | 31.75 | 26.94 | 30.33 | 31.97 | 35.82 | 25.64 | 30.16 | 34.28 | 37.59 |
| Scale | VR | 33.06 | 33.12 | 49.59 | 59.05 | 24.31 | 24.37 | 70.19 | 79.48 | 20.03 | 20.18 | 77.58 | 85.63 |
| | PM | **34.95** | **35.08** | 53.07 | 62.79 | 24.83 | 24.94 | 73.38 | 82.40 | 20.01 | 20.04 | **81.97** | **88.92** |
| | E | 31.49 | 31.81 | 46.19 | 55.45 | 23.63 | 23.77 | 66.65 | 76.48 | 19.21 | 19.49 | 73.98 | 82.95 |
| | AV | 26.24 | 26.37 | 34.74 | 41.71 | 23.57 | 24.22 | 50.06 | 57.43 | 22.03 | 23.46 | 54.38 | 61.20 |
| | MV | 24.47 | 24.64 | 31.43 | 38.02 | 22.32 | 23.08 | 46.09 | 53.41 | 20.73 | 22.42 | 50.57 | 57.34 |
| | B | 27.69 | 27.71 | 38.17 | 45.61 | 24.49 | 24.68 | 55.98 | 63.55 | 22.87 | 23.67 | 60.90 | 67.73 |
| Drop | VR | 32.26 | 32.34 | 48.26 | 57.81 | 24.32 | 24.33 | 68.79 | 78.27 | 19.99 | 20.11 | 76.25 | 84.80 |
| | PM | 33.41 | 33.65 | 50.55 | 60.60 | 24.66 | 24.76 | 71.27 | 80.90 | 20.26 | 20.21 | 80.70 | 87.94 |
| | E | 31.03 | 31.47 | 45.20 | 54.88 | 23.63 | 23.72 | 65.94 | 75.90 | 19.41 | 19.59 | 72.95 | 82.24 |
| | AV | 27.22 | 28.30 | 29.97 | 37.42 | 25.57 | 27.51 | 39.85 | 47.02 | 24.53 | 27.53 | 41.76 | 48.59 |
| | MV | 25.91 | 26.88 | 27.07 | 33.79 | 24.04 | 26.38 | 35.73 | 42.43 | 23.02 | 26.34 | 37.48 | 43.91 |
| | B | 28.84 | 29.05 | 34.44 | 42.68 | 26.69 | 27.84 | 46.91 | 54.52 | 25.52 | 27.71 | 49.72 | 56.72 |
| OneFormer Swin-L | | | | | | | | | | | | | |
| Base | VR | 32.94 | 33.23 | 49.05 | 57.91 | 24.85 | 24.96 | 69.57 | 77.97 | 20.84 | 20.77 | 78.74 | 84.70 |
| | PM | 34.55 | 34.74 | 51.98 | 61.02 | 25.30 | 25.40 | 72.51 | 80.80 | 20.86 | 20.73 | 81.77 | 87.42 |
| | E | 31.81 | 32.27 | 45.96 | 54.81 | 24.24 | 24.56 | 66.79 | 75.25 | 20.25 | 20.52 | 75.56 | 81.89 |
| Noise | VR | 33.11 | 33.47 | 49.38 | 58.24 | 24.97 | 25.05 | 70.12 | 78.27 | 20.99 | 20.94 | 79.10 | 84.90 |
| | PM | 34.73 | 34.88 | 52.24 | 61.27 | 25.36 | 25.39 | 72.93 | 80.98 | 20.70 | 20.58 | 81.90 | 87.55 |
| | E | 32.04 | 32.58 | 46.64 | 55.36 | 24.33 | 24.57 | 67.07 | 75.50 | 20.33 | 20.55 | 76.08 | 82.28 |
| | AV | 29.87 | 31.00 | 26.77 | 30.33 | 27.68 | 30.23 | 31.40 | 34.14 | 26.58 | 30.07 | 32.79 | 35.08 |
| | MV | 28.28 | 29.58 | 24.38 | 27.57 | 26.22 | 28.98 | 28.79 | 31.26 | 25.39 | 28.89 | 30.04 | 32.11 |
| | B | 31.24 | 31.81 | 30.48 | 34.60 | 28.89 | 30.78 | 36.27 | 39.23 | 27.91 | 30.66 | 38.01 | 40.31 |
| Scale | VR | 33.94 | 34.32 | 50.84 | 59.78 | 25.02 | 25.08 | 73.38 | 80.83 | 20.08 | 19.99 | 82.28 | 87.82 |
| | PM | **35.94** | **36.36** | 54.53 | 63.51 | 25.64 | 25.65 | 76.29 | 83.36 | 19.98 | 19.84 | **84.67** | **90.01** |
| | E | 32.44 | 32.95 | 47.77 | 56.50 | 24.29 | 24.38 | 70.42 | 78.41 | 19.44 | 19.53 | 79.54 | 85.80 |
| | AV | 27.75 | 27.90 | 37.81 | 44.46 | 24.89 | 25.01 | 57.01 | 62.72 | 23.77 | 24.16 | 63.79 | 67.53 |
| | MV | 25.97 | 26.06 | 34.85 | 40.95 | 23.92 | 24.09 | 53.77 | 59.28 | 22.74 | 23.24 | 60.39 | 64.22 |
| | B | 29.08 | 29.14 | 41.04 | 47.95 | 25.57 | 25.34 | 61.88 | 67.78 | 24.06 | 24.11 | 69.03 | 72.82 |
| Drop | VR | 32.99 | 33.47 | 49.35 | 58.22 | 24.84 | 24.91 | 70.67 | 78.76 | 20.58 | 20.55 | 79.87 | 85.54 |
| | PM | 34.51 | 34.86 | 52.25 | 61.35 | 25.26 | 25.31 | 73.34 | 81.33 | 20.42 | 20.32 | 82.53 | 88.16 |
| | E | 31.84 | 32.43 | 46.51 | 55.32 | 23.95 | 24.13 | 67.69 | 76.26 | 19.89 | 19.97 | 77.17 | 83.17 |
| | AV | 29.79 | 30.33 | 33.67 | 40.77 | 27.84 | 29.11 | 43.91 | 50.12 | 26.93 | 28.82 | 46.11 | 51.68 |
| | MV | 28.13 | 28.72 | 30.60 | 37.09 | 26.50 | 27.96 | 40.00 | 45.77 | 25.88 | 27.83 | 41.88 | 47.13 |
| | B | 31.21 | 31.36 | 38.42 | 46.14 | 28.73 | 29.50 | 50.96 | 57.41 | 27.71 | 29.06 | 53.38 | 59.08 |
| SegFormer B5 | | | | | | | | | | | | | |
| Base | VR | 36.62 | 36.67 | 54.75 | 63.87 | 27.19 | 26.95 | 74.08 | 81.94 | 24.94 | 24.49 | 80.02 | 86.05 |
| | PM | 36.39 | 36.44 | 54.79 | 63.96 | 26.68 | 26.48 | 74.42 | 82.37 | 23.88 | 23.45 | 81.16 | 87.19 |
| | E | 36.61 | 36.71 | 54.78 | 63.98 | 26.09 | 25.99 | 74.94 | 83.01 | 22.42 | 22.02 | 82.52 | 88.66 |
| Noise | VR | 36.50 | 36.54 | 54.61 | 63.79 | 27.16 | 26.91 | 74.01 | 81.90 | 24.87 | 24.42 | 80.04 | 86.09 |
| | PM | 36.29 | 36.34 | 54.60 | 63.82 | 26.65 | 26.45 | 74.41 | 82.36 | 23.82 | 23.39 | 81.15 | 87.21 |
| | E | 36.49 | 36.60 | 54.55 | 63.83 | 26.04 | 25.94 | 74.84 | 82.98 | 22.36 | 21.96 | 82.56 | 88.71 |
| | AV | **40.42** | **40.39** | 38.53 | 43.44 | 40.25 | 40.13 | 41.31 | 45.00 | 40.19 | 40.11 | 41.61 | 45.11 |
| | MV | **40.42** | 40.36 | 38.38 | 43.21 | 40.19 | 40.07 | 41.25 | 44.84 | 40.18 | 40.07 | 41.46 | 44.90 |
| | B | 35.96 | 36.06 | 42.17 | 48.35 | 34.23 | 34.20 | 52.27 | 56.64 | 34.07 | 34.05 | 53.50 | 57.22 |
| Scale | VR | 38.52 | 38.66 | 57.84 | 66.40 | 27.03 | 26.98 | 78.45 | 85.47 | 22.33 | 21.92 | 86.20 | 91.23 |
| | PM | 38.36 | 38.49 | 57.89 | 66.44 | 26.68 | 26.68 | 78.68 | 85.68 | 21.55 | 21.23 | 86.60 | 91.67 |
| | E | 37.88 | 38.15 | 56.12 | 65.23 | 26.49 | 26.57 | 78.44 | 85.68 | 20.75 | 20.54 | **86.91** | **92.12** |
| | AV | 34.28 | 34.28 | 49.87 | 57.04 | 29.54 | 29.07 | 72.25 | 78.02 | 28.45 | 27.81 | 77.68 | 81.50 |
| | MV | 34.09 | 34.09 | 49.70 | 56.83 | 29.38 | 28.91 | 72.37 | 78.12 | 28.26 | 27.62 | 77.88 | 81.69 |
| | B | 31.46 | 31.51 | 46.09 | 52.15 | 25.41 | 25.31 | 71.04 | 77.64 | 22.48 | 22.11 | 81.72 | 86.61 |

| Drop | VR | 36.62 | 36.67 | 54.79 | 63.92 | 27.17 | 26.93 | 74.15 | 82.01 | 24.81 | 24.35 | 80.25 | 86.25 |
|------|----|-------|-------|-------|-------|-------|-------|-------|-------|-------|-------|-------|-------|
|  | PM | 36.42 | 36.47 | 54.80 | 63.96 | 26.64 | 26.45 | 74.51 | 82.45 | 23.76 | 23.33 | 81.35 | 87.36 |
|  | E | 36.61 | 36.73 | 54.74 | 63.96 | 26.05 | 25.96 | 74.96 | 83.06 | 22.31 | 21.92 | 82.68 | 88.80 |
|  | AV | 38.44 | 38.37 | 47.21 | 54.70 | 37.02 | 36.69 | 56.13 | 61.53 | 37.02 | 36.66 | 56.64 | 61.71 |
|  | MV | 38.39 | 38.33 | 46.98 | 54.43 | 37.06 | 36.72 | 55.80 | 61.20 | 37.06 | 36.69 | 56.31 | 61.38 |
|  | B | 34.73 | 34.82 | 45.35 | 52.84 | 29.60 | 29.53 | 65.38 | 72.44 | 28.96 | 28.77 | 69.43 | 75.02 |

Table 14: Micro and macro-averaged ($\mu$ and M, respectively) Precision and Recall for thresholding based on a maximum fraction of pixels allowed on the ADE20K dataset.

| Scene | Unc. | Max 5% pixels | | | | Max 10% pixels | | | | Max 15% pixels | | | |
|-------|------|---------------|---|---|---|----------------|---|---|---|----------------|---|---|---|
|  | Metric | Precision ↑ | | Recall ↑ | | Precision ↑ | | Recall ↑ | | Precision ↑ | | Recall ↑ | |
|  |  | $\mu$ | M | $\mu$ | M | $\mu$ | M | $\mu$ | M | $\mu$ | M | $\mu$ | M |
| | | | | | | OneFormer ConvNeXt-L | | | | | | | |
| Base | VR | 47.06 | 48.64 | 13.71 | 26.63 | 41.24 | 42.80 | 22.95 | 39.37 | 37.98 | 39.10 | 30.06 | 47.02 |
|  | PM | 50.10 | 52.72 | 13.67 | 27.27 | 44.99 | 47.46 | 24.82 | 42.06 | 41.45 | 43.45 | 34.19 | 51.95 |
|  | E | 43.42 | 45.63 | 11.38 | 23.14 | 38.09 | 40.24 | 19.18 | 34.30 | 34.36 | 36.72 | 24.40 | 40.23 |
| Noise | VR | 46.82 | 48.40 | 13.60 | 26.50 | 40.80 | 42.62 | 22.69 | 39.25 | 37.48 | 38.92 | 29.51 | 46.74 |
|  | PM | 49.72 | 51.84 | 13.56 | 26.82 | 44.48 | 46.83 | 24.65 | 41.97 | 41.35 | 43.13 | 34.27 | 51.96 |
|  | E | 43.29 | 45.73 | 11.36 | 23.11 | 37.44 | 39.87 | 18.91 | 34.23 | 34.22 | 36.50 | 24.11 | 40.07 |
|  | AV | 35.66 | 37.79 | 08.32 | 14.39 | 34.12 | 36.78 | 14.30 | 21.51 | 32.70 | 36.13 | 18.94 | 26.12 |
|  | MV | 34.75 | 37.29 | 07.96 | 13.66 | 33.20 | 36.16 | 13.66 | 20.13 | 31.75 | 35.70 | 17.86 | 24.32 |
|  | B | 37.41 | 37.75 | 09.20 | 14.77 | 34.55 | 35.70 | 16.07 | 23.16 | 33.63 | 34.57 | 22.48 | 29.42 |
| Scale | VR | 48.64 | 50.01 | 14.37 | 27.55 | 41.80 | 43.48 | 23.98 | 40.57 | 38.22 | 39.40 | 31.20 | 48.36 |
|  | PM | **51.85** | **54.22** | 14.63 | 28.57 | 45.91 | 48.29 | 26.21 | 43.75 | 41.99 | 44.14 | **35.70** | **53.54** |
|  | E | 44.02 | 46.12 | 11.83 | 23.81 | 37.77 | 39.70 | 19.77 | 35.24 | 34.16 | 35.94 | 25.65 | 41.55 |
|  | AV | 33.02 | 33.00 | 08.90 | 15.87 | 31.13 | 32.04 | 16.00 | 25.62 | 30.02 | 31.46 | 22.35 | 32.39 |
|  | MV | 31.15 | 31.46 | 08.22 | 14.56 | 29.56 | 30.60 | 14.87 | 23.66 | 28.26 | 30.01 | 20.47 | 29.77 |
|  | B | 37.05 | 37.00 | 10.57 | 18.79 | 34.67 | 34.90 | 19.06 | 30.30 | 32.94 | 33.54 | 26.15 | 38.40 |
| Drop | VR | 47.23 | 48.70 | 13.88 | 26.96 | 41.36 | 42.79 | 23.23 | 39.93 | 37.82 | 38.97 | 30.23 | 47.41 |
|  | PM | 50.62 | 52.55 | 14.06 | 27.53 | 45.01 | 47.40 | 25.44 | 42.56 | 41.51 | 43.43 | 34.83 | 52.39 |
|  | E | 43.48 | 45.72 | 11.61 | 23.58 | 37.55 | 39.90 | 19.32 | 34.64 | 34.41 | 36.36 | 25.01 | 40.94 |
|  | AV | 32.91 | 34.23 | 08.39 | 15.68 | 31.96 | 33.95 | 15.30 | 24.70 | 30.97 | 33.53 | 21.32 | 31.35 |
|  | MV | 31.38 | 33.16 | 07.89 | 14.49 | 30.31 | 32.86 | 14.19 | 22.54 | 29.59 | 32.70 | 19.63 | 28.57 |
|  | B | 35.57 | 36.53 | 09.66 | 18.33 | 34.25 | 35.34 | 18.07 | 29.48 | 33.37 | 34.64 | 25.62 | 38.02 |
| | | | | | | OneFormer Swin-L | | | | | | | |
| Base | VR | 46.97 | 48.39 | 14.18 | 27.16 | 41.28 | 42.31 | 23.92 | 40.21 | 38.27 | 38.78 | 31.45 | 47.94 |
|  | PM | 49.35 | 51.85 | 14.19 | 27.81 | 44.07 | 46.13 | 25.34 | 42.57 | 41.18 | 42.29 | 35.43 | 52.68 |
|  | E | 43.36 | 45.43 | 11.87 | 23.72 | 37.71 | 39.28 | 20.01 | 34.95 | 34.84 | 35.71 | 26.00 | 41.07 |
| Noise | VR | 46.76 | 48.02 | 14.18 | 26.98 | 41.26 | 42.21 | 24.00 | 40.25 | 38.13 | 38.55 | 31.40 | 47.83 |
|  | PM | 49.39 | 51.30 | 14.30 | 27.85 | 44.03 | 45.67 | 25.47 | 42.60 | 40.67 | 41.89 | 35.15 | 52.55 |
|  | E | 42.99 | 45.13 | 11.89 | 23.66 | 37.56 | 39.34 | 19.86 | 34.81 | 34.67 | 35.65 | 26.07 | 41.10 |
|  | AV | 33.66 | 35.65 | 08.40 | 14.71 | 32.44 | 35.30 | 14.46 | 22.00 | 30.69 | 34.96 | 19.15 | 27.13 |
|  | MV | 32.74 | 34.93 | 07.97 | 13.72 | 31.41 | 34.61 | 13.67 | 20.48 | 29.65 | 34.08 | 18.00 | 25.09 |
|  | B | 36.48 | 37.19 | 09.60 | 16.23 | 33.93 | 35.36 | 17.18 | 25.26 | 32.10 | 34.11 | 23.16 | 31.76 |
| Scale | VR | 48.38 | 49.56 | 14.85 | 28.10 | 41.74 | 42.90 | 25.00 | 41.66 | 38.39 | 39.05 | 32.99 | 49.81 |
|  | PM | **51.02** | **52.71** | 15.09 | 28.78 | 45.13 | 46.90 | 27.08 | 44.40 | 41.36 | 42.87 | **36.75** | **54.48** |
|  | E | 43.48 | 45.43 | 12.25 | 24.32 | 37.57 | 39.17 | 20.63 | 35.87 | 34.33 | 35.33 | 26.97 | 42.45 |
|  | AV | 34.14 | 34.12 | 09.65 | 17.18 | 32.33 | 32.78 | 17.57 | 27.72 | 30.42 | 31.87 | 23.84 | 34.88 |
|  | MV | 32.14 | 32.40 | 08.90 | 15.81 | 30.23 | 31.10 | 16.14 | 25.39 | 28.56 | 30.32 | 21.88 | 31.85 |
|  | B | 37.85 | 37.71 | 11.38 | 20.03 | 35.90 | 35.87 | 20.74 | 32.48 | 33.65 | 34.27 | 28.15 | 41.15 |

| | | | | | | | | | | | | |
|---|---|---|---|---|---|---|---|---|---|---|---|---|
| Drop | VR | 47.08 | 48.53 | 14.39 | 27.37 | 41.44 | 42.34 | 24.48 | 40.77 | 38.36 | 38.68 | 31.94 | 48.42 |
| | PM | 50.04 | 52.30 | 14.62 | 28.21 | 44.52 | 46.35 | 26.12 | 43.17 | 41.48 | 42.47 | 36.28 | 53.25 |
| | E | 43.07 | 45.14 | 12.06 | 23.97 | 37.85 | 39.12 | 20.53 | 35.48 | 34.68 | 35.35 | 26.65 | 41.70 |
| | AV | 32.02 | 33.24 | 08.56 | 15.45 | 30.49 | 32.83 | 15.41 | 24.45 | 29.64 | 32.45 | 21.13 | 30.64 |
| | MV | 30.58 | 31.90 | 07.99 | 14.26 | 29.13 | 31.69 | 14.23 | 22.27 | 28.17 | 31.49 | 19.49 | 27.95 |
| | B | 34.35 | 35.55 | 09.68 | 17.86 | 33.17 | 34.43 | 18.12 | 29.11 | 32.11 | 33.38 | 25.36 | 37.35 |
| | | | | | | SegFormer B5 | | | | | | | |
| Base | VR | 55.19 | 55.90 | 16.38 | 29.25 | 48.88 | 49.68 | 29.08 | 46.12 | 44.74 | 45.38 | 39.35 | 57.38 |
| | PM | 53.48 | 54.15 | 15.97 | 28.81 | 48.18 | 48.93 | 28.83 | 45.86 | 44.27 | 44.92 | 39.22 | 57.25 |
| | E | 54.69 | 55.50 | 16.07 | 28.51 | 48.63 | 49.74 | 28.67 | 45.76 | 44.33 | 45.40 | 39.16 | 57.46 |
| Noise | VR | 55.21 | 55.90 | 16.39 | 29.24 | 48.90 | 49.68 | 29.10 | 46.14 | 44.73 | 45.37 | 39.34 | 57.38 |
| | PM | 53.50 | 54.13 | 15.97 | 28.81 | 48.19 | 48.93 | 28.82 | 45.87 | 44.26 | 44.89 | 39.19 | 57.25 |
| | E | 54.61 | 55.45 | 16.02 | 28.48 | 48.60 | 49.71 | 28.66 | 45.74 | 44.29 | 45.35 | 39.13 | 57.45 |
| | AV | 49.94 | 50.55 | 13.87 | 24.11 | 48.42 | 48.36 | 24.92 | 38.74 | 47.80 | 47.03 | 33.24 | 47.63 |
| | MV | 49.83 | 50.43 | 13.79 | 23.94 | 48.28 | 48.24 | 24.78 | 38.57 | 47.61 | 46.92 | 33.19 | 47.52 |
| | B | 48.63 | 49.49 | 13.84 | 23.18 | 45.82 | 46.51 | 25.20 | 38.80 | 44.00 | 44.23 | 34.62 | 49.73 |
| Scale | VR | **57.22** | **58.41** | 16.86 | 29.97 | 50.55 | 51.87 | 30.04 | 47.26 | 45.83 | 47.10 | **40.67** | **58.91** |
| | PM | 54.98 | 56.10 | 16.23 | 29.29 | 49.43 | 50.66 | 29.52 | 46.70 | 45.15 | 46.38 | 40.34 | 58.50 |
| | E | 55.19 | 56.35 | 16.16 | 28.46 | 49.43 | 50.91 | 29.07 | 46.08 | 45.05 | 46.61 | 39.71 | 58.12 |
| | AV | 44.61 | 46.20 | 13.17 | 22.04 | 43.34 | 44.56 | 25.23 | 38.40 | 42.16 | 42.98 | 35.75 | 50.37 |
| | MV | 44.17 | 45.76 | 13.04 | 21.84 | 42.86 | 44.17 | 24.99 | 38.12 | 41.82 | 42.68 | 35.53 | 50.13 |
| | B | 45.28 | 47.00 | 13.38 | 21.49 | 42.71 | 44.29 | 25.22 | 37.42 | 40.59 | 41.95 | 35.64 | 49.61 |
| Drop | VR | 55.26 | 55.97 | 16.40 | 29.26 | 48.92 | 49.72 | 29.12 | 46.18 | 44.71 | 45.40 | 39.38 | 57.45 |
| | PM | 53.53 | 54.18 | 15.98 | 28.81 | 48.20 | 48.95 | 28.85 | 45.89 | 44.27 | 44.93 | 39.23 | 57.30 |
| | E | 54.62 | 55.46 | 16.06 | 28.46 | 48.65 | 49.78 | 28.68 | 45.75 | 44.32 | 45.42 | 39.14 | 57.47 |
| | AV | 51.84 | 52.71 | 15.04 | 26.50 | 48.66 | 49.12 | 27.29 | 43.02 | 46.72 | 46.54 | 37.01 | 53.72 |
| | MV | 51.47 | 52.41 | 14.92 | 26.34 | 48.43 | 48.90 | 27.09 | 42.82 | 46.49 | 46.39 | 36.88 | 53.62 |
| | B | 51.86 | 52.96 | 15.10 | 25.75 | 47.11 | 48.29 | 27.14 | 42.33 | 43.76 | 44.77 | 37.09 | 53.92 |

Table 15: Micro and macro-averaged ($\mu$ and M, respectively) Precision and Recall for thresholding based on a maximum fraction of pixels allowed on the COCO dataset.

| Scene | Unc. | Max 5% pixels | | | | Max 10% pixels | | | | Max 15% pixels | | | |
|---|---|---|---|---|---|---|---|---|---|---|---|---|---|
| | Metric | Precision ↑ | | Recall ↑ | | Precision ↑ | | Recall ↑ | | Precision ↑ | | Recall ↑ | |
| | | $\mu$ | M | $\mu$ | M | $\mu$ | M | $\mu$ | M | $\mu$ | M | $\mu$ | M |
| | | | | | | OneFormer Swin-L | | | | | | | |
| Base | VR | 46.01 | 46.80 | 12.08 | 27.96 | 40.67 | 41.17 | 20.79 | 40.84 | 37.41 | 37.55 | 27.60 | 48.74 |
| | PM | 50.23 | 51.75 | 12.18 | 28.66 | 45.74 | 46.84 | 22.32 | 42.59 | 42.50 | 43.26 | 30.83 | 51.85 |
| | E | 41.53 | 43.09 | 10.15 | 25.04 | 35.64 | 36.69 | 17.15 | 36.08 | 31.90 | 32.46 | 22.25 | 42.34 |
| Scale | VR | 47.05 | 47.90 | 12.46 | 28.57 | 41.39 | 41.88 | 21.52 | 41.93 | 37.85 | 38.01 | 28.56 | 50.00 |
| | PM | **51.84** | **53.49** | **12.79** | **29.48** | **47.03** | **47.98** | **23.40** | **43.98** | **43.46** | **44.01** | **32.21** | **53.48** |
| | E | 41.84 | 43.12 | 10.46 | 25.40 | 35.77 | 36.51 | 17.76 | 36.88 | 31.98 | 32.24 | 23.10 | 43.47 |
| | AV | 37.56 | 36.93 | 9.12 | 19.31 | 35.82 | 35.22 | 16.22 | 29.34 | 34.69 | 34.33 | 22.28 | 36.28 |
| | MV | 35.45 | 34.86 | 8.42 | 17.90 | 33.82 | 33.48 | 15.03 | 27.32 | 32.85 | 32.78 | 20.86 | 33.97 |
| | B | 40.87 | 40.22 | 10.22 | 21.32 | 38.79 | 37.84 | 18.15 | 32.49 | 37.39 | 36.27 | 24.98 | 40.21 |

Considering all these results, we would suggest using either Base or Scale to gauge the trustworthiness of a network. Base is computationally much cheaper, but Scale generally provides better results. As for the uncertainty metric, it is best to use the simpler metrics like Variation Ratio, Probability Margin and Entropy

since they seem to provide the best results in most cases. It also seems that for in-domain data, thresholding based on a maximum fraction of pixels is better than using the largest difference.

## B   Dark Zurich

We now provide all results on the Dark Zurich dataset. Since Dark Zurich was used to test how well the metrics work when there is a shift in the input domain, all networks perform poorly on the dataset. Table 16 shows the base performance of the models on the dataset.

DRN performs very poorly, while OneFormer and SegFormer provide much more respectable results. Overall, the results are quite poor and show that, in general, these networks do not handle large changes in input domain well.

Table 16: Micro and macro-averaged ($\mu$ and M, respectively) Mean Intersection Over Union (mIoU), Expected Calibration Error (ECE) and Brier Score (BS) on the Dark Zurich dataset.

| Model | mIoU ↑ | | ECE ↓ | | BS ↓ | |
|---|---|---|---|---|---|---|
| | $\mu$ Avg. | M Avg. | $\mu$ Avg. | M Avg. | $\mu$ Avg. | M Avg. |
| DRN D-22 | 0.0710 | 0.0737 | 0.3067 | 0.5142 | 1.1303 | 1.1361 |
| DRN D-105 | 0.0951 | 0.0990 | 0.2877 | 0.4173 | 0.9259 | 0.9315 |
| OneFormer ConvNeXt-L | **0.3994** | 0.3752 | 0.5964 | 0.5755 | 0.8340 | 0.8345 |
| OneFormer Swin-L | **0.3994** | **0.3949** | 0.5988 | 0.6678 | 0.8259 | 0.8263 |
| SegFormer B5 | 0.3128 | 0.2686 | **0.2212** | **0.2715** | **0.6053** | **0.6063** |

Table 17 shows the AUROC and the performance of thresholding based on the largest difference, and Table 18 shows the results when we use a maximum fraction of pixels as the thresholding metric.

Even with such poor performance, we observe that the largest difference thresholding is able to achieve AUROC of 0.7 or greater. Precision suffers a bit in this scenario, but Recall is excellent, and while a large fraction of pixels are selected, it is often correct since most pixels are actually misclassified. Scale with Probability Margin as the uncertainty metric often performs the best, similar to earlier experiments.

On the other hand, thresholding based on a maximum fraction of pixels is less useful in this case, with Recall being much lower than the largest difference method. This is because, as explained earlier, most pixels are actually misclassified and restricting it to a maximum of 15% actually hurts performance.

Table 17: Micro and macro-averaged ($\mu$ and M, respectively) Area Under Receiver Operating Characteristic (AUROC) and Precision, Recall and the fraction of pixels selected for largest difference thresholding on the Dark Zurich dataset.

| Scenario | Uncertainty Metric | AUROC ↑ | | Largest Difference Thresholding | | | | | |
|---|---|---|---|---|---|---|---|---|---|
| | | | | Precision ↑ | | Recall ↑ | | Pixel % | |
| | | $\mu$ Avg. | M Avg. | $\mu$ Avg. | M Avg. | $\mu$ Avg. | M Avg. | $\mu$ Avg. | M Avg. |
| | | | | DRN D-22 | | | | | |
| Base | Var. Ratio | 0.6060 | 0.6061 | 70.14 | 70.63 | 74.73 | 75.07 | 70.05 | 70.01 |
| | Prob. Margin | 0.6056 | 0.6067 | 69.67 | 70.18 | 77.53 | 77.85 | 73.16 | 73.11 |
| | Entropy | 0.6054 | 0.6062 | 68.92 | 69.46 | 80.93 | 81.24 | 77.21 | 77.15 |
| Noise | Var. Ratio | 0.6027 | 0.6038 | 69.96 | 70.49 | 74.81 | 75.17 | 70.31 | 70.26 |
| | Prob. Margin | 0.6021 | 0.6041 | 69.50 | 70.05 | 77.57 | 77.91 | 73.39 | 73.33 |
| | Entropy | 0.6024 | 0.6038 | 68.77 | 69.35 | 80.95 | 81.28 | 77.39 | 77.32 |
| | Avg. Var. | 0.5892 | 0.5918 | **75.12** | **75.56** | 46.70 | 46.99 | 40.87 | 40.89 |
| | Max. Var. | 0.5898 | 0.5922 | 75.04 | 75.47 | 47.31 | 47.60 | 41.46 | 41.47 |
| | BALD | 0.6113 | 0.6185 | 73.39 | 74.13 | 63.25 | 63.62 | 56.66 | 56.62 |
| Scale | Var. Ratio | 0.6835 | 0.6853 | 69.46 | 70.14 | 94.25 | 94.42 | 89.22 | 89.21 |

|  |  |  |  |  |  |  |  |  |
|---|---|---|---|---|---|---|---|---|
|  | Prob. Margin | **0.6849** | **0.6864** | 68.47 | 69.26 | **95.12** | **95.40** | 91.34 | 91.31 |
|  | Entropy | 0.6680 | 0.6714 | 71.92 | 72.34 | 79.24 | 79.24 | 72.45 | 72.67 |
|  | Avg. Var. | 0.6769 | 0.6771 | 71.45 | 71.78 | 91.80 | 91.93 | 84.48 | 84.48 |
|  | Max. Var. | 0.6763 | 0.6766 | 71.16 | 71.51 | 92.49 | 92.61 | 85.46 | 85.46 |
|  | BALD | 0.6808 | 0.6809 | 69.02 | 69.77 | 93.96 | 94.32 | 89.51 | 89.41 |
| DRN D-105 |  |  |  |  |  |  |  |  |  |
| Base | Var. Ratio | 0.6909 | 0.6722 | 65.35 | 64.55 | 77.75 | 77.90 | 64.00 | 64.30 |
|  | Prob. Margin | 0.6944 | 0.6749 | 64.45 | 63.68 | 80.71 | 80.82 | 67.35 | 67.66 |
|  | Entropy | 0.6954 | 0.6756 | 63.41 | 62.68 | 83.74 | 83.88 | 71.03 | 71.35 |
| Noise | Var. Ratio | 0.6951 | 0.6765 | 65.42 | 64.54 | 78.16 | 78.25 | 64.26 | 64.55 |
|  | Prob. Margin | 0.6986 | 0.6793 | 64.50 | 63.65 | 81.08 | 81.13 | 67.61 | 67.91 |
|  | Entropy | 0.6994 | 0.6795 | 63.44 | 62.66 | 84.12 | 84.18 | 71.31 | 71.62 |
|  | Avg. Var. | 0.6597 | 0.6499 | **72.46** | **71.56** | 53.21 | 53.57 | 39.50 | 39.70 |
|  | Max. Var. | 0.6594 | 0.6496 | 72.30 | 71.41 | 53.50 | 53.90 | 39.80 | 40.00 |
|  | BALD | 0.6951 | 0.6820 | 68.93 | 68.17 | 69.57 | 69.91 | 54.28 | 54.56 |
| Scale | Var. Ratio | 0.7514 | 0.7291 | 61.32 | 60.82 | 91.13 | 91.52 | 79.94 | 80.29 |
|  | Prob. Margin | **0.7520** | **0.7295** | 60.48 | 60.08 | **91.99** | **92.49** | 81.80 | 82.15 |
|  | Entropy | 0.7410 | 0.7184 | 59.96 | 60.21 | 88.06 | 89.04 | 78.99 | 79.12 |
|  | Avg. Var. | 0.7369 | 0.7168 | 65.89 | 64.95 | 83.86 | 83.77 | 68.46 | 68.80 |
|  | Max. Var. | 0.7368 | 0.7149 | 65.51 | 64.57 | 84.64 | 84.53 | 69.49 | 69.83 |
|  | BALD | 0.7433 | 0.7225 | 62.10 | 61.33 | 90.92 | 90.90 | 78.75 | 79.09 |
| OneFormer ConvNeXt-L |  |  |  |  |  |  |  |  |  |
| Base | Var. Ratio | 0.5855 | 0.6168 | 38.21 | 39.15 | 60.89 | 64.90 | 46.06 | 46.13 |
|  | Prob. Margin | 0.6221 | 0.6604 | 40.17 | 41.14 | 66.90 | 70.26 | 48.14 | 48.38 |
|  | Entropy | 0.5667 | 0.5949 | 38.41 | 39.55 | 56.71 | 60.16 | 42.67 | 42.79 |
| Noise | Var. Ratio | 0.5810 | 0.6297 | 41.12 | 41.79 | 62.40 | 64.63 | 43.86 | 44.11 |
|  | Prob. Margin | 0.6235 | 0.6731 | 44.26 | 44.16 | 70.58 | 71.58 | 46.09 | 46.43 |
|  | Entropy | 0.5641 | 0.6036 | 41.33 | 42.15 | 56.58 | 58.65 | 39.57 | 39.84 |
|  | Avg. Var. | 0.6185 | 0.5884 | 45.33 | 43.47 | 49.41 | 46.04 | 31.51 | 31.94 |
|  | Max. Var. | 0.6061 | 0.5770 | 44.69 | 42.99 | 46.14 | 42.97 | 29.84 | 30.22 |
|  | BALD | 0.6408 | 0.6098 | 46.09 | 44.01 | 54.68 | 51.24 | 34.29 | 34.82 |
| Scale | Var. Ratio | 0.6645 | 0.7152 | 42.57 | 43.90 | 77.82 | 80.84 | 52.83 | 52.83 |
|  | Prob. Margin | 0.7136 | 0.7483 | 42.30 | 42.50 | **86.97** | **88.44** | 59.42 | 59.58 |
|  | Entropy | 0.6377 | 0.7012 | 42.47 | 43.69 | 73.21 | 77.11 | 49.82 | 49.79 |
|  | Avg. Var. | 0.7535 | 0.7529 | 47.01 | 47.51 | 79.58 | 79.77 | 48.93 | 49.27 |
|  | Max. Var. | 0.7375 | 0.7389 | **46.93** | 47.73 | 75.94 | 76.41 | 46.78 | 47.11 |
|  | BALD | **0.7782** | **0.7702** | 46.55 | **48.37** | 84.16 | 83.93 | 52.26 | 52.63 |
| Drop | Var. Ratio | 0.5911 | 0.6229 | 38.94 | 39.56 | 62.37 | 65.38 | 46.30 | 46.49 |
|  | Prob. Margin | 0.6249 | 0.6608 | 40.57 | 41.36 | 70.08 | 73.29 | 49.93 | 50.09 |
|  | Entropy | 0.5755 | 0.6089 | 39.10 | 40.23 | 57.44 | 61.24 | 42.46 | 42.61 |
|  | Avg. Var. | 0.6007 | 0.5568 | 41.51 | 42.21 | 50.29 | 46.64 | 35.02 | 35.36 |
|  | Max. Var. | 0.5884 | 0.5424 | 39.94 | 40.71 | 47.21 | 42.98 | 34.16 | 34.50 |
|  | BALD | 0.6246 | 0.5831 | 43.23 | 43.76 | 61.41 | 57.44 | 41.06 | 41.61 |
| OneFormer Swin-L |  |  |  |  |  |  |  |  |  |
| Base | Var. Ratio | 0.6779 | 0.7245 | 30.32 | 31.88 | 76.77 | 79.86 | 51.50 | 51.69 |
|  | Prob. Margin | 0.7280 | 0.7592 | 30.33 | 31.68 | 82.61 | 85.43 | 55.40 | 55.62 |
|  | Entropy | 0.6555 | 0.7020 | 30.87 | 31.99 | 72.48 | 75.92 | 47.75 | 48.00 |
| Noise | Var. Ratio | 0.6483 | 0.6946 | 29.73 | 31.70 | 75.11 | 78.96 | 51.38 | 51.44 |
|  | Prob. Margin | 0.6926 | 0.7253 | 30.44 | 31.83 | 82.02 | 84.50 | 54.80 | 54.92 |
|  | Entropy | 0.6293 | 0.6770 | 29.85 | 31.36 | 70.65 | 74.42 | 48.13 | 48.15 |
|  | Avg. Var. | 0.6410 | 0.6416 | 33.52 | 35.01 | 56.22 | 56.33 | 34.11 | 34.37 |
|  | Max. Var. | 0.6307 | 0.6288 | 33.44 | 35.05 | 53.37 | 53.36 | 32.46 | 32.70 |
|  | BALD | 0.6670 | 0.6635 | **34.19** | 35.52 | 63.24 | 62.72 | 37.62 | 37.98 |

| | | | | | | | | |
|---|---|---|---|---|---|---|---|---|
| Scale | Var. Ratio | 0.7211 | 0.7414 | 29.35 | 30.90 | 87.33 | 89.37 | 60.52 | 60.78 |
| | Prob. Margin | **0.7577** | **0.7662** | 30.10 | 31.51 | **93.48** | **94.12** | 63.17 | 63.49 |
| | Entropy | 0.7008 | 0.7259 | 29.76 | 31.32 | 80.13 | 82.50 | 54.77 | 54.87 |
| | Avg. Var. | 0.6925 | 0.6958 | 29.62 | 31.29 | 82.56 | 83.32 | 56.69 | 57.03 |
| | Max. Var. | 0.6752 | 0.6806 | 29.78 | 31.03 | 78.57 | 79.66 | 53.65 | 53.92 |
| | BALD | 0.7250 | 0.7221 | 30.32 | 31.52 | 85.74 | 86.40 | 57.50 | 57.76 |
| Drop | Var. Ratio | 0.6956 | 0.7290 | 29.63 | 31.56 | 79.85 | 83.26 | 54.81 | 54.88 |
| | Prob. Margin | 0.7379 | 0.7591 | 29.56 | 31.19 | 86.79 | 88.97 | 59.71 | 59.94 |
| | Entropy | 0.6736 | 0.7098 | 30.07 | 31.84 | 74.97 | 78.39 | 50.70 | 50.75 |
| | Avg. Var. | 0.6046 | 0.6040 | 31.75 | 35.61 | 49.89 | 50.91 | 31.95 | 31.96 |
| | Max. Var. | 0.5834 | 0.5822 | 30.39 | 34.46 | 44.06 | 45.19 | 29.49 | 29.42 |
| | BALD | 0.6471 | 0.6387 | 33.59 | **36.63** | 59.41 | 59.76 | 35.97 | 36.06 |
| | | | | | SegFormer B5 | | | | |
| Base | Var. Ratio | 0.7999 | 0.7978 | 61.71 | 60.47 | 83.55 | 83.24 | 48.95 | 49.18 |
| | Prob. Margin | 0.8053 | 0.8033 | 60.56 | 59.34 | 85.81 | 85.53 | 51.23 | 51.46 |
| | Entropy | 0.8155 | 0.8156 | 58.55 | 57.43 | 89.42 | 89.24 | 55.22 | 55.47 |
| Noise | Var. Ratio | 0.7949 | 0.7903 | 62.18 | 60.54 | 82.73 | 82.46 | 48.10 | 48.26 |
| | Prob. Margin | 0.8012 | 0.7966 | 61.02 | 59.40 | 85.26 | 84.98 | 50.52 | 50.69 |
| | Entropy | 0.8130 | 0.8092 | 58.85 | 57.44 | 89.09 | 88.89 | 54.74 | 54.93 |
| | Avg. Var. | 0.6740 | 0.6721 | 67.72 | 67.33 | 47.74 | 47.73 | 25.48 | 25.56 |
| | Max. Var. | 0.6743 | 0.6724 | 67.80 | 67.37 | 47.72 | 47.71 | 25.45 | 25.51 |
| | BALD | 0.7243 | 0.7217 | 65.13 | 64.02 | 63.62 | 63.44 | 35.31 | 35.39 |
| Scale | Var. Ratio | 0.8474 | 0.8490 | 56.87 | 55.85 | 94.71 | 94.56 | 60.21 | 60.54 |
| | Prob. Margin | 0.8483 | 0.8491 | 55.76 | 54.76 | 95.73 | 95.62 | 62.06 | 62.38 |
| | Entropy | **0.8519** | **0.8566** | 53.81 | 52.90 | **97.22** | **97.22** | 65.32 | 65.63 |
| | Avg. Var. | 0.8064 | 0.8045 | 62.63 | 61.51 | 84.98 | 84.79 | 49.06 | 49.35 |
| | Max. Var. | 0.8088 | 0.8060 | 62.39 | 61.25 | 85.59 | 85.37 | 49.60 | 49.90 |
| | BALD | 0.8247 | 0.8249 | 57.76 | 56.68 | 93.09 | 92.86 | 58.27 | 58.59 |
| Drop | Var. Ratio | 0.8013 | 0.7990 | 61.58 | 60.33 | 83.92 | 83.58 | 49.27 | 49.50 |
| | Prob. Margin | 0.8065 | 0.8043 | 60.42 | 59.19 | 86.15 | 85.83 | 51.55 | 51.78 |
| | Entropy | 0.8165 | 0.8163 | 58.37 | 57.24 | 89.68 | 89.48 | 55.55 | 55.80 |
| | Avg. Var. | 0.6980 | 0.6990 | 69.68 | 68.43 | 51.47 | 51.90 | 26.70 | 26.78 |
| | Max. Var. | 0.6979 | 0.6976 | **70.03** | **68.67** | 51.20 | 51.42 | 26.43 | 26.50 |
| | BALD | 0.7539 | 0.7532 | 65.18 | 64.02 | 69.57 | 69.40 | 38.59 | 38.75 |

Table 18: Micro and macro-averaged ($\mu$ and M, respectively) Precision and Recall for thresholding based on a maximum fraction of pixels allowed on the Dark Zurich dataset.

| Scene | Unc. Metric | Max 5% pixels | | | | Max 10% pixels | | | | Max 15% pixels | | | |
|---|---|---|---|---|---|---|---|---|---|---|---|---|---|
| | | Precision ↑ | | Recall ↑ | | Precision ↑ | | Recall ↑ | | Precision ↑ | | Recall ↑ | |
| | | $\mu$ | M | $\mu$ | M | $\mu$ | M | $\mu$ | M | $\mu$ | M | $\mu$ | M |
| | | | | | | DRN D-22 | | | | | | | |
| Base | VR | 80.81 | 80.86 | 05.86 | 05.94 | 79.04 | 79.14 | 11.65 | 11.79 | 77.96 | 78.11 | 17.41 | 17.61 |
| | PM | 78.91 | 79.08 | 05.74 | 05.81 | 78.24 | 78.41 | 11.63 | 11.76 | 77.62 | 77.77 | 17.41 | 17.59 |
| | E | 81.07 | 81.12 | 05.92 | 06.01 | 78.88 | 78.96 | 11.57 | 11.74 | 77.51 | 77.58 | 17.03 | 17.23 |
| Noise | VR | 79.99 | 80.04 | 05.75 | 05.83 | 78.31 | 78.44 | 11.49 | 11.63 | 77.40 | 77.56 | 17.24 | 17.43 |
| | PM | 77.93 | 78.09 | 05.63 | 05.69 | 77.36 | 77.56 | 11.45 | 11.58 | 76.98 | 77.16 | 17.27 | 17.45 |
| | E | 80.71 | 80.78 | 05.92 | 06.01 | 78.48 | 78.56 | 11.59 | 11.74 | 77.12 | 77.22 | 16.98 | 17.19 |
| | AV | 77.78 | 77.89 | 05.65 | 05.69 | 77.46 | 77.62 | 11.13 | 11.20 | 77.28 | 77.46 | 16.58 | 16.68 |
| | MV | 77.75 | 77.81 | 05.64 | 05.68 | 77.27 | 77.53 | 11.18 | 11.25 | 77.17 | 77.43 | 16.57 | 16.69 |
| | B | 77.99 | 78.09 | 05.65 | 05.70 | 77.99 | 78.30 | 11.22 | 11.32 | 77.93 | 78.23 | 16.78 | 16.92 |

| | | | | | | | | | | | | |
|---|---|---|---|---|---|---|---|---|---|---|---|---|
| Scale | VR | 83.99 | 84.07 | 06.03 | 06.13 | 82.50 | 82.54 | 11.97 | 12.16 | 81.54 | 81.62 | 17.83 | 18.10 |
| | PM | 82.04 | 82.13 | 05.49 | 05.56 | 81.63 | 81.73 | 11.75 | 11.92 | 81.09 | 81.22 | **17.87** | **18.13** |
| | E | **84.60** | **84.61** | 06.08 | 06.18 | 81.69 | 81.80 | 11.88 | 12.08 | 80.24 | 80.34 | 17.67 | 17.94 |
| | AV | 76.09 | 76.20 | 05.45 | 05.50 | 76.80 | 76.91 | 11.22 | 11.33 | 77.37 | 77.53 | 16.96 | 17.13 |
| | MV | 75.39 | 75.64 | 05.48 | 05.52 | 76.65 | 76.87 | 11.27 | 11.37 | 77.42 | 77.68 | 17.04 | 17.21 |
| | B | 81.40 | 81.51 | 05.87 | 05.94 | 79.81 | 79.95 | 11.60 | 11.73 | 79.19 | 79.32 | 17.43 | 17.61 |
| DRN D-105 | | | | | | | | | | | | | |
| Base | VR | 79.49 | 79.57 | 06.99 | 07.54 | 77.15 | 77.21 | 13.95 | 14.92 | 75.53 | 75.57 | 20.69 | 21.99 |
| | PM | 78.21 | 78.21 | 07.00 | 07.49 | 76.79 | 76.80 | 14.02 | 14.95 | 75.38 | 75.42 | 20.73 | 22.02 |
| | E | 78.96 | 79.07 | 07.03 | 07.59 | 76.02 | 76.04 | 13.50 | 14.44 | 74.43 | 74.49 | 19.73 | 21.04 |
| Noise | VR | 79.45 | 79.67 | 07.01 | 07.55 | 77.69 | 77.87 | 14.08 | 15.07 | 76.24 | 76.40 | 20.83 | 22.21 |
| | PM | 78.74 | 78.86 | 07.01 | 07.54 | 77.46 | 77.60 | 14.10 | 15.06 | 76.13 | 76.30 | 20.91 | 22.28 |
| | E | 78.57 | 78.76 | 06.98 | 07.54 | 76.16 | 76.30 | 13.49 | 14.45 | 74.78 | 74.96 | 19.91 | 21.27 |
| | AV | 82.54 | 82.55 | 07.32 | 07.80 | 80.39 | 80.35 | 14.29 | 15.18 | 78.53 | 78.51 | 20.94 | 22.18 |
| | MV | 82.35 | 82.37 | 07.34 | 07.82 | 80.21 | 80.12 | 14.26 | 15.14 | 78.36 | 78.36 | 20.82 | 22.01 |
| | B | 83.58 | 83.48 | 07.41 | 07.91 | 81.03 | 81.08 | 14.36 | 15.31 | 79.27 | 79.24 | 21.10 | 22.38 |
| Scale | VR | **85.07** | **85.04** | 07.44 | 08.09 | 83.39 | 83.39 | 14.81 | 15.95 | 81.64 | 81.69 | 21.89 | 23.48 |
| | PM | 83.96 | 83.95 | 07.22 | 07.84 | 82.98 | 82.98 | 14.79 | 15.93 | 81.75 | 81.75 | **22.28** | **23.85** |
| | E | 81.47 | 81.53 | 07.09 | 07.70 | 78.75 | 78.91 | 13.96 | 15.06 | 76.88 | 77.00 | 20.53 | 21.96 |
| | AV | 80.17 | 80.07 | 07.04 | 07.52 | 79.30 | 79.18 | 14.21 | 15.06 | 78.35 | 78.17 | 21.11 | 22.31 |
| | MV | 79.46 | 79.40 | 07.02 | 07.49 | 78.23 | 78.17 | 14.07 | 14.93 | 77.44 | 77.35 | 20.89 | 22.10 |
| | B | 83.78 | 83.56 | 07.35 | 07.95 | 81.43 | 81.31 | 14.51 | 15.53 | 79.77 | 79.67 | 21.41 | 22.75 |
| OneFormer ConvNeXt-L | | | | | | | | | | | | | |
| Base | VR | 55.89 | 55.76 | 08.77 | 09.96 | 50.95 | 51.16 | 16.68 | 18.75 | 49.33 | 49.48 | 24.12 | 27.14 |
| | PM | 56.90 | 57.26 | 08.92 | 10.33 | 52.74 | 52.69 | 17.00 | 19.15 | 50.50 | 50.72 | 25.34 | 28.66 |
| | E | 53.66 | 57.98 | 07.10 | 07.95 | 50.51 | 52.77 | 15.43 | 17.23 | 48.45 | 48.76 | 23.19 | 26.27 |
| Noise | VR | 50.88 | 50.21 | 08.01 | 09.27 | 48.60 | 48.81 | 15.88 | 18.17 | 48.02 | 48.27 | 23.52 | 26.45 |
| | PM | 53.23 | 53.87 | 08.42 | 09.64 | 51.40 | 51.91 | 16.85 | 19.14 | 51.08 | 51.35 | 25.31 | 28.26 |
| | E | 49.02 | 51.82 | 06.52 | 07.39 | 49.29 | 48.66 | 15.02 | 17.19 | 48.25 | 48.43 | 22.81 | 25.61 |
| | AV | 46.85 | 45.04 | 06.36 | 07.00 | 44.08 | 43.39 | 12.61 | 13.80 | 43.62 | 43.65 | 17.48 | 19.20 |
| | MV | 45.42 | 43.87 | 06.00 | 06.62 | 43.74 | 42.74 | 11.67 | 12.69 | 42.86 | 42.59 | 16.37 | 17.96 |
| | B | 48.90 | 47.01 | 07.34 | 08.14 | 46.53 | 45.87 | 13.75 | 15.10 | 46.43 | 46.20 | 20.19 | 22.12 |
| Scale | VR | 52.04 | 51.60 | 07.88 | 08.93 | 48.94 | 49.20 | 15.42 | 17.39 | 50.30 | 50.41 | 24.27 | 27.58 |
| | PM | 46.41 | 47.38 | 06.59 | 07.49 | 48.77 | 48.99 | 15.34 | 17.39 | 50.86 | 50.86 | 24.99 | 28.57 |
| | E | 51.66 | 50.42 | 06.92 | 08.08 | 51.24 | 51.17 | 14.99 | 17.19 | 51.33 | 51.43 | 23.35 | 26.61 |
| | AV | 52.01 | 51.73 | 07.79 | 08.84 | 56.87 | 55.85 | 16.98 | 19.21 | 57.48 | 56.90 | 25.91 | 29.41 |
| | MV | 51.34 | 49.82 | 06.96 | 07.83 | 55.74 | 53.54 | 14.94 | 17.66 | 57.64 | 55.91 | 24.48 | 28.40 |
| | B | **57.51** | **57.99** | 09.13 | 10.25 | 58.43 | 58.94 | 18.77 | 21.02 | 58.65 | 59.66 | **28.41** | **31.66** |
| Drop | VR | 51.98 | 51.69 | 08.09 | 09.25 | 49.13 | 49.10 | 16.15 | 18.31 | 48.15 | 48.37 | 23.93 | 26.97 |
| | PM | 51.11 | 51.05 | 07.99 | 09.14 | 49.69 | 49.70 | 16.39 | 18.56 | 49.00 | 49.12 | 24.54 | 27.56 |
| | E | 52.52 | 50.47 | 06.85 | 07.76 | 50.17 | 48.08 | 15.28 | 17.34 | 47.78 | 47.82 | 22.84 | 25.84 |
| | AV | 38.57 | 37.21 | 04.57 | 05.25 | 39.58 | 38.91 | 09.37 | 10.53 | 37.29 | 36.68 | 13.84 | 15.12 |
| | MV | 38.98 | 36.21 | 04.45 | 05.15 | 37.04 | 34.95 | 08.49 | 09.52 | 36.13 | 35.49 | 13.12 | 14.32 |
| | B | 39.47 | 38.96 | 04.86 | 05.60 | 39.04 | 40.32 | 10.22 | 11.43 | 39.46 | 39.81 | 15.44 | 17.16 |
| OneFormer Swin-L | | | | | | | | | | | | | |
| Base | VR | 56.31 | 57.18 | 12.60 | 13.72 | 52.36 | 52.90 | 24.12 | 26.45 | 48.69 | 49.42 | 33.07 | 36.21 |
| | PM | **56.72** | **57.57** | 11.37 | 13.35 | 55.15 | 55.24 | 25.23 | 28.12 | 51.94 | 52.22 | **36.61** | **40.07** |
| | E | 52.69 | 55.74 | 09.64 | 11.01 | 50.94 | 53.44 | 21.54 | 23.85 | 47.18 | 49.77 | 29.67 | 32.64 |
| Noise | VR | 47.88 | 49.03 | 10.66 | 11.51 | 45.92 | 46.34 | 21.15 | 22.78 | 42.87 | 43.36 | 29.80 | 31.88 |
| | PM | 47.28 | 49.55 | 10.06 | 11.08 | 46.03 | 46.35 | 21.56 | 23.40 | 44.13 | 44.37 | 31.53 | 33.76 |
| | E | 47.89 | 50.21 | 09.52 | 10.16 | 45.17 | 47.59 | 19.38 | 21.13 | 42.15 | 42.67 | 28.22 | 30.05 |
| | AV | 40.68 | 39.68 | 08.21 | 08.92 | 37.55 | 38.22 | 15.33 | 16.84 | 37.95 | 38.76 | 22.46 | 24.85 |
| | MV | 37.52 | 38.19 | 07.38 | 08.13 | 37.55 | 38.72 | 14.63 | 16.60 | 35.51 | 36.92 | 21.04 | 23.13 |

| | | | | | | | | | | | | | |
|---|---|---|---|---|---|---|---|---|---|---|---|---|---|
| | B | 42.65 | 42.53 | 09.14 | 09.80 | 41.62 | 41.63 | 17.70 | 19.21 | 39.95 | 40.22 | 25.51 | 27.80 |
| Scale | VR | 48.65 | 49.65 | 10.46 | 11.12 | 46.42 | 47.06 | 20.92 | 22.38 | 45.17 | 45.57 | 31.19 | 33.47 |
| | PM | 43.23 | 46.15 | 08.91 | 09.53 | 44.32 | 44.66 | 20.25 | 21.83 | 45.21 | 45.28 | 31.92 | 34.26 |
| | E | 46.59 | 48.57 | 09.34 | 10.13 | 46.59 | 48.38 | 19.30 | 21.02 | 44.81 | 45.35 | 29.41 | 31.71 |
| | AV | 36.17 | 35.70 | 08.00 | 08.83 | 36.06 | 35.52 | 16.22 | 17.54 | 36.04 | 36.23 | 24.61 | 26.69 |
| | MV | 33.22 | 32.79 | 07.13 | 08.06 | 34.45 | 33.58 | 14.99 | 16.49 | 35.30 | 34.60 | 23.07 | 25.30 |
| | B | 42.63 | 42.15 | 09.55 | 09.89 | 41.67 | 41.58 | 19.41 | 20.26 | 40.87 | 41.04 | 28.37 | 29.89 |
| Drop | VR | 53.68 | 54.48 | 11.89 | 12.78 | 50.24 | 50.64 | 23.10 | 25.08 | 47.42 | 47.96 | 33.15 | 35.84 |
| | PM | 50.47 | 51.85 | 10.43 | 11.89 | 51.13 | 51.63 | 23.81 | 26.40 | 49.95 | 50.33 | 35.11 | 38.40 |
| | E | 52.31 | 56.24 | 09.76 | 10.67 | 49.54 | 52.10 | 21.05 | 23.13 | 46.24 | 48.85 | 30.17 | 33.00 |
| | AV. | 32.22 | 36.01 | 05.65 | 06.56 | 35.33 | 37.06 | 12.58 | 14.02 | 33.60 | 35.12 | 16.58 | 18.30 |
| | MV. | 31.64 | 35.13 | 05.12 | 05.95 | 32.18 | 33.85 | 10.85 | 12.03 | 30.54 | 32.84 | 15.57 | 17.30 |
| | B | 36.19 | 37.58 | 07.08 | 08.17 | 38.70 | 40.07 | 13.87 | 15.61 | 37.65 | 39.60 | 19.59 | 21.82 |
| | | | | | | SegFormer B5 | | | | | | | |
| Base | VR | 72.70 | 72.67 | 09.58 | 11.22 | 71.27 | 71.21 | 19.25 | 22.15 | 69.95 | 69.87 | 28.38 | 32.22 |
| | PM | 72.24 | 72.19 | 09.73 | 11.32 | 71.05 | 71.00 | 19.32 | 22.18 | 69.75 | 69.64 | 28.57 | 32.32 |
| | E | 72.93 | 72.80 | 09.47 | 11.01 | 71.84 | 71.79 | 18.83 | 21.77 | 70.47 | 70.46 | 28.09 | 32.03 |
| Noise | VR | 70.16 | 70.08 | 09.29 | 10.96 | 69.24 | 69.22 | 18.66 | 21.70 | 68.26 | 68.20 | 27.78 | 31.80 |
| | PM | 69.36 | 69.36 | 09.30 | 10.92 | 69.00 | 68.94 | 18.82 | 21.81 | 68.08 | 68.02 | 27.87 | 31.87 |
| | E | 71.56 | 71.48 | 09.34 | 10.99 | 70.72 | 70.54 | 18.61 | 21.78 | 68.74 | 68.77 | 27.30 | 31.54 |
| | AV | 69.60 | 69.58 | 08.94 | 10.26 | 68.89 | 68.89 | 17.37 | 19.75 | 68.13 | 68.16 | 25.14 | 28.08 |
| | MV | 69.83 | 69.78 | 09.03 | 10.39 | 68.77 | 68.92 | 17.49 | 20.04 | 68.30 | 68.16 | 25.17 | 28.07 |
| | B | 68.81 | 68.80 | 08.88 | 10.21 | 67.70 | 67.64 | 16.99 | 19.23 | 66.81 | 66.66 | 24.81 | 27.69 |
| Scale | VR | **77.93** | **77.94** | 10.22 | 11.91 | 76.02 | 76.08 | 20.21 | 23.45 | 74.41 | 74.45 | **30.01** | **34.19** |
| | PM | 75.47 | 75.56 | 09.87 | 11.43 | 74.68 | 74.77 | 20.04 | 23.06 | 73.60 | 73.63 | 29.95 | 34.02 |
| | E | 75.50 | 75.60 | 09.76 | 11.42 | 75.39 | 75.33 | 19.60 | 22.63 | 74.43 | 74.41 | 29.45 | 33.66 |
| | AV. | 67.95 | 68.02 | 08.96 | 10.33 | 68.42 | 68.45 | 18.39 | 21.15 | 69.37 | 69.32 | 28.13 | 31.94 |
| | MV. | 66.71 | 66.83 | 08.83 | 10.13 | 68.17 | 68.16 | 18.40 | 21.12 | 69.45 | 69.33 | 28.25 | 31.95 |
| | B | 69.76 | 69.68 | 09.21 | 10.68 | 68.56 | 68.62 | 18.32 | 21.08 | 68.93 | 68.88 | 27.78 | 31.49 |
| Drop | VR | 72.80 | 72.78 | 09.66 | 11.28 | 71.46 | 71.37 | 19.38 | 22.24 | 70.06 | 69.97 | 28.56 | 32.37 |
| | PM | 72.48 | 72.43 | 09.75 | 11.30 | 71.25 | 71.16 | 19.40 | 22.19 | 69.84 | 69.74 | 28.59 | 32.34 |
| | E | 72.92 | 72.78 | 09.47 | 10.98 | 71.92 | 71.81 | 18.99 | 21.88 | 70.55 | 70.56 | 27.98 | 31.86 |
| | AV | 75.17 | 74.96 | 09.59 | 10.93 | 73.69 | 73.33 | 18.39 | 20.67 | 72.07 | 71.87 | 26.53 | 29.74 |
| | MV | 75.22 | 74.96 | 09.58 | 10.84 | 73.62 | 73.33 | 18.30 | 20.62 | 72.26 | 72.06 | 26.56 | 29.70 |
| | B | 74.30 | 73.99 | 09.48 | 10.74 | 72.36 | 72.05 | 18.21 | 20.52 | 70.46 | 70.54 | 26.30 | 29.47 |

The results on Dark Zurich suggest much the same things as those on Cityscapes and ADE20K that Scale with simpler metrics is probably the best choice. One major difference is that if domain shift is expected, it is better to use the largest difference thresholding method than a maximum fraction of allowed pixels.

## C   Calibration Experiments

In order to improve our results we run experiments on calibrated OneFormer ConvNeXt-L and SegFormer B5 models. Specifically, we use temperature scaling on the logits during softmax computation to calibrate the models. We use temperatures of 0.2 and 2 for OneFormer and SegFormer models respectively. We first show the calibration results of such scaling in Table 19. Model calibration is evaluated using Expected Calibration Error (ECE) and Brier Score (BS). As can be seen after temperature scaling both models are better calibrated.

Table 19: Micro and macro-averaged ($\mu$ and M, respectively) Expected Calibration Error (ECE) and Brier Score (BS) for OneFormer ConvNeXt-L and SegFormer B5 models on the Cityscapes dataset

| Calibration | ECE ↓ | | BS ↓ | |
|---|---|---|---|---|
| Status | $\mu$ Avg. | M Avg. | $\mu$ Avg. | M Avg. |
| OneFormer ConvNeXt-L | | | | |
| Uncalibrated | 0.7887 | 0.8383 | 0.8012 | 0.8011 |
| Calibrated | **0.1173** | **0.1084** | **0.0668** | **0.0681** |
| SegFormer B5 | | | | |
| Uncalibrated | 0.0631 | 0.0160 | 0.0516 | 0.0524 |
| Calibrated | **0.0282** | **0.0125** | **0.0506** | **0.0514** |

We now show the Area Under the Receiver Operating Characteristics (AUROC) as well as the Area Under the Precision Recall Curve (AUPRC) in Table 20 and Table 21, respectively. We observe that in almost all cases the calibrated models perform better than uncalibrated ones. The only departure from this pattern is AUPRC for SegFormer B5 with the Scale setting.

Table 20: Micro and macro-averaged ($\mu$ and M, respectively) Area Under Receiver Operating Characteristic (AUROC) for uncalibrated and calibrated models on the Cityscapes dataset.

| Scenario | Uncertainty Metric | Uncalibrated AUROC ↑ | | Calibrated AUROC ↑ | |
|---|---|---|---|---|---|
| | | $\mu$ Avg. | M Avg. | $\mu$ Avg. | M Avg. |
| | OneFormer ConvNeXt-L | | | | |
| Base | Var. Ratio | *0.7265* | *0.8822* | **0.8458** | **0.8848** |
| | Prob. Margin | *0.7893* | **0.9073** | **0.8543** | *0.8913* |
| | Entropy | *0.7089* | *0.8569* | **0.8387** | **0.8896** |
| Scale | Var. Ratio | *0.7647* | *0.8972* | **0.8896** | **0.9163** |
| | Prob. Margin | *0.8305* | *0.9215* | **0.8959** | **0.9221** |
| | Entropy | *0.7350* | *0.8736* | **0.8805** | **0.9166** |
| | Avg. Var. | *0.7835* | *0.7880* | **0.8268** | **0.8324** |
| | Max. Var. | *0.7637* | *0.7677* | **0.8383** | **0.8453** |
| | BALD | *0.8173* | *0.8213* | **0.8496** | **0.8563** |
| | SegFormer B5 | | | | |
| Base | Var. Ratio | *0.8830* | *0.9087* | **0.9328** | **0.9468** |
| | Prob. Margin | *0.8892* | *0.9142* | **0.9351** | **0.9485** |
| | Entropy | *0.8997* | *0.9234* | **0.9357** | **0.9467** |
| Scale | Var. Ratio | *0.9212* | *0.9391* | **0.9459** | **0.9569** |
| | Prob. Margin | *0.9245* | *0.9418* | **0.9477** | **0.9582** |
| | Entropy | *0.9290* | *0.9459* | **0.9437** | **0.9522** |
| | Avg. Var. | *0.8678* | *0.8826* | **0.8887** | **0.9010** |
| | Max. Var. | *0.8687* | *0.8832* | **0.8864** | **0.8978** |
| | BALD | *0.8947* | **0.9091** | **0.8990** | *0.9067* |

Table 21: Micro and macro-averaged ($\mu$ and M, respectively) Area Under Precision Recall Curve (AUPRC) for uncalibrated and calibrated models on the Cityscapes dataset.

| Scenario | Uncertainty | Uncalibrated | Calibrated |
|---|---|---|---|

| | Metric | AUPRC-Error ↑ | | AUPRC-Success ↑ | | AUPRC-Error ↑ | | AUPRC-Success ↑ | |
|---|---|---|---|---|---|---|---|---|---|
| | | μ Avg. | M Avg. | μ Avg. | M Avg. | μ Avg. | M Avg. | μ Avg. | M Avg. |
| | | | | OneFormer ConvNeXt-L | | | | | |
| Base | Var. Ratio | 0.1310 | 0.3090 | 0.9853 | 0.9633 | 0.2834 | 0.3173 | 0.9888 | 0.9611 |
| | Prob. Margin | 0.2300 | 0.3562 | 0.9885 | 0.9722 | 0.3107 | 0.3397 | 0.9892 | 0.9645 |
| | Entropy | 0.1193 | 0.2832 | 0.9847 | 0.9548 | 0.2685 | 0.3078 | 0.9885 | 0.9669 |
| Scale | Var. Ratio | 0.1704 | 0.3258 | 0.9871 | 0.9741 | 0.3169 | 0.3417 | 0.9918 | 0.9786 |
| | Prob. Margin | 0.2921 | 0.3808 | 0.9905 | 0.9794 | 0.3565 | 0.3717 | 0.9922 | 0.9797 |
| | Entropy | 0.1424 | 0.2903 | 0.9860 | 0.9702 | 0.2858 | 0.3198 | 0.9922 | 0.9822 |
| | Avg. Var. | 0.1567 | 0.2303 | 0.9931 | 0.9886 | 0.3136 | 0.2989 | 0.9949 | 0.9917 |
| | Max. Var. | 0.1391 | 0.2101 | 0.9924 | 0.9883 | 0.3029 | 0.2885 | 0.9951 | 0.9915 |
| | BALD | 0.1916 | 0.2515 | 0.9928 | 0.9897 | 0.2984 | 0.2959 | 0.9941 | 0.9918 |
| | | | | SegFormer B5 | | | | | |
| Base | Var. Ratio | 0.3873 | 0.4049 | 0.9969 | 0.9963 | 0.3901 | 0.4062 | 0.9975 | 0.9962 |
| | Prob. Margin | 0.3771 | 0.3935 | 0.9969 | 0.9963 | 0.3831 | 0.3998 | 0.9976 | 0.9963 |
| | Entropy | 0.3840 | 0.4038 | 0.9966 | 0.9961 | 0.3382 | 0.3486 | 0.9975 | 0.9965 |
| Scale | Var. Ratio | 0.4335 | 0.4422 | 0.9976 | 0.9971 | 0.4204 | 0.4327 | 0.9980 | 0.9968 |
| | Prob. Margin | 0.4185 | 0.4256 | 0.9976 | 0.9971 | 0.4146 | 0.4258 | 0.9980 | 0.9969 |
| | Entropy | 0.4099 | 0.4233 | 0.9975 | 0.9965 | 0.3439 | 0.3524 | 0.9977 | 0.9970 |
| | Avg. Var. | 0.3371 | 0.3334 | 0.9975 | 0.9971 | 0.2716 | 0.2673 | 0.9961 | 0.9954 |
| | Max. Var. | 0.3352 | 0.3303 | 0.9975 | 0.9971 | 0.2596 | 0.2546 | 0.9960 | 0.9952 |
| | BALD | 0.3186 | 0.3155 | 0.9960 | 0.9955 | 0.2409 | 0.2391 | 0.9958 | 0.9949 |

# D   Experiments With Noisy Inputs

Here, we provide the results of our experiments with noisy inputs. This experiment was only performed on the Cityscapes dataset with DRN D-22 and OneFormer ConvNeXt-L models. Table 22 shows the base performances across varying noise levels.

As noise levels increase, we see a decrease in performance for both models, although, as seen earlier DRN gives much worse outputs than OneFormer. DRN's calibration also worsens with each noise level, whereas OneFormer's calibration stays mostly the same.

Table 22: Micro and macro-averaged ($\mu$ and M, respectively) Mean Intersection Over Union (mIoU), Expected Calibration Error (ECE) and Brier Score (BS) on the Cityscapes dataset for noisy inputs.

| Noise Level | mIoU ↑ | | ECE ↓ | | BS ↓ | |
|---|---|---|---|---|---|---|
| | μ Avg. | M Avg. | μ Avg. | M Avg. | μ Avg. | M Avg. |
| | | | DRN D-22 | | | |
| None | **0.6790** | **0.5479** | 0.0380 | **0.0177** | **0.0808** | **0.0824** |
| Std. 5 | 0.4961 | 0.4497 | 0.0897 | 0.0619 | 0.1810 | 0.1828 |
| Std. 10 | 0.3066 | 0.2928 | **0.0288** | 0.1733 | 0.4327 | 0.4325 |
| Std. 25 | 0.1071 | 0.1034 | 0.0700 | 0.3995 | 0.9390 | 0.9349 |
| Std. 50 | 0.0311 | 0.0312 | 0.0917 | 0.5574 | 1.2908 | 1.2878 |
| | | | OneFormer ConvNeXt-L | | | |
| None | **0.8287** | **0.6733** | 0.7887 | 0.8383 | 0.8012 | 0.8011 |
| Std. 5 | 0.8149 | 0.6639 | 0.7912 | 0.8365 | 0.8014 | 0.8014 |
| Std. 10 | 0.7979 | 0.6525 | 0.7874 | 0.8325 | 0.8013 | 0.8013 |
| Std. 25 | 0.7413 | 0.6053 | 0.7693 | 0.8176 | 0.8012 | 0.8009 |
| Std. 50 | 0.6202 | 0.5179 | **0.7505** | **0.7770** | **0.7996** | **0.7992** |

Table 23 shows the results of using the largest difference as a thresholding technique on noisy inputs and AUROC values. Table 24 shows the results of using a maximum fraction of pixels allowed as thresholding.

As seen with Dark Zurich, the largest thresholding technique becomes better as noise levels increase, with better Precision and Recall. In contrast, a maximum fraction of pixels thresholding works better at lower noise.

Table 23: Micro and macro-averaged ($\mu$ and M, respectively) Area Under Receiver Operating Characteristic (AUROC) and Precision, Recall and the fraction of pixels selected for largest difference thresholding on the Cityscapes dataset for noisy inputs.

| Noise Level | Uncertainty Metric | AUROC ↑ | | Largest Difference Thresholding | | | | | |
| | | | | Precision ↑ | | Recall ↑ | | Pixel % | |
| | | $\mu$ Avg. | M Avg. | $\mu$ Avg. | M Avg. | $\mu$ Avg. | M Avg. | $\mu$ Avg. | M Avg. |
|---|---|---|---|---|---|---|---|---|---|
| | | | | DRN D-22 | | | | | |
| None | Var. Ratio | 0.9118 | 0.9277 | 21.10 | 20.18 | 92.13 | 94.71 | 23.29 | 23.51 |
| | Prob. Margin | 0.9152 | 0.9302 | 19.71 | 18.84 | 93.31 | 95.62 | 25.26 | 25.48 |
| | Entropy | **0.9205** | **0.9337** | 17.52 | 16.74 | 95.04 | **96.89** | 28.94 | 29.21 |
| Std. 5 | Var. Ratio | 0.8413 | 0.8618 | 30.23 | 29.08 | 84.43 | 87.70 | 31.78 | 31.86 |
| | Prob. Margin | 0.8450 | 0.8650 | 28.71 | 27.64 | 86.27 | 89.29 | 34.19 | 34.27 |
| | Entropy | 0.8554 | 0.8732 | 26.34 | 25.36 | 89.41 | 91.86 | 38.63 | 38.74 |
| Std. 10 | Var. Ratio | 0.7718 | 0.7805 | 43.73 | 42.55 | 82.88 | 84.59 | 50.24 | 50.14 |
| | Prob. Margin | 0.7735 | 0.7816 | 42.39 | 41.23 | 85.17 | 86.68 | 53.25 | 53.15 |
| | Entropy | 0.7866 | 0.7934 | 40.22 | 39.09 | 89.29 | 90.40 | 58.84 | 58.75 |
| Std. 25 | Var. Ratio | 0.6966 | 0.6857 | 67.14 | 66.38 | 92.13 | 92.24 | 80.87 | 80.71 |
| | Prob. Margin | 0.6887 | 0.6776 | 66.38 | 65.63 | 93.47 | 93.55 | 82.98 | 82.83 |
| | Entropy | 0.7156 | 0.7041 | 65.11 | 64.43 | **96.37** | 96.45 | 87.23 | 87.10 |
| Std. 50 | Var. Ratio | 0.6294 | 0.6082 | **84.63** | **84.46** | 95.09 | 95.14 | 92.66 | 92.61 |
| | Prob. Margin | 0.6240 | 0.6027 | 84.40 | 84.22 | 95.88 | 95.92 | 93.68 | 93.64 |
| | Entropy | 0.6427 | 0.6213 | 83.67 | 84.06 | 88.34 | 89.15 | 87.06 | 87.31 |
| | | | | OneFormer ConvNeXt-L | | | | | |
| None | Var. Ratio | 0.7265 | 0.8822 | 15.02 | 16.52 | 84.68 | 87.75 | 18.03 | 18.17 |
| | Prob. Margin | 0.7893 | 0.9073 | 14.37 | 15.63 | 89.19 | 91.32 | 19.84 | 19.98 |
| | Entropy | 0.7089 | 0.8569 | 14.13 | 16.23 | 81.30 | 84.88 | 18.40 | 18.65 |
| Std. 5 | Var. Ratio | 0.7189 | 0.8828 | 15.66 | 17.20 | 85.86 | 88.09 | 18.59 | 18.74 |
| | Prob. Margin | 0.7846 | **0.9082** | 14.70 | 15.91 | 90.27 | **91.91** | 20.82 | 20.98 |
| | Entropy | 0.6975 | 0.8504 | 14.96 | 16.88 | 80.37 | 84.16 | 18.23 | 18.31 |
| Std. 10 | Var. Ratio | 0.7225 | 0.8737 | 15.70 | 17.35 | 85.73 | 87.67 | 20.51 | 20.69 |
| | Prob. Margin | **0.7902** | 0.9039 | 15.00 | 16.13 | **90.42** | 91.79 | 22.64 | 22.83 |
| | Entropy | 0.7008 | 0.8431 | 15.25 | 17.26 | 80.37 | 83.66 | 19.79 | 20.01 |
| Std. 25 | Var. Ratio | 0.7071 | 0.8506 | 16.76 | 18.45 | 82.89 | 85.96 | 25.13 | 25.36 |
| | Prob. Margin | 0.7771 | 0.8858 | 16.01 | 17.45 | 88.52 | 90.84 | 28.08 | 28.29 |
| | Entropy | 0.6810 | 0.8173 | 16.21 | 18.35 | 78.04 | 81.50 | 24.46 | 24.61 |
| Std. 50 | Var. Ratio | 0.6706 | 0.7929 | 19.69 | 21.88 | 75.11 | 79.44 | 33.35 | 33.54 |
| | Prob. Margin | 0.7432 | 0.8416 | **20.11** | 21.58 | 84.53 | 86.92 | 36.74 | 36.94 |
| | Entropy | 0.6375 | 0.7544 | 19.36 | **22.07** | 68.96 | 73.90 | 31.14 | 31.33 |

Table 24: Micro and macro-averaged ($\mu$ and M, respectively) Precision and Recall for thresholding based on a maximum fraction of pixels allowed on the Cityscapes dataset for noisy inputs.

| Noise Level | Unc. Metric | Max 5% pixels | | | | Max 10% pixels | | | | Max 15% pixels | | | |
|---|---|---|---|---|---|---|---|---|---|---|---|---|---|
| | | Precision ↑ | | Recall ↑ | | Precision ↑ | | Recall ↑ | | Precision ↑ | | Recall ↑ | |
| | | $\mu$ | M | $\mu$ | M | $\mu$ | M | $\mu$ | M | $\mu$ | M | $\mu$ | M |
| | | | | | | DRN D-22 | | | | | | | |
| None | VR | 45.19 | 45.40 | 41.52 | 50.74 | 35.04 | 35.18 | 64.13 | 73.92 | 28.48 | 28.45 | 76.44 | 84.64 |
| | PM | 44.66 | 44.83 | 41.35 | 50.58 | 34.77 | 34.92 | 64.13 | 73.95 | 28.14 | 28.19 | **76.51** | **84.74** |
| | E | 44.04 | 44.23 | 39.76 | 48.74 | 34.80 | 35.03 | 63.64 | 73.61 | 28.11 | 28.23 | 76.42 | 84.73 |
| Std. 5 | VR | 54.97 | 55.12 | 23.54 | 29.69 | 47.63 | 47.73 | 41.16 | 49.70 | 42.10 | 42.13 | 54.23 | 63.15 |
| | PM | 53.45 | 53.56 | 23.09 | 29.26 | 47.06 | 47.17 | 40.89 | 49.50 | 41.71 | 41.77 | 54.16 | 63.14 |
| | E | 55.09 | 55.13 | 23.23 | 28.93 | 48.08 | 48.25 | 40.97 | 49.43 | 42.23 | 42.35 | 54.47 | 63.39 |
| Std. 10 | VR | 67.13 | 67.11 | 12.20 | 14.46 | 62.14 | 62.10 | 22.95 | 26.78 | 58.77 | 58.69 | 32.73 | 37.61 |
| | PM | 63.41 | 63.41 | 11.63 | 13.83 | 60.68 | 60.64 | 22.57 | 26.38 | 57.95 | 57.88 | 32.44 | 37.32 |
| | E | 68.83 | 68.73 | 12.52 | 14.77 | 64.07 | 63.95 | 23.34 | 27.10 | 60.27 | 60.19 | 32.97 | 37.82 |
| Std. 25 | VR | 81.70 | 81.65 | 06.61 | 06.99 | 79.06 | 79.02 | 12.92 | 13.61 | 77.02 | 76.97 | 19.09 | 20.07 |
| | PM | 75.67 | 75.68 | 06.03 | 06.37 | 75.10 | 75.09 | 12.37 | 13.03 | 74.55 | 74.52 | 18.63 | 19.58 |
| | E | 83.90 | 83.86 | 06.77 | 07.16 | 82.12 | 82.05 | 13.40 | 14.13 | 80.52 | 80.45 | 19.86 | 20.87 |
| Std. 50 | VR | 88.49 | 88.46 | 05.06 | 05.11 | 88.09 | 88.06 | 10.24 | 10.33 | 87.80 | 87.76 | 15.46 | 15.58 |
| | PM | 86.81 | 86.76 | 04.82 | 04.85 | 86.71 | 86.65 | 10.07 | 10.14 | 86.63 | 86.56 | 15.35 | 15.46 |
| | E | **89.38** | **89.35** | 05.12 | 05.17 | 89.02 | 88.97 | 10.32 | 10.40 | 88.78 | 88.72 | 15.56 | 15.69 |
| | | | | | | OneFormer ConvNeXt-L | | | | | | | |
| None | VR | 32.16 | 32.31 | 47.73 | 57.40 | 24.77 | 24.84 | 67.92 | 77.28 | 20.80 | 21.03 | 74.47 | 82.96 |
| | PM | 33.41 | 33.78 | 50.18 | 60.30 | 24.95 | 24.99 | 71.11 | 80.37 | 20.58 | 20.68 | 79.04 | **86.67** |
| | E | 30.89 | 31.47 | 44.48 | 54.09 | 23.82 | 24.11 | 64.18 | 74.15 | 20.07 | 20.51 | 70.79 | 79.86 |
| Std. 5 | VR | 33.39 | 33.47 | 46.82 | 55.89 | 25.65 | 25.84 | 67.10 | 76.08 | 21.72 | 21.79 | 76.17 | 83.03 |
| | PM | 34.87 | 35.01 | 49.54 | 58.84 | 26.23 | 26.34 | 70.75 | 79.39 | 21.61 | 21.63 | **79.90** | 86.48 |
| | E | 32.06 | 32.15 | 43.67 | 52.72 | 24.64 | 24.94 | 63.04 | 72.49 | 20.95 | 21.27 | 71.34 | 79.12 |
| Std. 10 | VR | 34.62 | 34.73 | 44.15 | 52.27 | 26.98 | 27.09 | 65.58 | 73.98 | 22.53 | 22.66 | 74.42 | 81.57 |
| | PM | 36.58 | 36.73 | 47.19 | 55.44 | 27.76 | 27.82 | 69.51 | 77.30 | 22.83 | 22.83 | 79.17 | 85.34 |
| | E | 33.13 | 33.47 | 41.06 | 49.09 | 25.98 | 26.28 | 61.66 | 70.25 | 21.70 | 21.95 | 70.92 | 78.15 |
| Std. 25 | VR | 38.64 | 38.71 | 36.11 | 42.76 | 31.73 | 31.80 | 58.76 | 66.09 | 26.75 | 26.78 | 69.95 | 76.44 |
| | PM | 41.41 | 41.45 | 39.21 | 46.05 | 33.05 | 33.15 | 62.47 | 69.61 | 27.47 | 27.47 | 74.83 | 80.59 |
| | E | 36.77 | 36.65 | 32.98 | 39.34 | 30.38 | 30.56 | 54.93 | 62.39 | 25.65 | 25.85 | 65.64 | 72.39 |
| Std. 50 | VR | 47.30 | 47.37 | 25.49 | 31.07 | 41.07 | 41.14 | 44.32 | 51.89 | 35.91 | 35.85 | 56.27 | 63.68 |
| | PM | **49.59** | **49.81** | 27.03 | 32.95 | 43.02 | 42.94 | 47.25 | 54.86 | 37.62 | 37.51 | 61.20 | 68.13 |
| | E | 44.83 | 44.97 | 22.80 | 28.00 | 39.28 | 39.36 | 40.75 | 48.10 | 34.38 | 34.49 | 51.40 | 59.02 |

# E  Classwise Results

Finally, we provide classwise results for some of the experiments. Table 25 shows classwise Mean Intersection Over Union (mIoU) for all tested networks on the Cityscapes dataset, and Table 26 shows the same for ADE20K dataset. We observe that the best and worst classes in each dataset are consistent across models.

Table 25: Classwise micro and macro averaged ($\mu$ and M respectively) Mean Intersection Over Union (mIou) on the Cityscapes dataset across different models. Higher is better. The top five classes for every model are highlighted in bold green, and the worst five classes are highlighted in red italics.

| Class | DRN D-22 | DRN D-105 | OF ConvNeXt-L | OF Swin-L | Segformer B5 |
|---|---|---|---|---|---|

| | μ Avg. | M Avg. | μ Avg. | M Avg. | μ Avg. | M Avg. | μ Avg. | M Avg. | μ Avg. | M Avg. |
|---|---|---|---|---|---|---|---|---|---|---|
| road | **0.9721** | **0.9390** | **0.9815** | **0.9492** | **0.9858** | **0.9553** | **0.9847** | **0.9561** | **0.9847** | **0.9524** |
| sidewalk | 0.7966 | 0.6962 | 0.8498 | 0.7424 | 0.8775 | 0.7710 | 0.8673 | 0.7646 | 0.8735 | 0.7671 |
| building | **0.9019** | **0.8461** | **0.9249** | **0.8744** | **0.9371** | **0.8926** | **0.9401** | **0.8965** | **0.9375** | **0.8947** |
| wall | *0.3805* | *0.1526* | *0.4826* | *0.2291* | *0.5081* | *0.3053* | *0.6663* | *0.2973* | *0.6960* | *0.3196* |
| fence | *0.4654* | *0.1716* | *0.5877* | *0.2515* | *0.6937* | *0.2809* | *0.6951* | *0.2886* | *0.6754* | *0.3224* |
| pole | 0.5877 | 0.5258 | 0.6632 | 0.6062 | *0.7235* | 0.6665 | *0.7217* | 0.6683 | *0.6964* | 0.6454 |
| traffic light | 0.6301 | 0.3537 | 0.7289 | 0.4734 | 0.7678 | 0.5816 | 0.7644 | 0.5589 | 0.7547 | 0.5632 |
| traffic sign | 0.7300 | 0.6192 | 0.8053 | 0.7024 | 0.8445 | 0.7566 | 0.8517 | 0.7639 | 0.8231 | 0.7385 |
| vegetation | **0.9114** | **0.8696** | **0.9261** | **0.8914** | 0.9352 | **0.9058** | **0.9322** | **0.9006** | **0.9316** | **0.8999** |
| terrain | 0.5697 | 0.2557 | *0.6229* | 0.2790 | *0.6968* | *0.3332* | *0.6589* | *0.3072* | *0.6684* | *0.3351* |
| sky | **0.9372** | **0.8121** | **0.9511** | **0.8418** | **0.9573** | **0.8787** | **0.9588** | **0.8719** | **0.9567** | **0.8711** |
| person | 0.7741 | 0.5214 | 0.8316 | 0.6098 | 0.8743 | 0.6760 | 0.8692 | 0.6546 | 0.8494 | 0.6344 |
| rider | *0.5208* | 0.3254 | 0.6300 | 0.4560 | 0.7465 | 0.5362 | 0.7267 | 0.5055 | *0.6856* | 0.4777 |
| car | **0.9234** | **0.8230** | **0.9484** | **0.8794** | **0.9649** | **0.9029** | **0.9642** | **0.8985** | **0.9571** | **0.8971** |
| truck | *0.4067* | *0.1188* | *0.5999* | *0.2143* | 0.9029 | *0.3239* | 0.9004 | *0.3501* | 0.8632 | *0.3581* |
| bus | 0.6826 | 0.2365 | 0.8080 | 0.4289 | **0.9410** | 0.5502 | 0.9299 | 0.5614 | 0.9214 | 0.5964 |
| train | 0.5291 | *0.0765* | *0.5681* | *0.2004* | 0.8790 | 0.4827 | 0.8473 | 0.4778 | 0.8354 | 0.3840 |
| motorcycle | *0.4534* | *0.1361* | 0.6456 | *0.2572* | *0.7136* | *0.2762* | *0.6961* | *0.3006* | 0.7412 | *0.3570* |
| bicycle | 0.7287 | 0.4793 | 0.7918 | 0.5730 | 0.7953 | 0.6099 | 0.7717 | 0.5694 | 0.8022 | 0.5964 |

Table 26: Classwise micro and macro averaged (μ and M respectively) Mean Intersection Over Union (mIou) on the ADE20K dataset across different models. Higher is better. The top five classes for every model are highlighted in bold green, and the worst five classes are highlighted in red italics.

| Class | OF ConvNeXt-L | | OF Swin-L | | Segformer B5 | |
|---|---|---|---|---|---|---|
| | μ Avg. | M Avg. | μ Avg. | M Avg. | μ Avg. | M Avg. |
| wall | 0.8067 | 0.6516 | 0.8145 | 0.6618 | 0.7856 | 0.6381 |
| building | 0.8439 | 0.6402 | 0.8566 | 0.6479 | 0.8214 | 0.6160 |
| sky | **0.9483** | **0.8759** | **0.9506** | **0.8790** | **0.9453** | **0.8648** |
| floor | 0.8459 | 0.7415 | 0.8456 | 0.7434 | 0.8220 | 0.7262 |
| tree | 0.7759 | 0.6058 | 0.7753 | 0.6108 | 0.7477 | 0.5786 |
| ceiling | 0.8583 | 0.7567 | 0.8542 | 0.7483 | 0.8490 | **0.7379** |
| road | 0.8463 | 0.6934 | 0.8603 | 0.7142 | 0.8580 | 0.6668 |
| bed | **0.9170** | **0.8345** | **0.9106** | **0.8051** | **0.8991** | **0.7446** |
| window | 0.6355 | 0.5212 | 0.6638 | 0.5158 | 0.6120 | 0.4749 |
| grass | 0.7153 | 0.4215 | 0.7346 | 0.4164 | 0.7024 | 0.3966 |
| cabinet | 0.6241 | 0.3613 | 0.6379 | 0.4008 | 0.6327 | 0.3527 |
| sidewalk | 0.7021 | 0.5364 | 0.7059 | 0.5269 | 0.6682 | 0.4679 |
| person | 0.8623 | 0.6387 | 0.8622 | 0.6348 | 0.8258 | 0.5858 |
| ground | 0.3729 | 0.1986 | 0.4212 | 0.2239 | 0.3781 | 0.2129 |
| door | 0.5428 | 0.3631 | 0.5742 | 0.3691 | 0.4766 | 0.3215 |
| table | 0.6608 | 0.4943 | 0.6908 | 0.5068 | 0.6112 | 0.4095 |
| mountain | 0.6218 | 0.4125 | 0.6210 | 0.4709 | 0.6123 | 0.4474 |
| plant | 0.5788 | 0.3617 | 0.5737 | 0.3593 | 0.4870 | 0.3139 |
| curtain | 0.7856 | 0.6775 | 0.8282 | 0.7015 | 0.7346 | 0.5954 |
| chair | 0.6585 | 0.4655 | 0.6731 | 0.4784 | 0.5906 | 0.3688 |
| car | 0.8771 | 0.6539 | 0.8848 | 0.6554 | 0.8599 | 0.5979 |
| water | 0.5550 | 0.3294 | 0.5965 | 0.3596 | 0.5875 | 0.3113 |
| painting | 0.7703 | 0.6583 | 0.7710 | 0.6517 | 0.7460 | 0.5801 |
| sofa | 0.7240 | 0.5361 | 0.7717 | 0.6058 | 0.6761 | 0.4785 |

| | | | | | | |
|---|---|---|---|---|---|---|
| shelf | 0.4346 | 0.2814 | 0.5085 | 0.2834 | 0.4334 | 0.1944 |
| house | 0.4782 | 0.2273 | 0.4705 | 0.2188 | 0.3589 | 0.2050 |
| sea | 0.6542 | 0.6539 | 0.6563 | 0.6657 | 0.6163 | 0.6404 |
| mirror | 0.7409 | 0.4859 | 0.7459 | 0.5405 | 0.6822 | 0.4619 |
| rug | 0.6931 | 0.5074 | 0.6697 | 0.5201 | 0.5712 | 0.4658 |
| field | 0.3654 | 0.2385 | 0.3744 | 0.2600 | 0.3234 | 0.2377 |
| armchair | 0.4767 | 0.3774 | 0.5408 | 0.4324 | 0.4380 | 0.3323 |
| seat | 0.6794 | 0.4377 | 0.6271 | 0.3935 | 0.6062 | 0.2640 |
| fence | 0.4816 | 0.2270 | 0.4778 | 0.2090 | 0.4450 | 0.1653 |
| desk | 0.5720 | 0.3196 | 0.5498 | 0.3065 | 0.5174 | 0.2196 |
| rock | 0.6209 | 0.3134 | 0.6251 | 0.3511 | 0.4836 | 0.2421 |
| wardrobe | 0.4637 | 0.3900 | 0.5461 | 0.3966 | 0.4957 | 0.2762 |
| lamp | 0.7549 | 0.5837 | 0.7439 | 0.5656 | 0.6543 | 0.4512 |
| bathtub | 0.7890 | 0.6519 | 0.7881 | 0.7054 | 0.7722 | 0.5617 |
| railing | 0.4077 | 0.1737 | 0.4024 | 0.1923 | 0.3233 | 0.1239 |
| cushion | 0.7104 | 0.6211 | 0.7144 | 0.6059 | 0.5829 | 0.4591 |
| pedestal | 0.3440 | 0.1316 | 0.3931 | 0.1337 | 0.3005 | 0.1151 |
| box | 0.4007 | 0.2085 | 0.3680 | 0.1998 | 0.3042 | 0.1335 |
| column | 0.5654 | 0.2827 | 0.5452 | 0.2992 | 0.3971 | 0.1823 |
| sign | 0.4285 | 0.3156 | 0.4494 | 0.3097 | 0.3753 | 0.2077 |
| chest | 0.4499 | 0.3279 | 0.3611 | 0.3373 | 0.4488 | 0.2941 |
| counter | 0.5034 | 0.2014 | 0.4112 | 0.2114 | 0.2756 | 0.1485 |
| sand | 0.5074 | 0.4246 | 0.4304 | 0.4166 | 0.3661 | 0.2842 |
| sink | 0.7603 | 0.6281 | 0.8013 | 0.6238 | 0.7293 | 0.4948 |
| skyscraper | 0.3880 | 0.4413 | 0.4748 | 0.4636 | 0.6001 | 0.4912 |
| fireplace | 0.7373 | 0.5610 | 0.7466 | 0.5621 | 0.7959 | 0.5376 |
| refrigerator | 0.8125 | 0.6262 | 0.7967 | 0.6240 | 0.7766 | 0.5658 |
| grandstand | 0.5877 | 0.4089 | 0.4502 | 0.3187 | 0.4576 | 0.2347 |
| path | 0.2789 | 0.1794 | 0.2696 | 0.1930 | 0.2415 | 0.1313 |
| stairs | 0.3472 | 0.1767 | 0.3306 | 0.1939 | 0.3179 | 0.1396 |
| runway | 0.6795 | 0.5788 | 0.7157 | 0.6200 | 0.7075 | 0.4809 |
| case | 0.6180 | 0.2870 | 0.5921 | 0.3194 | 0.4926 | 0.1707 |
| pool table | **0.9529** | 0.7290 | **0.9521** | **0.8312** | **0.9378** | **0.7310** |
| pillow | 0.6874 | 0.5569 | 0.6626 | 0.5333 | 0.5727 | 0.4080 |
| screen door | 0.6210 | 0.6015 | 0.7174 | 0.7501 | 0.6945 | 0.4878 |
| stairway | 0.3473 | 0.2017 | 0.4628 | 0.1861 | 0.2846 | 0.1354 |
| river | 0.2326 | 0.2446 | 0.1674 | 0.1943 | 0.1530 | 0.1384 |
| bridge | 0.7893 | 0.3932 | 0.8101 | 0.4119 | 0.6948 | 0.2788 |
| bookcase | 0.3632 | 0.3125 | 0.3668 | 0.3179 | 0.3821 | 0.2812 |
| blind | 0.4365 | 0.2617 | 0.5201 | 0.2866 | 0.4640 | 0.2325 |
| coffee table | 0.6210 | 0.4195 | 0.6698 | 0.4621 | 0.5341 | 0.3168 |
| toilet | 0.9087 | **0.8842** | 0.8984 | **0.8076** | 0.8761 | **0.7845** |
| flower | 0.5273 | 0.3844 | 0.5180 | 0.3990 | 0.4448 | 0.3256 |
| book | 0.5470 | 0.2751 | 0.5712 | 0.2600 | 0.4917 | 0.1820 |
| hill | *0.1413* | *0.0699* | *0.0882* | *0.0782* | *0.0699* | *0.0596* |
| bench | 0.7182 | 0.2913 | 0.4896 | 0.2537 | 0.4612 | 0.1472 |
| countertop | 0.6699 | 0.3158 | 0.6848 | 0.3225 | 0.5890 | 0.3261 |
| stove | 0.8335 | 0.6621 | 0.8505 | 0.7050 | 0.7762 | 0.5391 |
| palm | 0.5766 | 0.4356 | 0.5272 | 0.4224 | 0.4802 | 0.3737 |
| kitchen island | 0.4398 | 0.1622 | 0.3594 | 0.1433 | 0.4026 | 0.2282 |
| computer | 0.7565 | 0.4210 | 0.7788 | 0.4522 | 0.7503 | 0.3573 |
| swivel chair | 0.6038 | 0.4412 | 0.5583 | 0.3940 | 0.3974 | 0.3466 |
| boat | 0.4278 | 0.3011 | 0.6905 | 0.3579 | 0.5051 | 0.2265 |
| bar | 0.6650 | 0.3924 | 0.5365 | 0.2896 | 0.3440 | 0.2020 |

| | | | | | | |
|---|---|---|---|---|---|---|
| arcade machine | 0.8454 | 0.2722 | 0.6547 | 0.2077 | 0.8334 | 0.2863 |
| hut | 0.4318 | 0.2134 | 0.3258 | 0.2070 | 0.3131 | 0.1490 |
| bus | **0.9460** | 0.3583 | **0.9115** | 0.2936 | **0.9068** | 0.2478 |
| towel | 0.7651 | 0.5525 | 0.7642 | 0.5077 | 0.6295 | 0.3371 |
| light | 0.6364 | 0.4461 | 0.6482 | 0.4464 | 0.5536 | 0.3497 |
| truck | 0.4274 | 0.1781 | 0.4698 | 0.2005 | 0.4223 | 0.1556 |
| tower | 0.2338 | 0.1999 | 0.3230 | 0.2109 | 0.1143 | 0.1274 |
| chandelier | 0.7500 | 0.5696 | 0.7364 | 0.5751 | 0.6495 | 0.4677 |
| awning | 0.4043 | 0.2681 | 0.3715 | 0.2658 | 0.2873 | 0.1627 |
| streetlight | 0.4746 | 0.3086 | 0.4414 | 0.3132 | 0.2739 | 0.1378 |
| booth | 0.5275 | 0.2094 | 0.6864 | 0.2372 | 0.3458 | 0.2258 |
| television | 0.7962 | 0.6454 | 0.7413 | 0.6022 | 0.7161 | 0.4806 |
| airplane | 0.6213 | 0.4914 | 0.6869 | 0.4720 | 0.6163 | 0.3258 |
| dirt track | *0.0322* | *0.0492* | *0.0306* | *0.0237* | *0.0490* | *0.0274* |
| clothes | 0.3764 | 0.1749 | 0.4861 | 0.2240 | 0.3336 | 0.1717 |
| pole | 0.3596 | 0.1927 | 0.3444 | 0.1895 | 0.2177 | 0.1025 |
| land | *0.0627* | *0.0412* | *0.0747* | 0.0999 | *0.0058* | *0.0056* |
| bannister | 0.2125 | 0.1589 | 0.2319 | 0.1424 | 0.1263 | 0.0660 |
| escalator | 0.2659 | 0.3568 | 0.5813 | 0.3277 | 0.5315 | 0.2616 |
| ottoman | 0.4856 | 0.2862 | 0.5834 | 0.3402 | 0.5115 | 0.1919 |
| bottle | 0.4579 | 0.2554 | 0.4654 | 0.2594 | 0.3638 | 0.1409 |
| buffet | 0.4452 | 0.2182 | 0.4719 | 0.2150 | 0.3274 | 0.1564 |
| poster | 0.3500 | 0.1344 | 0.2816 | 0.1522 | 0.2479 | 0.0986 |
| stage | 0.1463 | 0.1667 | 0.1337 | 0.2894 | 0.1580 | 0.1140 |
| van | 0.4994 | 0.1954 | 0.5293 | 0.2069 | 0.4345 | 0.1864 |
| ship | 0.6307 | 0.3798 | 0.8245 | 0.4695 | 0.7014 | 0.2476 |
| fountain | 0.3263 | 0.2975 | 0.4673 | 0.4499 | 0.2149 | 0.1999 |
| conveyer belt | 0.7296 | 0.3313 | 0.6991 | 0.2257 | 0.7720 | 0.2220 |
| canopy | 0.4703 | 0.2735 | 0.3512 | 0.2969 | 0.4163 | 0.2003 |
| washing machine | 0.7302 | 0.7117 | 0.8698 | 0.7630 | 0.7532 | 0.6962 |
| toy | 0.4918 | 0.2208 | 0.3575 | 0.2384 | 0.2614 | 0.1435 |
| swimming pool | 0.8168 | 0.6303 | 0.7866 | 0.4143 | 0.5739 | 0.3583 |
| stool | 0.5461 | 0.2621 | 0.5905 | 0.2739 | 0.3919 | 0.1236 |
| barrel | 0.8925 | **0.8798** | 0.6715 | 0.5162 | 0.4894 | 0.3102 |
| basket | 0.3604 | 0.2936 | 0.4157 | 0.3148 | 0.3615 | 0.2327 |
| waterfall | 0.7514 | 0.5450 | 0.5517 | 0.5178 | 0.6443 | 0.3568 |
| tent | **0.9594** | 0.4856 | **0.9292** | 0.5659 | **0.9487** | 0.3551 |
| bag | 0.2520 | 0.1483 | 0.2170 | 0.1410 | 0.1267 | *0.0616* |
| motorbike | 0.7769 | 0.3167 | 0.7472 | 0.3723 | 0.7456 | 0.3757 |
| cradle | 0.8654 | **0.7638** | 0.8667 | **0.8741** | 0.7678 | 0.7065 |
| oven | 0.5860 | 0.2807 | 0.5741 | 0.3720 | 0.4922 | 0.2726 |
| ball | 0.4123 | 0.2837 | 0.5238 | 0.2504 | 0.5260 | 0.1281 |
| food | 0.6462 | 0.2817 | 0.6334 | 0.3565 | 0.3681 | 0.2434 |
| step | 0.1872 | 0.0850 | 0.0899 | *0.0758* | 0.1861 | 0.0869 |
| tank | 0.5648 | 0.1351 | 0.5138 | 0.1449 | 0.5763 | 0.1914 |
| trade name | 0.3477 | 0.2577 | 0.3493 | 0.2611 | 0.2678 | 0.1511 |
| microwave | 0.8695 | 0.6115 | 0.8657 | 0.6096 | 0.7648 | 0.5329 |
| pot | 0.5798 | 0.3344 | 0.5933 | 0.3199 | 0.4460 | 0.2539 |
| animal | 0.6246 | 0.4527 | 0.6330 | 0.5182 | 0.5985 | 0.4072 |
| bicycle | 0.6379 | 0.3518 | 0.6383 | 0.3425 | 0.5816 | 0.2529 |
| lake | *0.0000* | *0.0000* | 0.4752 | *0.0994* | 0.6027 | 0.1241 |
| dishwasher | 0.7900 | 0.5149 | 0.7234 | 0.4633 | 0.6610 | 0.4105 |
| screen | 0.6464 | 0.4130 | 0.6395 | 0.4824 | 0.7059 | 0.3711 |
| blanket | 0.3817 | 0.3033 | 0.3442 | 0.3109 | 0.1784 | 0.1382 |

| | | | | | | |
|---|---|---|---|---|---|---|
| sculpture | 0.6732 | 0.1532 | 0.6341 | 0.1354 | 0.6072 | 0.0820 |
| hood | 0.6624 | 0.6031 | 0.6562 | 0.6131 | 0.5749 | 0.5388 |
| sconce | 0.6015 | 0.3542 | 0.6167 | 0.3599 | 0.4927 | 0.2539 |
| vase | 0.5359 | 0.3630 | 0.5246 | 0.3155 | 0.4137 | 0.2371 |
| traffic light | 0.4124 | 0.3142 | 0.4778 | 0.2960 | 0.3619 | 0.1691 |
| tray | 0.1839 | 0.1243 | 0.2635 | 0.1621 | 0.1173 | 0.0869 |
| ashcan | 0.4290 | 0.2278 | 0.4943 | 0.2422 | 0.4235 | 0.1743 |
| fan | 0.7102 | 0.4734 | 0.7327 | 0.5408 | 0.6441 | 0.3386 |
| pier | 0.3749 | 0.1524 | 0.3767 | 0.2290 | 0.6224 | 0.1378 |
| crt screen | 0.2180 | *0.0496* | *0.0694* | *0.0374* | 0.1122 | *0.0338* |
| plate | 0.6085 | 0.2960 | 0.6328 | 0.2942 | 0.5125 | 0.1629 |
| monitor | 0.6631 | 0.2648 | 0.1195 | 0.1170 | *0.0759* | 0.0717 |
| bulletin board | 0.6488 | 0.2437 | 0.5838 | 0.1777 | 0.4835 | 0.1756 |
| shower | *0.0315* | 0.1646 | *0.0358* | 0.2362 | *0.0665* | 0.1209 |
| radiator | 0.5876 | 0.3435 | 0.6938 | 0.3621 | 0.6611 | 0.4215 |
| glass | 0.2507 | 0.1983 | 0.2497 | 0.1725 | 0.1732 | 0.1312 |
| clock | 0.3627 | 0.3053 | 0.5175 | 0.3060 | 0.4064 | 0.2080 |
| flag | 0.7495 | 0.4274 | 0.5406 | 0.3913 | 0.5256 | 0.3048 |

Table 27 and Table 28 show the Area Under Receiver Operating Characteristic (AUROC) on Cityscapes and ADE20K datasets across a few scenarios. All of the results use entropy as the uncertainty metric. Here also we observe that the best and worst performing classes are generally consistent across different scenarios, especially for Cityscapes.

Table 27: Classwise micro and macro averaged ($\mu$ and M respectively) Area Under Receiver Operating Characteristic (AUROC) on the Cityscapes dataset across a couple different models and settings. Higher is better. The top five classes for every model are highlighted in bold green, and the worst five classes are highlighted in red italics.

| Class | DRN D-105 Base | | DRN D-105 Scale | | Segformer B5 Base | | Segformer B5 Scale | |
|---|---|---|---|---|---|---|---|---|
| | $\mu$ Avg. | M Avg. | $\mu$ Avg. | M Avg. | $\mu$ Avg. | M Avg. | $\mu$ Avg. | M Avg. |
| road | **0.9241** | **0.9783** | **0.9615** | **0.9861** | 0.9118 | **0.9693** | 0.9335 | **0.9801** |
| sidewalk | 0.8606 | 0.8756 | 0.8996 | 0.8742 | 0.8642 | 0.8826 | 0.8962 | 0.8795 |
| building | **0.9235** | **0.9332** | 0.9349 | **0.9346** | **0.9148** | **0.9293** | 0.9280 | **0.9353** |
| wall | *0.6513* | *0.4907* | *0.6980* | *0.4717* | *0.7152* | *0.5491* | *0.7167* | *0.5132* |
| fence | *0.6946* | 0.5896 | *0.7400* | 0.5685 | *0.7387* | *0.6148* | 0.8026 | *0.5933* |
| pole | *0.7529* | 0.7270 | 0.7550 | 0.7205 | *0.7565* | 0.7447 | *0.7680* | 0.7438 |
| traffic light | 0.8339 | 0.7137 | 0.8336 | 0.6879 | 0.8376 | 0.7542 | 0.8491 | 0.7127 |
| traffic sign | 0.8718 | 0.8388 | 0.8773 | 0.8174 | 0.8523 | 0.8506 | 0.8775 | 0.8292 |
| vegetation | **0.9362** | **0.9366** | **0.9534** | **0.9453** | **0.9286** | **0.9341** | **0.9511** | **0.9484** |
| terrain | *0.7133* | *0.5592* | *0.7487* | *0.5510* | *0.7117* | *0.5568* | *0.7751* | *0.5526* |
| sky | **0.9681** | **0.9226** | **0.9708** | **0.9169** | **0.9660** | **0.9223** | **0.9628** | **0.9064** |
| person | 0.9078 | 0.7937 | 0.9152 | 0.7751 | 0.8952 | 0.7961 | 0.9134 | 0.7807 |
| rider | *0.7214* | 0.6861 | *0.7483* | 0.6808 | *0.8223* | 0.7011 | *0.8396* | 0.6993 |
| car | **0.9672** | **0.9422** | **0.9727** | **0.9428** | **0.9464** | **0.9451** | **0.9654** | **0.9495** |
| truck | 0.7554 | *0.4891* | 0.8498 | *0.4899* | 0.8737 | *0.5390* | **0.9379** | *0.5555* |
| bus | 0.9031 | 0.7551 | **0.9349** | 0.7501 | **0.9203** | 0.7927 | **0.9631** | 0.7892 |
| train | 0.7721 | *0.5658* | *0.7414* | *0.5000* | 0.8993 | 0.6744 | 0.9125 | 0.6772 |
| motorcycle | 0.8312 | *0.5072* | 0.8179 | *0.5009* | 0.8516 | *0.5942* | 0.8607 | *0.5732* |
| bicycle | 0.8795 | 0.7646 | 0.8984 | 0.7691 | 0.8828 | 0.7628 | 0.9021 | 0.7596 |

Table 28: Classwise micro and macro averaged ($\mu$ and M respectively) Area Under Receiver Operating Characteristic (AUROC) on the ADE20K dataset across a couple different models and settings. Higher is better. The top five classes for every model are highlighted in bold green, and the worst five classes are highlighted in red italics.

| Class | OF ConvNeXt-L Base | | OF ConvNeXt-L Scale | | Segformer B5 Base | | Segformer B5 Scale | |
|---|---|---|---|---|---|---|---|---|
| | $\mu$ Avg. | M Avg. | $\mu$ Avg. | M Avg. | $\mu$ Avg. | M Avg. | $\mu$ Avg. | M Avg. |
| wall | 0.5739 | 0.6680 | 0.5838 | 0.6911 | 0.8406 | 0.8069 | 0.8743 | 0.8182 |
| building | 0.5608 | 0.6452 | 0.5531 | 0.6701 | 0.8300 | 0.8045 | 0.8605 | 0.8039 |
| sky | 0.5670 | 0.7676 | 0.5786 | 0.7708 | 0.9233 | **0.9184** | 0.9339 | **0.9159** |
| floor | 0.5547 | 0.7259 | 0.5663 | 0.7346 | 0.8556 | 0.8752 | 0.8674 | 0.8720 |
| tree | 0.6203 | 0.7113 | 0.6428 | 0.7331 | 0.8665 | 0.7693 | 0.8745 | 0.7638 |
| ceiling | 0.5189 | 0.7149 | 0.5099 | 0.7141 | 0.8375 | 0.8589 | 0.8665 | 0.8536 |
| road | 0.4516 | 0.7259 | 0.4608 | 0.7366 | 0.8314 | 0.8193 | 0.8476 | 0.8192 |
| bed | 0.7180 | 0.7278 | 0.6719 | 0.7245 | **0.9456** | **0.9118** | **0.9646** | **0.9123** |
| window | 0.5302 | 0.6360 | 0.5263 | 0.6552 | 0.7547 | 0.6916 | 0.7667 | 0.6798 |
| grass | 0.5200 | 0.6441 | 0.5906 | 0.6610 | 0.8606 | 0.7107 | 0.8761 | 0.7139 |
| cabinet | 0.6989 | 0.5732 | 0.7458 | 0.5806 | 0.7957 | 0.5975 | 0.8219 | 0.5716 |
| sidewalk | 0.5447 | 0.7314 | 0.5405 | 0.7414 | 0.7642 | 0.7200 | 0.8085 | 0.6995 |
| person | 0.6048 | 0.7202 | 0.6161 | 0.7258 | 0.9108 | 0.7334 | 0.9062 | 0.7070 |
| ground | 0.5967 | 0.4664 | 0.6169 | 0.4740 | 0.6065 | 0.4299 | 0.6027 | 0.4227 |
| door | 0.6446 | 0.6255 | 0.7004 | 0.6337 | 0.6358 | 0.5255 | 0.6586 | 0.4951 |
| table | 0.5518 | 0.6847 | 0.5349 | 0.6946 | 0.7932 | 0.6604 | 0.8172 | 0.6386 |
| mountain | 0.7045 | 0.6829 | 0.6920 | 0.7049 | 0.8066 | 0.7067 | 0.8241 | 0.6949 |
| plant | 0.5941 | 0.5722 | 0.5973 | 0.5666 | 0.7097 | 0.4936 | 0.7293 | 0.4755 |
| curtain | 0.5007 | 0.7722 | 0.4830 | 0.7625 | 0.8088 | 0.8119 | 0.8318 | 0.8035 |
| chair | 0.5607 | 0.6705 | 0.5258 | 0.6773 | 0.6841 | 0.6138 | 0.7018 | 0.5877 |
| car | 0.6087 | 0.8017 | 0.6218 | 0.8141 | 0.9002 | 0.7805 | 0.9236 | 0.7674 |
| water | 0.5340 | 0.4311 | 0.5104 | 0.4271 | 0.7816 | 0.5628 | 0.8001 | 0.5702 |
| painting | 0.5304 | 0.7683 | 0.5556 | 0.7635 | 0.8534 | 0.7571 | 0.8708 | 0.7384 |
| sofa | 0.6279 | 0.6968 | 0.6681 | 0.7309 | 0.8479 | 0.7624 | 0.8404 | 0.7457 |
| shelf | 0.6358 | 0.4825 | 0.6114 | 0.4893 | 0.7001 | 0.4210 | 0.7571 | 0.4163 |
| house | 0.4963 | 0.4237 | 0.5240 | 0.4215 | 0.5942 | 0.3930 | 0.5882 | 0.3653 |
| sea | 0.8416 | 0.7356 | **0.8649** | 0.7661 | 0.5105 | 0.7815 | 0.6722 | 0.7845 |
| mirror | 0.5035 | 0.6449 | 0.4964 | 0.6652 | 0.7857 | 0.6698 | 0.7806 | 0.6288 |
| rug | 0.4027 | 0.5136 | 0.4490 | 0.5264 | 0.6194 | 0.6740 | 0.6000 | 0.6428 |
| field | 0.6289 | 0.4627 | 0.6797 | 0.4573 | 0.7246 | 0.4810 | 0.7626 | 0.4843 |
| armchair | 0.6309 | 0.6297 | 0.6719 | 0.6332 | 0.5639 | 0.5896 | 0.6109 | 0.5765 |
| seat | 0.5390 | 0.5249 | 0.6303 | 0.5425 | 0.8414 | 0.5146 | 0.8518 | 0.5117 |
| fence | 0.6256 | 0.4773 | 0.6257 | 0.4841 | 0.7178 | 0.3680 | 0.7237 | 0.3705 |
| desk | 0.5602 | 0.6556 | 0.6061 | 0.6902 | 0.7862 | 0.6218 | 0.8023 | 0.6075 |
| rock | 0.6126 | 0.4923 | 0.6155 | 0.5125 | 0.7262 | 0.5525 | 0.7262 | 0.5262 |
| wardrobe | 0.6437 | 0.5709 | 0.7209 | 0.5765 | 0.7010 | 0.5167 | 0.7658 | 0.5172 |
| lamp | 0.7204 | 0.7042 | 0.7100 | 0.7101 | 0.8199 | 0.6966 | 0.8165 | 0.6669 |
| bathtub | 0.6990 | 0.7776 | 0.6818 | **0.8174** | 0.8999 | 0.8481 | 0.8828 | 0.8086 |
| railing | 0.6131 | 0.4368 | 0.6146 | 0.4408 | 0.6652 | 0.3275 | 0.6494 | 0.3161 |
| cushion | 0.4820 | 0.6654 | 0.4748 | 0.6584 | 0.6816 | 0.7281 | 0.7192 | 0.7051 |
| pedestal | 0.6809 | 0.4428 | 0.6873 | 0.4416 | 0.5481 | 0.3212 | 0.5147 | 0.3059 |
| box | 0.6084 | 0.3712 | 0.5978 | 0.3630 | 0.6921 | 0.2932 | 0.6581 | 0.2552 |
| column | *0.3089* | 0.4293 | 0.4210 | 0.4016 | 0.6474 | 0.4266 | 0.5982 | 0.3984 |
| sign | 0.5225 | 0.6147 | 0.5353 | 0.5977 | 0.6341 | 0.3961 | 0.5934 | 0.3425 |
| chest | 0.4561 | 0.5929 | 0.4538 | 0.6150 | 0.5828 | 0.5883 | 0.6105 | 0.5485 |
| counter | 0.5578 | 0.3881 | 0.6440 | 0.3824 | 0.6070 | 0.3743 | 0.6560 | 0.3476 |

| | | | | | | | | |
|---|---|---|---|---|---|---|---|---|
| sand | 0.7243 | 0.5635 | 0.7637 | 0.5650 | 0.7968 | 0.5653 | 0.8045 | 0.5613 |
| sink | 0.5907 | 0.7345 | 0.5826 | 0.7294 | 0.8554 | 0.6905 | 0.8545 | 0.6610 |
| skyscraper | 0.4182 | 0.6289 | 0.3830 | 0.5981 | 0.6903 | 0.7200 | 0.7077 | 0.7068 |
| fireplace | 0.5034 | 0.7696 | 0.6717 | **0.8441** | 0.9120 | 0.8033 | 0.9060 | 0.7539 |
| refrigerator | 0.8346 | 0.7137 | 0.8122 | 0.7293 | 0.8574 | 0.7904 | 0.8890 | 0.7767 |
| grandstand | **0.8782** | 0.5326 | 0.7779 | 0.4291 | 0.8273 | 0.5412 | 0.8325 | 0.4975 |
| path | 0.4835 | 0.3132 | 0.4781 | 0.3176 | 0.5924 | 0.3287 | 0.5574 | 0.3038 |
| stairs | 0.5280 | 0.3520 | 0.5338 | 0.3616 | 0.7095 | 0.3098 | 0.5954 | 0.2824 |
| runway | *0.2676* | 0.4812 | *0.3359* | 0.4756 | **0.9490** | 0.6891 | **0.9581** | 0.6515 |
| case | 0.6746 | 0.5060 | 0.7389 | 0.5134 | 0.8008 | 0.6118 | 0.8432 | 0.6066 |
| pool table | 0.7485 | **0.8723** | 0.7391 | **0.8845** | 0.9305 | 0.8488 | **0.9528** | 0.8458 |
| pillow | 0.5753 | 0.7081 | 0.5782 | 0.7042 | 0.7406 | 0.6275 | 0.7278 | 0.5904 |
| screen door | 0.3675 | 0.6923 | *0.3557* | 0.6841 | 0.6830 | 0.6751 | 0.6335 | 0.6286 |
| stairway | 0.3363 | 0.3674 | 0.3572 | 0.3712 | 0.7469 | 0.2621 | 0.7850 | 0.2590 |
| river | 0.4717 | 0.3615 | 0.4526 | 0.3756 | 0.3881 | 0.3104 | 0.3797 | 0.2734 |
| bridge | 0.6281 | 0.6242 | 0.7111 | 0.6332 | 0.7987 | 0.6093 | 0.8159 | 0.6052 |
| bookcase | 0.5989 | 0.6119 | 0.5601 | 0.5919 | 0.6138 | 0.5141 | 0.6564 | 0.5434 |
| blind | 0.7091 | 0.3400 | 0.3707 | 0.3050 | 0.5208 | 0.3672 | 0.4756 | 0.3452 |
| coffee table | 0.5752 | 0.6190 | 0.5177 | 0.6519 | 0.8910 | 0.6605 | 0.8744 | 0.6495 |
| toilet | 0.7409 | 0.7570 | 0.7696 | 0.7848 | 0.9246 | **0.9070** | 0.9500 | **0.8863** |
| flower | 0.6332 | 0.5608 | 0.6670 | 0.5665 | 0.6797 | 0.5289 | 0.6650 | 0.4774 |
| book | 0.4242 | 0.5053 | 0.5302 | 0.5146 | 0.7241 | 0.4159 | 0.7423 | 0.3906 |
| hill | 0.6818 | 0.2886 | **0.8445** | 0.3153 | 0.3824 | 0.1686 | 0.4030 | 0.1693 |
| bench | 0.6265 | 0.5043 | 0.6425 | 0.4843 | 0.7980 | 0.3212 | 0.7693 | 0.2662 |
| countertop | 0.3637 | 0.5977 | 0.4136 | 0.6233 | 0.7284 | 0.7519 | 0.7343 | 0.7309 |
| stove | 0.4978 | 0.7285 | 0.6257 | 0.7418 | 0.9168 | 0.7212 | 0.9310 | 0.6986 |
| palm | 0.6845 | 0.6145 | 0.6536 | 0.6336 | 0.6720 | 0.5673 | 0.7011 | 0.5387 |
| kitchen island | 0.5199 | 0.4266 | 0.6521 | 0.4332 | 0.7225 | 0.6503 | 0.7809 | 0.6426 |
| computer | 0.8235 | 0.5815 | 0.8232 | 0.5919 | 0.9081 | 0.7305 | 0.8766 | 0.7051 |
| swivel chair | 0.5784 | 0.6860 | 0.6574 | 0.7054 | 0.6079 | 0.5702 | 0.6943 | 0.5242 |
| boat | **0.8885** | 0.5479 | **0.9060** | 0.5646 | 0.7252 | 0.4369 | 0.7538 | 0.4301 |
| bar | 0.6554 | 0.4605 | 0.6270 | 0.4680 | 0.7454 | 0.3915 | 0.7891 | 0.3932 |
| arcade machine | 0.4072 | 0.5503 | 0.5758 | 0.6500 | 0.8680 | 0.7148 | 0.6931 | 0.6238 |
| hut | 0.5054 | 0.4863 | 0.4311 | 0.4959 | 0.6138 | 0.3689 | 0.6393 | 0.3997 |
| bus | 0.3467 | 0.4428 | 0.3641 | 0.4371 | **0.9765** | 0.3753 | **0.9795** | 0.3688 |
| towel | 0.7283 | 0.6239 | 0.7306 | 0.6176 | 0.8701 | 0.6178 | 0.8842 | 0.5967 |
| light | 0.6130 | 0.7420 | 0.6070 | 0.7311 | 0.6760 | 0.4859 | 0.6491 | 0.4352 |
| truck | 0.5780 | 0.4361 | 0.7299 | 0.4487 | 0.7291 | 0.2699 | 0.5865 | 0.2357 |
| tower | 0.6639 | 0.2167 | 0.7170 | 0.2161 | 0.3517 | 0.3008 | 0.3304 | 0.2651 |
| chandelier | 0.6425 | 0.7446 | 0.6891 | 0.7516 | 0.7727 | 0.7477 | 0.7470 | 0.7108 |
| awning | 0.7975 | 0.4976 | 0.7861 | 0.4995 | 0.4719 | 0.3072 | 0.4494 | 0.2566 |
| streetlight | 0.6850 | 0.6345 | 0.6393 | 0.6176 | 0.6430 | 0.2739 | 0.6165 | 0.2448 |
| booth | 0.3981 | 0.2835 | 0.5152 | 0.2781 | 0.6733 | 0.4017 | 0.8289 | 0.4596 |
| television | 0.4803 | 0.7019 | 0.5252 | 0.6495 | 0.9151 | 0.7664 | 0.9109 | 0.7065 |
| airplane | 0.8537 | 0.6929 | 0.8063 | 0.6951 | 0.5502 | 0.6165 | 0.5442 | 0.5876 |
| dirt track | 0.4029 | *0.0504* | 0.4449 | *0.0566* | *0.2451* | *0.1159* | *0.2267* | *0.1355* |
| clothes | 0.8325 | 0.4714 | 0.7844 | 0.3999 | 0.8503 | 0.4640 | 0.8761 | 0.4604 |
| pole | 0.5662 | 0.5002 | 0.6806 | 0.5169 | 0.5469 | 0.2786 | 0.5646 | 0.2776 |
| land | 0.4633 | *0.1282* | 0.5031 | *0.1432* | *0.1227* | *0.0465* | *0.1449* | *0.0565* |
| bannister | 0.3417 | 0.3438 | 0.3737 | 0.3525 | 0.4584 | 0.1728 | 0.3413 | 0.1535 |
| escalator | 0.6141 | 0.4464 | 0.6299 | 0.3113 | 0.7641 | 0.5747 | 0.6754 | 0.5215 |
| ottoman | 0.6320 | 0.4822 | 0.6135 | 0.4618 | 0.8452 | 0.4835 | 0.8102 | 0.4721 |
| bottle | 0.6183 | 0.4295 | 0.5480 | 0.4040 | 0.6643 | 0.2820 | 0.5900 | 0.2440 |
| buffet | 0.5471 | 0.3650 | *0.3454* | 0.3767 | 0.6931 | 0.2672 | 0.7233 | 0.2774 |

| | | | | | | | | |
|---|---|---|---|---|---|---|---|---|
| poster | 0.7482 | 0.3111 | 0.8322 | 0.2891 | 0.6782 | 0.2700 | 0.6315 | 0.2369 |
| stage | 0.7343 | 0.4193 | 0.7956 | 0.4082 | 0.4942 | 0.4548 | 0.4788 | 0.4133 |
| van | 0.4960 | 0.5453 | 0.5783 | 0.5172 | 0.7748 | 0.3088 | 0.7363 | 0.2792 |
| ship | 0.7172 | 0.6277 | 0.7486 | 0.6264 | **0.9629** | 0.5954 | 0.9360 | 0.5856 |
| fountain | *0.2365* | 0.6323 | 0.3632 | 0.6578 | 0.7300 | 0.4035 | 0.6050 | 0.3776 |
| conveyer belt | 0.7016 | 0.7340 | 0.7865 | 0.7000 | 0.8565 | 0.6961 | 0.7608 | 0.6925 |
| canopy | 0.6832 | 0.5018 | 0.7821 | 0.6180 | 0.7657 | 0.5034 | 0.7937 | 0.4958 |
| washing machine | **0.9583** | **0.9192** | **0.9575** | **0.9109** | 0.9189 | 0.7947 | 0.9414 | 0.8013 |
| toy | 0.4970 | 0.4003 | 0.5326 | 0.4043 | 0.5191 | 0.4129 | 0.4756 | 0.3660 |
| swimming pool | 0.8311 | **0.8127** | 0.8263 | 0.7989 | 0.8580 | 0.8935 | 0.8550 | 0.8604 |
| stool | 0.6112 | 0.4539 | 0.6849 | 0.4576 | 0.7468 | 0.3398 | 0.7656 | 0.3160 |
| barrel | **0.8543** | **0.8331** | 0.8104 | 0.7544 | 0.9399 | 0.7760 | 0.9281 | 0.7556 |
| basket | 0.6128 | 0.5168 | 0.5646 | 0.5100 | 0.7601 | 0.4798 | 0.6847 | 0.3967 |
| waterfall | 0.6714 | 0.6821 | 0.7421 | 0.7442 | 0.8017 | 0.7390 | 0.8338 | 0.7768 |
| tent | 0.6744 | **0.8092** | 0.8069 | **0.8461** | **0.9881** | **0.9836** | **0.9890** | **0.9885** |
| bag | 0.5270 | 0.3574 | 0.5706 | 0.3445 | 0.5483 | *0.1569* | 0.5100 | *0.1343* |
| motorbike | 0.7851 | 0.6201 | 0.7966 | 0.6435 | 0.9041 | 0.5442 | 0.8624 | 0.4961 |
| cradle | 0.4388 | 0.7722 | 0.5862 | 0.8150 | 0.9401 | **0.9206** | 0.9374 | **0.9253** |
| oven | 0.6352 | 0.4992 | 0.7394 | 0.4804 | 0.6788 | 0.4870 | 0.6820 | 0.4598 |
| ball | 0.7694 | 0.4742 | 0.7446 | 0.4316 | 0.7972 | 0.2225 | 0.8621 | 0.2144 |
| food | 0.6721 | 0.4196 | 0.6289 | 0.4441 | 0.6922 | 0.4929 | 0.6790 | 0.4987 |
| step | 0.3730 | *0.1241* | 0.4539 | *0.1345* | 0.7090 | *0.1597* | 0.6085 | *0.1420* |
| tank | 0.7262 | 0.4008 | 0.3890 | 0.3767 | 0.4738 | 0.3926 | 0.4686 | 0.3903 |
| trade name | 0.6069 | 0.4242 | 0.6257 | 0.4519 | 0.3866 | 0.2330 | 0.3488 | 0.2042 |
| microwave | 0.6897 | 0.6949 | 0.6350 | 0.6884 | 0.8810 | 0.6676 | 0.9135 | 0.6587 |
| pot | 0.5060 | 0.5151 | 0.4797 | 0.5167 | 0.7731 | 0.4351 | 0.7853 | 0.3942 |
| animal | 0.5093 | 0.2945 | 0.4863 | 0.2946 | 0.7762 | 0.5719 | 0.7497 | 0.5512 |
| bicycle | 0.6692 | 0.5936 | 0.7153 | 0.5852 | 0.8083 | 0.3472 | 0.8002 | 0.3154 |
| lake | *0.0234* | *0.0002* | *0.0860* | *0.0001* | 0.7880 | 0.1997 | 0.8605 | 0.1997 |
| dishwasher | 0.6327 | 0.6098 | **0.8802** | 0.6309 | 0.9222 | 0.6692 | 0.8628 | 0.6224 |
| screen | 0.8307 | 0.4992 | 0.7236 | 0.5010 | 0.8256 | 0.6681 | 0.7968 | 0.6055 |
| blanket | 0.3294 | 0.3658 | 0.3940 | 0.3488 | 0.4154 | 0.2845 | 0.3097 | 0.1913 |
| sculpture | 0.5847 | 0.3240 | 0.5768 | 0.3206 | 0.8343 | 0.2758 | 0.8373 | 0.2807 |
| hood | 0.5973 | 0.6423 | 0.7049 | 0.6036 | 0.5808 | 0.6882 | 0.5094 | 0.5935 |
| sconce | 0.7347 | 0.5145 | 0.7293 | 0.5030 | 0.6491 | 0.4032 | 0.6226 | 0.3479 |
| vase | 0.6813 | 0.6166 | 0.6540 | 0.5930 | 0.7775 | 0.4159 | 0.7499 | 0.3745 |
| traffic light | 0.5853 | 0.6974 | 0.6836 | 0.6796 | 0.6773 | 0.3443 | 0.6240 | 0.3111 |
| tray | 0.6858 | 0.3117 | 0.6344 | 0.3108 | 0.5955 | 0.2606 | 0.5262 | 0.2263 |
| ashcan | 0.7771 | 0.4834 | 0.7593 | 0.4636 | 0.8191 | 0.3523 | 0.8039 | 0.3130 |
| fan | 0.6371 | 0.7163 | 0.6260 | 0.6670 | 0.7506 | 0.5828 | 0.7302 | 0.5493 |
| pier | **0.8935** | 0.3398 | 0.8430 | 0.3722 | 0.9237 | 0.4679 | 0.8972 | 0.4443 |
| crt screen | 0.4179 | *0.1100* | 0.5170 | *0.1263* | *0.3179* | *0.1178* | *0.2905* | *0.1186* |
| plate | 0.6950 | 0.5008 | 0.7655 | 0.4871 | 0.8537 | 0.3461 | 0.8517 | 0.3406 |
| monitor | 0.7184 | 0.3013 | 0.7747 | 0.2437 | *0.1745* | 0.2068 | *0.1027* | 0.1705 |
| bulletin board | 0.8242 | 0.4924 | 0.7822 | 0.4840 | 0.8263 | 0.4846 | 0.6846 | 0.4034 |
| shower | *0.3092* | 0.4837 | 0.5107 | 0.4570 | *0.1334* | 0.2059 | *0.1273* | 0.1660 |
| radiator | 0.7408 | 0.5331 | 0.7187 | 0.5401 | 0.8359 | 0.6907 | 0.8150 | 0.6724 |
| glass | 0.5036 | 0.5833 | 0.5382 | 0.5610 | 0.4016 | 0.2441 | 0.4431 | 0.2217 |
| clock | 0.4518 | 0.4883 | *0.3102* | 0.4624 | 0.6924 | 0.3115 | 0.6216 | 0.2572 |
| flag | 0.5878 | 0.6707 | 0.5870 | 0.6537 | 0.6334 | 0.3984 | 0.5834 | 0.3099 |

## F    Qualitative results on ADE20K

We show a few qualitative results on the ADE20K dataset for a couple of best and worst performing classes in Figure 10 and Figure 11, respectively. The figures show the original image, the class label being looked at, misclassified pixels, entropy, thresholded entropy values (using the largest difference technique), and the detection performance (green areas show correctly identified misclassified pixels, blue areas are false positives and red areas are false negatives). All of these results are from the Base setting of OneFormer ConvNeXt-L model.

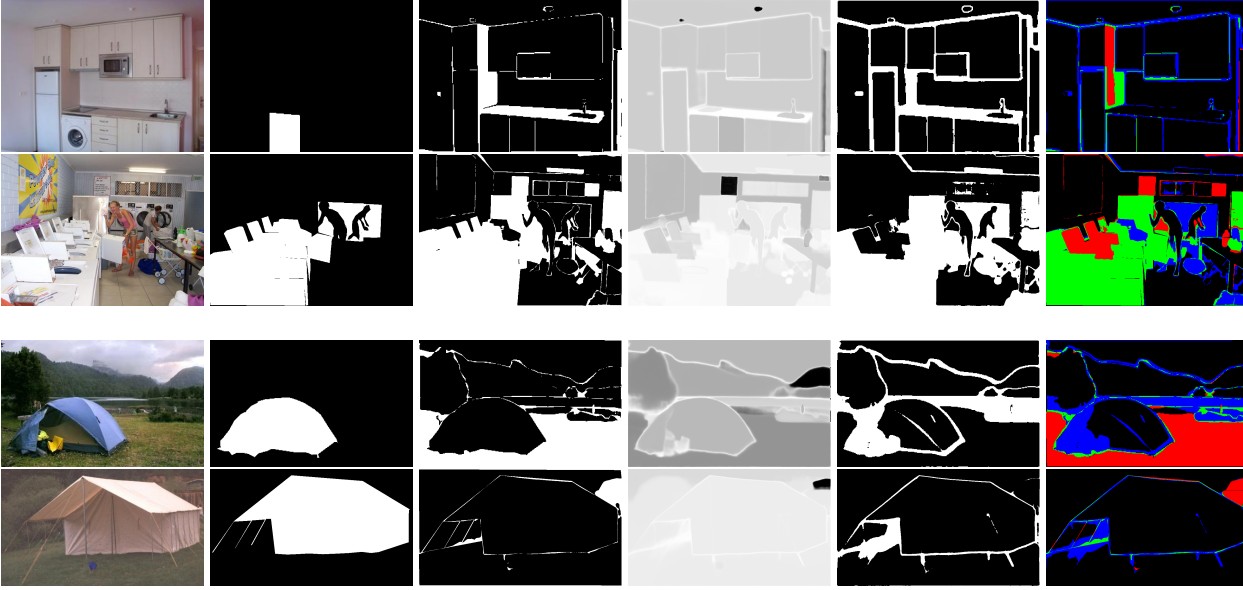

Figure 10: A few images showing a couple of best performing classes in the ADE20K dataset. The first two images include the washing machine class while the last two include the tent class. For each image, we show from left to right, the image, the highlighted class, misclassified pixels, entropy, thresholded entropy, detection mask.

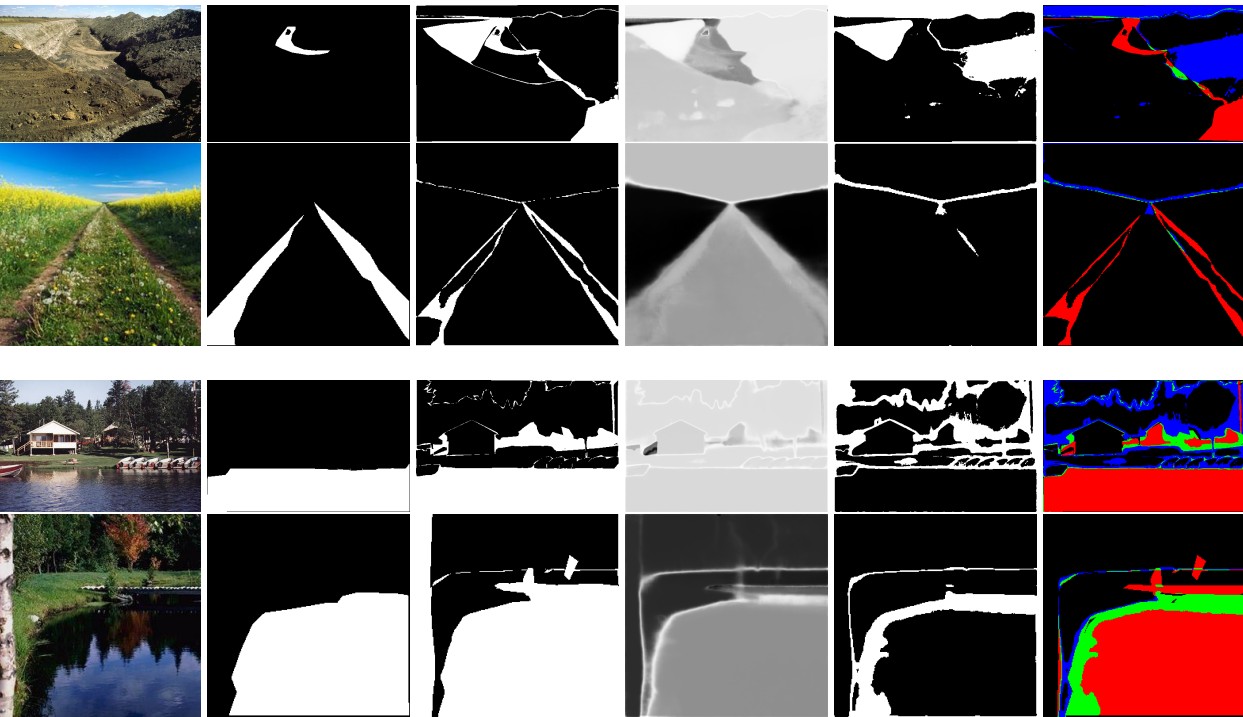

Figure 11: A few images showing a couple of worst performing classes in the ADE20K dataset. The first two images include dirt track class while the last two include the lake class. For each image, we show from left to right, the image, the highlighted class, misclassified pixels, entropy, thresholded entropy, detection mask.

