# OpenReview forum: "Black Box Uncertainty Analysis for Semantic Segmentation"
_TMLR — Rejected by TMLR_

### Review · Reviewer_RLBx · 2024-09-09

**Summary Of Contributions:**

The paper reports on the results of an experimental study that investigates how various uncertainty metrics correlate with misclassifications for the task of semantic segmentation. The paper explores how model calibration impacts the correlation, and investigates the effects of input domain shift and additive noise.

**Audience:**

Yes

**Claims And Evidence:**

No

**Requested Changes:**

1.	(Required). The paper should report an assessment of variability (bootstrap-based confidence intervals, for example).

2.	(Required). Statistical significance testing should be conducted to establish whether differences are meaningful. The correct statistical tests should be selected given the nature of the experiments, and there should be verification that necessary assumptions are met (or approximately met).

3.	(Required). The reporting should go beyond collapsed numbers such as AUROCs. At least some figures should be provided (e.g. ROCs). The figures should be designed to provide better insight into the key conclusions that can be drawn from the experiments. Currently the paper contains 8 tables in the main paper and at least 10 or 15 more in the appendix. This is a very ineffective way to communicate results.

4.	(Required). The discussion of the results should be enhanced to provide more meaningful insights into the outcomes of the experiments, and to provide increased clarity concerning why some techniques outperform, or why performance on some datasets is better.

5.	(Required). The techniques for choosing thresholds for each image should be re-designed and based on more principled techniques. Alternatively, thought should be given as to how to construct or choose uncertainty metrics that are more consistent across different images.

**Strengths And Weaknesses:**

Strengths

S1. The paper addresses an important task and conducts interesting experiments to explore how a variety of uncertainty metrics correlate with misclassifications for the task of semantic segmentation.

S2. The experiments are conducted for multiple datasets and models.

S3. The paper is clearly written and provides some interesting insights concerning performance.

Weaknesses

W1. The authors observe that the analyzed uncertainty metrics “vary quite a lot across different models and even across different images for a single model”. The attempts to address this are heuristic and there is no justification for them. The second technique appears to assume that the same number of misclassified pixels occurs for every image, and there is no analysis to support this conclusion. Without justified and principled processes for selecting a threshold, or a meaningful attempt to render the metric more stable across images, the value of a detection system based on the proposed uncertainty metrics becomes highly questionable.

W2, The paper does not report any assessment of the variability associated with the reported metrics.

W3. The paper does not assess whether any of the differences are statistically significant.

W4. The paper collapses results into single numbers (AUROCs) rather than displaying ROCs, which are much more informative. The same AUROC can be achieved in multiple ways. The portion of the ROC that corresponds to a high false alarm is often of little interest, because it does not align with an acceptable operating point for a detection system (e.g., if the system is falsely declaring 40 percent of the pixels to be misclassification, it is not useable). However, these regions can contribute significantly to AUROC measures.

W5. The analysis of the experimental results, the discussion, and the drawn conclusions are insufficiently insightful. For example, in the primary analysis, the commentary and conclusions do not extend beyond very simple observations from the reported results – “precision suffers quite a bit in these cases”, “recall improves at the cost of precision”, “we can get 60-70% recall using Entropy as an uncertainty metric”. For the calibration experiments, the only insight is that “AUROC scores using Entropy as the uncertainty metric do improve on the calibrated models”. Given that this is a paper reporting an experimental analysis, it is important that there is an attempt to answer “why?” questions, rather than just reporting performance numbers.

---

> ### Author Response · Authors · 2025-03-26
> **Rebuttal**
>
> We thank **Reviewer RLBx** for their time and valuable insights. We have taken their comments into consideration and have updated our paper. To facilitate easier perusal, we have highlighted all of the changes in the paper's main text in blue. We now address the weaknesses mentioned one by one.
>
> >W1. The authors observe that the analyzed uncertainty metrics “vary quite a lot across different models and even across different images for a single model”. The attempts to address this are heuristic and there is no justification for them. The second technique appears to assume that the same number of misclassified pixels occurs for every image, and there is no analysis to support this conclusion. Without justified and principled processes for selecting a threshold, or a meaningful attempt to render the metric more stable across images, the value of a detection system based on the proposed uncertainty metrics becomes highly questionable.
>
> Our first proposed dynamic thresholding technique assumes that the correctly classified and misclassified pixels are somewhat clustered. Thus, whenever we cross a threshold, we suddenly select a lot more pixels, that are correctly classified and see a large jump. To support this assumption, we check the difference in entropy within correctly classified pixels, within misclassified pixels and between correctly classified and misclassified pixels. We obtain the following results:
>
>     Within correctly classified: 0.1336
>     Within misclassified: 0.4346
>     Between correctly classified and misclassified: 0.6572
>
> These results suggest that they are indeed clustered. In addition to our existing technique, we have now added K-Means as an alternative dynamic thresholding technique. The results for the same are provided in Table 3 on page 11 of the revised manuscript. We observe that this is a more general approach and works well for DRN and Segformer. Unfortunately, it is worse for OneFormer where our dynamic thresholding works better. We have included all these results in the updated paper.
>
> For the second technique, the general assumption is that most of the images will not have too many misclassifications for a good segmentation model. We concede that this technique is not particularly elegant. However, it helps when we are somewhat certain that our errors will be within a certain limit.
>
> >W2, The paper does not report any assessment of the variability associated with the reported metrics.
>
> We have added bootstrap-based confidence intervals for our primary results and now report the bootstrap mean along with the intervals. Please check Tables 1, 2 and 3 in the revised paper. We hope that this revision addresses the concerns raised effectively.
>
> >W3. The paper does not assess whether any of the differences are statistically significant.
>
> Statistical testing to see whether the differences between various uncertainties are significant or not has been shown using Wilcoxon signed-rank tests. Please check Table 4 for the results in the revised version. As can be observed, most of the results are statistically significant. We hope that this revision addresses the concerns raised effectively.
>
> >W4. The paper collapses results into single numbers (AUROCs) rather than displaying ROCs, which are much more informative. The same AUROC can be achieved in multiple ways. The portion of the ROC that corresponds to a high false alarm is often of little interest, because it does not align with an acceptable operating point for a detection system (e.g., if the system is falsely declaring 40 percent of the pixels to be misclassification, it is not useable). However, these regions can contribute significantly to AUROC measures.
>
> We have updated the AUROC and AUPRC reporting to include the complete curves instead of just the collapsed values. Please check Figures 4, 5, 6, 7 and 8. We have also provided the collapse AU values within the plots themselves. We hope this addresses the concerns raised.
>
> >W5. The analysis of the experimental results, the discussion, and the drawn conclusions are insufficiently insightful. For example, in the primary analysis, the commentary and conclusions do not extend beyond very simple observations from the reported results – “precision suffers quite a bit in these cases”, “recall improves at the cost of precision”, “we can get 60-70% recall using Entropy as an uncertainty metric”. For the calibration experiments, the only insight is that “AUROC scores using Entropy as the uncertainty metric do improve on the calibrated models”. Given that this is a paper reporting an experimental analysis, it is important that there is an attempt to answer “why?” questions, rather than just reporting performance numbers.
>
> Thanks for pointing out. We have added a discussion section to the updated paper on page 14 that provides a detailed analysis of the experimental results. Please let us know whether this addresses the point raised.

---

### Review · Reviewer_EDUU · 2024-12-29

**Summary Of Contributions:**

This paper proposes to measure the uncertainty of segmentation prediction to evaluate the correctness of the model's output at test time.  This paper shows that uncertainty metrics, such *probability score*, *entropy*, *variance* and *Bayesian Active Learning by Disagreemen*, are highly correlated with the correctness of the model's output, even under a wide range of scenarios.

**Audience:**

Yes

**Claims And Evidence:**

No

**Requested Changes:**

1. I'd suggest the authors to encompass more discussion of the prior works, that also use *uncertainty* to measure the correctness of the model's output, in both introduction and related work.

2. The authors need to explain my question of "why doesn't the summation of correctly-classified and mis-classified pixels equal to 100%"?

3. The authors need to include comparison to other kinds of baselines that also measure the correctness of the model's output.

**Strengths And Weaknesses:**

Strengths:

1. The idea of using "uncertainty* to measure the correctness of the model's output is technically sound.

---

Weaknesses:

1. Although the idea is technically sound, it has already been demonstrated in prior works.  For example, in domain adaptation, [1] calculate the entropy of segmentation prediction to select pseudo labels for unlabeled images; in active learning, [2] measures the entropy of segmentation predict to select camera view points.  In my opinion, I believe "the entropy of model prediction highly correlates with the correctness of the output" is a well known fact in the machine learning / computer vision community.  This paper does not demonstrate newer ideas than these known facts.

2. This paper lacks thorough literature review.  As pointed in the `[Weakness 1]`, it doesn't discuss prior works that measures uncertainty in other research topics (e.g. active learning / domain adaptation / semi-supervised learning).

3. Figure 3 is confusion.  I'm confused why doesn't the summation of correctly-classified and mis-classified pixels equal to 100%?

4. The experimental results do not compare to other baselines.  This paper proposes to measure the correctness of the model's prediction using the *uncertainty* of the output.  However, the experiments do not compare to other methods, but ablate the effectiveness of different forms of uncertainty.  It's impossible to draw the conclusion if *uncertain* is a good measurement of the correctness of the model's prediction.

5. Similar to `[Weakness 4]`, the experimental results are confusing without thorough comparison.  In Table 5, the precision is no higher than 40%.  Without comparison to other baselines, it's easier to draw the conclusion that the proposed method doesn't work.

---

Reference:
[1] Uncertainty-aware Pseudo Label Refinery for Domain Adaptive Semantic Segmentation.  Wang et al.  ICCV 2021.
[2] ViewAL: Active Learning With Viewpoint Entropy for Semantic Segmentation.  Siddiqui et al.  CVPR 2020.

---

> ### Author Response · Authors · 2025-03-26
> **Rebuttal**
>
> We thank **Reviewer EDUU** for their time and valuable insights. We have taken their comments into consideration and have updated our paper. To facilitate easier perusal, we have highlighted all of the changes in the paper's main text in blue. We now address the weaknesses mentioned one by one.
>
> >1. Although the idea is technically sound, it has already been demonstrated in prior works. For example, in domain adaptation, [1] calculate the entropy of segmentation prediction to select pseudo labels for unlabeled images; in active learning, [2] measures the entropy of segmentation predict to select camera view points. In my opinion, I believe "the entropy of model prediction highly correlates with the correctness of the output" is a well known fact in the machine learning / computer vision community. This paper does not demonstrate newer ideas than these known facts.
> >
> >2. This paper lacks thorough literature review. As pointed in the `[Weakness 1]`, it doesn't discuss prior works that measures uncertainty in other research topics (e.g. active learning / domain adaptation / semi-supervised learning).
> >
> >Reference: [1] Uncertainty-aware Pseudo Label Refinery for Domain Adaptive Semantic Segmentation. Wang et al. ICCV 2021. [2] ViewAL: Active Learning With Viewpoint Entropy for Semantic Segmentation. Siddiqui et al. CVPR 2020.
>
> We thank the reviewer for bringing these papers to our attention and have added them to the introduction and related works section. Please check Sections 1 and 2 on pages 2 and 3.
>
> While this type of work has been done before in general, we believe we are the first to show them specifically for a wide variety of segmentation models. We hope this addresses these concerns effectively.
>
> >3. Figure 3 is confusion. I'm confused why doesn't the summation of correctly-classified and mis-classified pixels equal to 100%?
>
> The graph does not show the summation of correctly classified and misclassified pixels. Instead, it shows what portion of them are accurately captured/classified if we threshold uncertainty at a particular value. Suppose one chooses a threshold value on the x-axis. In that case, the graph shows what fraction of correctly classified pixels have uncertainty less than that value and what fraction of misclassified pixels have uncertainty higher than that value. We have reworded the figure's caption and relevant paragraph in the text for better clarity. Please check page 5. We hope this clarifies this issue.
>
> >4. The experimental results do not compare to other baselines. This paper proposes to measure the correctness of the model's prediction using the uncertainty of the output. However, the experiments do not compare to other methods, but ablate the effectiveness of different forms of uncertainty. It's impossible to draw the conclusion if uncertain is a good measurement of the correctness of the model's prediction.
> >
> >5. Similar to `[Weakness 4]`, the experimental results are confusing without thorough comparison. In Table 5, the precision is no higher than 40%. Without comparison to other baselines, it's easier to draw the conclusion that the proposed method doesn't work.
>
> Unfortunately, comparison with other baselines is difficult since other works either don't use the same networks or don't use the same datasets. At best, we can make very crude comparisons to Xia et al. `[1]` and Rahman et al. `[2]` where the AUROC, AUPRC Error and AUPRC Success are in the `90%`, `50-70%`, and `95%` ranges, whereas ours are in `70-90%`, `40%` and `95%` ranges respectively. However, the referenced papers both train models specifically for such tasks, whereas we only work with pre-trained models and test-time uncertainties. To the best of our knowledge, no such systematic study of prediction of failure analysis for semantic segmentation has been carried out. We hope this addresses these issues effectively.
>
> `[1]` Yingda Xia, Yi Zhang, Fengze Liu, Wei Shen, and Alan L Yuille. Synthesize then compare: Detecting failures and anomalies for semantic segmentation. In _Computer Vision–ECCV 2020: 16th European Conference, Glasgow, UK, August 23–28, 2020, Proceedings, Part I 16_, pp. 145–161. Springer, 2020.
> `[2]` Quazi Marufur Rahman, Niko Sünderhauf, Peter Corke, and Feras Dayoub. Fsnet: A failure detection framework for semantic segmentation. _IEEE Robotics and Automation Letters_, 7(2):3030–3037, 2022.

---

### Review · Reviewer_nEzJ · 2025-03-12

**Summary Of Contributions:**

This paper explores simple yet effective methods for estimating uncertainty in semantic segmentation models. The authors demonstrate that pixel-wise entropy serves as a strong uncertainty measure for these models, capturing the likelihood of misclassification at a fine-grained level. Additionally, they propose heuristic-based techniques that aggregate pixel-wise entropy values across an image to identify misclassified regions. The study provides empirical validation on the Cityscapes dataset, highlighting the correlation between high entropy and misclassification and evaluating thresholding strategies to improve uncertainty estimation.

**Audience:**

No

**Claims And Evidence:**

Yes

**Requested Changes:**

Unfortunately, it seems to me that the primary insight here is not generally surprising or interesting to a general audience. The proposed Dynamic Thresholding method is not widely applicable to real-world use cases (or even diverse datasets).

**Strengths And Weaknesses:**

## Strengths
- The paper tackles the important problem of uncertainty estimation in semantic segmentation, a crucial aspect of deploying models in real-world applications such as autonomous driving and medical imaging. The emphasis on methods that are architecture-agnostic enhances the potential applicability of these techniques across different segmentation models.
- The study systematically evaluates entropy as an uncertainty measure, first through qualitative visualizations and then quantitatively with threshold-based heuristics. The structured approach strengthens the credibility of the findings and demonstrates that entropy is indeed correlated with misclassifications.
- The paper presents its findings in a clear and structured manner, making it easy to follow the motivation, methodology, and results. The visualizations of entropy maps and misclassified pixels provide an intuitive understanding of the proposed approach.


## Weaknesses
- The observation that entropy correlates with misclassification is well-established in broader machine learning literature. While this paper validates this phenomenon in the context of semantic segmentation, it does not introduce fundamentally new insights or methods. This may limit its appeal to the TMLR audience.
- The proposed "Dynamic Thresholding" methods are highly dependent on dataset-specific assumptions. For example, method (1) relies on a fixed ratio of predicted misclassified pixels to total pixels, which is unlikely to generalize across diverse datasets where misclassification rates vary significantly. Similarly, method (2) hard-codes a maximum misclassification ratio, further restricting generalizability. Without adaptation mechanisms for different datasets, these methods may have limited real-world applicability.
- While the paper investigates entropy as an uncertainty measure, Table 2 suggests that entropy-based methods underperform compared to alternative uncertainty quantification techniques. This raises concerns about whether entropy is the most effective approach for estimating misclassification in segmentation models. At the very least, a deeper exploration of why entropy performs worse than other methods and potential ways to improve it might be needed here.
- The empirical results focus predominantly on the Cityscapes dataset, and it remains unclear whether the proposed techniques would perform similarly well on other segmentation benchmarks. Given that the utility of entropy-based uncertainty estimation is likely to depend on dataset characteristics (e.g., class imbalance, annotation noise), an evaluation on multiple datasets would provide stronger evidence of generalizability.

---

> ### Author Response · Authors · 2025-03-26
> **Rebuttal**
>
> We thank **Reviewer nEzJ** for their time and valuable insights. We have taken their comments into consideration and have updated our paper. To facilitate easier perusal, we have highlighted all of the changes in the paper's main text in blue. We now address the weaknesses mentioned one by one.
>
> >The observation that entropy correlates with misclassification is well-established in broader machine learning literature. While this paper validates this phenomenon in the context of semantic segmentation, it does not introduce fundamentally new insights or methods. This may limit its appeal to the TMLR audience.
>
> We have added a discussion section to the updated paper on page 14 that provides a detailed analysis of the experimental results. Please let us know whether this addresses the point raised.
>
> >The proposed "Dynamic Thresholding" methods are highly dependent on dataset-specific assumptions. For example, method (1) relies on a fixed ratio of predicted misclassified pixels to total pixels, which is unlikely to generalize across diverse datasets where misclassification rates vary significantly. Similarly, method (2) hard-codes a maximum misclassification ratio, further restricting generalizability. Without adaptation mechanisms for different datasets, these methods may have limited real-world applicability.
> >
> >The empirical results focus predominantly on the Cityscapes dataset, and it remains unclear whether the proposed techniques would perform similarly well on other segmentation benchmarks. Given that the utility of entropy-based uncertainty estimation is likely to depend on dataset characteristics (e.g., class imbalance, annotation noise), an evaluation on multiple datasets would provide stronger evidence of generalizability.
>
> We have added results on the COCO dataset for the OneFormer Swin-L model. Unfortunately, we do not train the models ourselves and only use whatever the authors have made available since one of our primary goals was to estimate existing models' performance without any training. Due to brevity, most of the results for COCO have been added in the appendix. Please check Tables 8, 11 and 15 on pages 18, 22 and 27, respectively. We hope this address the concerns regarding datasets.
>
> >While the paper investigates entropy as an uncertainty measure, Table 2 suggests that entropy-based methods underperform compared to alternative uncertainty quantification techniques. This raises concerns about whether entropy is the most effective approach for estimating misclassification in segmentation models. At the very least, a deeper exploration of why entropy performs worse than other methods and potential ways to improve it might be needed here.
>
> While it is true that simple entropy does not always perform the best, it generally always performs well enough that more involved techniques such as BALD, which requires multiple forward passes to generate multiple segmentations, may not be worthwhile. We also note that entropy mostly underperforms in the case of OneFormer, which has unnaturally high entropy values, as discussed in Section 3. We have added possible reasons why OneFormer is an outlier in the newly added Discussion section on page 14. We hope this addresses the concerns raised effectively.

---

### Decision · Action_Editor_o45C · 2025-04-21

**Recommendation:** Reject

**Comment:**

Overall, the main reason for rejection is that the paper does not corroborate the claims it makes with sufficient evidence.
As reviewers RLBx and neJz have pointed out, the experiments involve too few datasets and models, and the dynamic thresholding method is highly cumbersome to use, and the paper does not offer any novel insights. This is fine -- at TMLR, papers do not get penalized for lack of novelty, but the bar for acceptance at TMLR is sufficient soundness. This paper unfortunately seems to lack in this aspect, as pointed out by reviewer RLBx.

**Audience:**

Audience would be the section of the community interested in uncertainty estimation and Computer vision.

**Claims And Evidence:**

The main claims in the paper are:
C2. “Through several experiments on three different models across three datasets, we show that simple measures such as entropy can be used to capture misclassification with high recall.”

C3. We can improve upon our results by using calibrated models.

C4. “We show that these techniques work even when we consider domain shifts … or noise … in the input.”

However, multiple reviewers have pointed out that these claims are not sufficiently well-supported by evidence. Either the claims are supported on very few datasets or models, or there is low precision (for C2) so the results are not very meaningful. Consequently I do not think this paper passes the bar for TMLR and I would recommend a rejection.

**Resubmission Of Major Revision:**

The authors may consider submitting a major revision at a later time.